# What Makes a Reward Model a Good Teacher? An Optimization Perspective

**Noam Razin, Zixuan Wang, Hubert Strauss, Stanley Wei,
Jason D. Lee, Sanjeev Arora**

Princeton Language and Intelligence, Princeton University

## Abstract

The success of *Reinforcement Learning from Human Feedback (RLHF)* critically depends on the quality of the reward model. However, while this quality is primarily evaluated through *accuracy*, it remains unclear whether accuracy fully captures what makes a reward model an effective teacher. We address this question from an optimization perspective. First, we prove that regardless of how accurate a reward model is, if it induces low *reward variance*, then the RLHF objective suffers from a flat landscape. Consequently, even a perfectly accurate reward model can lead to extremely slow optimization, underperforming less accurate models that induce higher reward variance. We additionally show that a reward model that works well for one language model can induce low reward variance, and thus a flat objective landscape, for another. These results establish a fundamental limitation of evaluating reward models solely based on accuracy or independently of the language model they guide. Experiments using models of up to 8B parameters corroborate our theory, demonstrating the interplay between reward variance, accuracy, and reward maximization rate. Overall, our findings highlight that beyond accuracy, a reward model needs to induce sufficient variance for efficient optimization.

## 1 Introduction

Training safe and helpful language models requires aligning them with desirable traits that are difficult to specify explicitly, be it through examples or hand-crafted reward functions. The widely adopted *Reinforcement Learning from Human Feedback (RLHF)* pipeline [11, 95, 55, 6] circumvents this difficulty through a two-step approach. It first trains a *reward model* $r_{\mathrm{RM}}$ based on access to preference data, under the assumption that this data reflects an unknown *ground truth reward* $r_{\mathrm{G}}$, which encodes said desirable traits. Then, the language model $\pi_\theta$ (referred to as a *policy*) is aligned by maximizing $r_{\mathrm{RM}}$ via *policy gradient* methods such as PPO [68], RLOO [42, 2], and GRPO [69].

Although the success of RLHF depends heavily on the quality of the reward model, it remains unclear how this quality should be measured. Current benchmarks for evaluating reward models focus mostly on *accuracy* [44, 93, 25, 47]. Namely, for a dataset of prompts, each associated with outputs ranked according to human (or AI) preferences, accuracy measures the proportion of output pairs that the reward model ranks correctly. Accuracy is a natural yardstick. It quantifies the extent to which maximizing the proxy reward $r_{\mathrm{RM}}$ is likely to increase the ground truth reward $r_{\mathrm{G}}$ (given that $r_{\mathrm{G}}$ underlies the preference rankings in the dataset). However, recent empirical evidence suggests that, by itself, accuracy may not be indicative of how good a reward model is: [37, 10, 80] observed that more accurate reward models do not necessarily yield stronger language models after RLHF. So,

*what makes a reward model a good teacher for RLHF?*

39th Conference on Neural Information Processing Systems (NeurIPS 2025).

Figure 1: Illustration of how accuracy (Definition 1) and reward variance (Definition 2) affect the RLHF objective landscape (Equation (1)). Accuracy and reward variance capture distinct aspects of a reward model: the former controls alignment with the ground truth reward, while the latter determines the flatness of the objective landscape. The lower the accuracy, the more prone a reward model is to *reward hacking* [5, 70, 57, 26] — directions considered beneficial by the reward model may not increase the ground truth reward. On the other hand, even if the reward model is perfectly accurate, low reward variance implies a flat landscape that hinders the efficiency of policy gradient methods, as we prove in Section 3 and demonstrate empirically in Section 4.

We address the question above from an optimization perspective. We begin by proving that, regardless of how accurate a reward model is, it can induce a flat objective landscape that hinders optimization. Specifically, if $r_{\mathrm{RM}}$ induces low *reward variance* (Definition 2), which occurs when it does not sufficiently separate outputs that are probable under $\pi_\theta$, then both the proxy reward $r_{\mathrm{RM}}$ and the ground truth reward $r_{\mathrm{G}}$ increase at an extremely slow rate during policy gradient (Section 3.2). This result builds on a connection between reward variance and the gradient of the RLHF objective [64], and reveals that an effective reward model needs to ensure the variance is not too low.[1] See Figure 1 for an illustration of how accuracy and reward variance affect the RLHF objective landscape.

We then prove two implications of the relation between reward variance and optimization rate under policy gradient. First, more accurate reward models are not necessarily better teachers (Section 3.3). In particular, since reward variance is not tied to accuracy, even a perfectly accurate reward model can lead to arbitrarily slow *ground truth reward* maximization compared to a relatively inaccurate model. Second, for different language models, different rewards models work better (Section 3.4). The core insight behind this result is that the same reward model can induce high reward variance for one language model, yet low reward variance, and consequently a flat objective landscape, for another. Taken together, Sections 3.3 and 3.4 establish a fundamental limitation of evaluating reward models solely based on accuracy or in isolation from the language model they guide [44, 93, 25, 47].

Experiments (Section 4) over models of up to 8B parameters and standard RLHF datasets corroborate our theory, demonstrating the interplay between reward variance, accuracy, and reward maximization rate. Remarkably, we find that even when the ground truth reward is accessible, using a proxy reward model can lead to better performance if it induces a higher reward variance. Overall, our results highlight that alongside being accurate, a reward model needs to induce sufficient reward variance for enabling efficient optimization. We hope insights from this work will inform improved methodologies for reward model training and evaluation that account for properties beyond accuracy.

**Related work.** We discuss related work throughout and defer an extended account to Appendix A.

## 2  Preliminaries

Let $\mathcal{V}$ be a finite vocabulary of tokens and $\mathcal{V}^*$ denote the set of all finite-length token sequences. We use $\pi_\theta$ to denote a language model, parameterized by $\theta$, that receives a prompt $\mathbf{x} \in \mathcal{X}$ and produces a

---

[1]Reward variance (Definition 2) should not be confused with the variance of gradient estimates [30], which is a more common notion of variance in the reinforcement learning literature.

distribution over outputs $\mathbf{y} \in \mathcal{Y}$, where $\mathcal{X}, \mathcal{Y} \subseteq \mathcal{V}^*$ are the sets of possible input prompts and outputs, respectively. Following convention in the RLHF literature, we also refer to $\pi_\theta$ as a *policy* (*cf.* [55]).

## 2.1 Aligning Language Models via Reinforcement Learning

The standard RLHF pipeline for aligning language models consists of two main steps, outlined below (*cf.* [95, 71, 55, 6]). These steps are usually preceded by a supervised finetuning (SFT) phase, in which the base language model is trained with a next-token prediction loss on pairs of input prompts and desirable outputs. The resulting model, denoted $\pi_{\theta_{\mathrm{ref}}}$, serves as the initial policy for RLHF.

**Step 1: Reward model training or selection.** RLHF assumes that there exists an unknown *ground truth reward* $r_\mathrm{G} : \mathcal{X} \times \mathcal{Y} \to [-1, 1]$, which encodes (at least some aspects of) human preferences.[2] For a given prompt $\mathbf{x} \in \mathcal{X}$, we wish to maximize the expected ground truth reward that our policy $\pi_\theta$ achieves, *i.e.* $\mathbb{E}_{\mathbf{y} \sim \pi_\theta(\cdot|\mathbf{x})}[r_\mathrm{G}(\mathbf{x}, \mathbf{y})]$. However, since $r_\mathrm{G}$ is not directly accessible, the first step of RLHF is to train a proxy *reward model* $r_\mathrm{RM} : \mathcal{X} \times \mathcal{Y} \to [-1, 1]$. This is typically done through a Bradley-Terry [9] log-likelihood loss over a dataset of human preferences. Alternatively, one can forgo reward model training and choose a publicly available model instead (*e.g.*, see [76, 87, 44]).

**Step 2: Reward maximization via policy gradient.** With a reward model $r_\mathrm{RM}$ in hand, the policy $\pi_\theta$ is updated by using *policy gradient* methods, such as PPO [68], RLOO [42, 2], and GRPO [69], to maximize:

$$\phi_{\mathrm{RLHF}}(\theta) := \mathbb{E}_{\mathbf{x} \sim \mathcal{S}}\Big[\mathbb{E}_{\mathbf{y} \sim \pi_\theta(\cdot|\mathbf{x})}\big[r_{\mathrm{RM}}(\mathbf{x}, \mathbf{y})\big] - \lambda \cdot \mathrm{KL}\big(\pi_\theta(\cdot|\mathbf{x})||\pi_{\theta_{\mathrm{ref}}}(\cdot|\mathbf{x})\big)\Big]. \tag{1}$$

Here, $\mathcal{S} \subseteq \mathcal{X}$ is a training set of prompts, $\mathbf{x} \sim \mathcal{S}$ denotes sampling $\mathbf{x}$ uniformly from $\mathcal{S}$, and $\lambda \geq 0$ is a KL regularization coefficient that allows controlling how much $\pi_\theta$ will deviate from the initial policy $\pi_{\theta_{\mathrm{ref}}}$ (*i.e.*, the policy produced by the SFT phase). The hope is that by increasing the expected proxy reward we will also increase the expected ground truth reward, while the KL regularization helps preserve capabilities acquired during pretraining and SFT.

## 2.2 Accuracy

Currently, reward models are primarily evaluated through *accuracy* [44, 93, 25, 47], which measures their agreement with the ground truth reward in terms of ranking output pairs.

**Definition 1.** Given a prompt $\mathbf{x} \in \mathcal{X}$, the *accuracy* of a reward model $r_\mathrm{RM} : \mathcal{X} \times \mathcal{Y} \to [-1, 1]$ with respect to a distribution $\mathcal{D}$ over unordered output pairs is defined by:

$$\mathrm{acc}_{\mathbf{x}, \mathcal{D}}(r_\mathrm{RM}) := \mathbb{E}_{\{\mathbf{y}, \mathbf{y}'\} \sim \mathcal{D}}\Big[\mathbb{1}\big[\mathrm{sign}\big(r_\mathrm{RM}(\mathbf{x}, \mathbf{y}) - r_\mathrm{RM}(\mathbf{x}, \mathbf{y}')\big) = \mathrm{sign}(r_\mathrm{G}(\mathbf{x}, \mathbf{y}) - r_\mathrm{G}(\mathbf{x}, \mathbf{y}'))\big]\Big],$$

where $r_\mathrm{G}$ is the ground truth reward, $\mathbb{1}[\cdot]$ is an indicator function, and $\mathrm{sign} : \mathbb{R} \to \{-1, 0, 1\}$ is the sign function.[3] For a set of prompts, accuracy refers to the mean accuracy over the set.

**On which prompts and output pairs should accuracy be measured?** Accuracy is often estimated using benchmarks containing diverse prompts with ranked output pairs [44, 93, 25, 47]. However, during RLHF the reward model is applied exclusively to prompts in the policy gradient training set $\mathcal{S}$ and on-policy outputs (*i.e.*, outputs sampled from the current policy). As a result, when prompts in $\mathcal{S}$ or on-policy outputs differ substantially from those in existing (off-policy) benchmarks, the estimated accuracy may be misleading (*cf.* [80]). To avoid this potential issue, our analysis considers the accuracy of a reward model over $\mathcal{S}$ and supports any output pair distribution.[4]

## 2.3 Reward Variance

Aside from accuracy, another quantity key to our work is the *reward variance* induced by a reward model for a policy and prompt. As elaborated in Section 3, our theory builds on a recent connection between the gradient of the RLHF objective (Equation (1)) and reward variance [64].

---

[2]We assume rewards lie in $[-1, 1]$ without loss of generality; any bounded interval can be used instead.

[3]That is, $\mathrm{sign}(u) = 1$ if $u > 0$, $\mathrm{sign}(u) = 0$ if $u = 0$, and $\mathrm{sign}(u) = -1$ otherwise.

[4]The prompts relevant for accuracy evaluation may vary across alignment procedures. For example, Best-of-N [52] applies the reward model to prompts encountered at test-time.

**Definition 2.** Given a policy $\pi_\theta$, prompt $\mathbf{x} \in \mathcal{X}$, and reward model $r_{\mathrm{RM}} : \mathcal{X} \times \mathcal{Y} \to [-1, 1]$, the *reward variance* induced by $r_{\mathrm{RM}}$ for $\pi_\theta$ and $\mathbf{x}$ is defined by:

$$\mathrm{Var}_{\mathbf{y} \sim \pi_\theta(\cdot|\mathbf{x})}[r_{\mathrm{RM}}(\mathbf{x}, \mathbf{y})] := \mathbb{E}_{\mathbf{y} \sim \pi_\theta(\cdot|\mathbf{x})}\left[\left(r_{\mathrm{RM}}(\mathbf{x}, \mathbf{y}) - \mathbb{E}_{\mathbf{y}' \sim \pi_\theta(\cdot|\mathbf{x})}[r_{\mathrm{RM}}(\mathbf{x}, \mathbf{y}')]\right)^2\right].$$

Intuitively, reward variance measures how well $r_{\mathrm{RM}}$ separates the rewards assigned to outputs that are probable under $\pi_\theta(\cdot|\mathbf{x})$. This becomes apparent from the fact that:

$$\mathrm{Var}_{\mathbf{y} \sim \pi_\theta(\cdot|\mathbf{x})}[r_{\mathrm{RM}}(\mathbf{x}, \mathbf{y})] = \frac{1}{2}\,\mathbb{E}_{\mathbf{y},\mathbf{y}' \overset{\mathrm{i.i.d.}}{\sim} \pi_\theta(\cdot|\mathbf{x})}\left[\left(r_{\mathrm{RM}}(\mathbf{x}, \mathbf{y}) - r_{\mathrm{RM}}(\mathbf{x}, \mathbf{y}')\right)^2\right],$$

and stands in contrast to accuracy, which depends only on how outputs are ranked, disregarding the degree of separation between their rewards.

**Reward scale.** The scale of rewards affects their variance. For example, multiplying all rewards by $c \geq 0$ scales reward variance by a factor of $c^2$. It is common practice to account for differences in scale across reward models through normalization (*e.g.*, as done in the experiments of Section 4). This ensures that reward variance meaningfully measures the degree of separation between outputs, and not merely the scale of rewards.

## 3 Theory: Optimization Perspective on What Makes a Good Reward Model

Our analysis considers the time it takes for the expected ground truth reward $r_{\mathrm{G}}$ to increase by a desired amount when maximizing the RLHF objective (Equation (1)) via policy gradient. The shorter the time, the better. We prove that, if a reward model $r_{\mathrm{RM}}$ induces low reward variance for the initial policy, then both $r_{\mathrm{RM}}$ and $r_{\mathrm{G}}$ increase at a slow rate due to a flat objective landscape (Section 3.2). Thus, $r_{\mathrm{RM}}$ needs to ensure that the reward variance is not too low for efficient optimization.

We then establish two main implications of the relation between reward variance and optimization rate. First, since reward variance is not tied to accuracy, more accurate reward models are not necessarily better teachers (Section 3.3). Second, since the same reward model can induce high reward variance for one policy but low reward variance for another, for different initial policies, different reward models work better (Section 3.4). Our results formalize a limitation inherent to benchmarks evaluating reward models solely based on accuracy or independently of the language model they guide [44, 93, 25, 47].

Section 3.1 lays out the technical setting. For conciseness, Sections 3.2 to 3.4 include abridged versions of the theoretical results. Detailed theorem statements are provided in Appendix B.

### 3.1 Technical Setting

In line with standard practice, we take $\mathcal{Y}$ to be the set of sequences with length bounded by some maximal value $L \in \mathbb{N}$. Without loss of generality, we may treat $\mathcal{Y}$ as $\mathcal{V}^L$ — the set of sequences of length equal to $L$ — where shorter sequences are padded to length $L$ with a dedicated token.

**Policy parameterization.** Given a prompt $\mathbf{x} \in \mathcal{X}$, language models usually produce a distribution over output sequences $\mathbf{y} \in \mathcal{Y}$ autoregressively:

$$\textit{(general autoregressive policy)} \quad \pi_\theta(\mathbf{y}|\mathbf{x}) = \prod_{l=1}^{|\mathbf{y}|} \pi_\theta(\mathbf{y}_l|\mathbf{x}, \mathbf{y}_{<l}) = \prod_{l=1}^{|\mathbf{y}|} \mathrm{softmax}\big(f_\theta(\mathbf{x}, \mathbf{y}_{<l})\big)_{\mathbf{y}_l}, \quad (2)$$

where $f_\theta : \mathcal{V}^* \to \mathbb{R}^{|\mathcal{V}|}$ is a function, differentiable with respect to its parameters $\theta$ (*e.g.*, a neural network), that maps a sequence of tokens to logits for the next-token distribution, $\mathbf{y}_l$ and $\mathbf{y}_{<l}$ are the $l$th token and first $l-1$ tokens of $\mathbf{y}$, respectively, and $\mathrm{softmax}(\mathbf{z})_v := \exp(\mathbf{z}_v)/\sum_{v' \in \mathcal{V}} \exp(\mathbf{z}_{v'})$ for $\mathbf{z} \in \mathbb{R}^{|\mathcal{V}|}$. The lower bound in Section 3.2 on the time it takes for the expected proxy and ground truth rewards to increase applies to policies parameterized according to Equation (2).

The results in Sections 3.3 and 3.4 require proving separations in optimization time between different reward models. Due to the non-concavity of the RLHF objective (Equation (1)), providing guarantees of efficient optimization has proven challenging (*cf.* [1, 45]). Thus, existing analyses often consider

a *tabular policy* parameterization (*e.g.*, [49, 48, 1]), which can be viewed as a special case of Equation (2), wherein each output is assigned its own trainable logit. That is, for $\theta \in \mathbb{R}^{|\mathcal{Y}| \times |\mathcal{X}|}$:

$$(tabular\ policy) \quad \pi_\theta(\mathbf{y}|\mathbf{x}) = \text{softmax}(\theta_{:,\mathbf{x}})_\mathbf{y}, \tag{3}$$

where $\theta_{:,\mathbf{x}} \in \mathbb{R}^{|\mathcal{Y}|}$ is the column of $\theta$ corresponding to $\mathbf{x}$. Following prior work, Sections 3.3 and 3.4 consider tabular policies. We note that despite the apparent simplicity of tabular policies, the RLHF objective remains non-concave.

**Optimization.** RLHF is typically carried out with small learning rates [55, 84, 2, 80]. Accordingly, we analyze policy gradient at the small learning rate limit, *i.e. gradient flow*:

$$\frac{d}{dt}\theta(t) = \nabla\phi_{\text{RLHF}}(\theta(t)) \ , \ t \geq 0, \tag{4}$$

where $\theta(t)$ denotes the parameters of the policy $\pi_{\theta(t)}$ at time $t$ of training and $\theta(0) = \theta_{\text{ref}}$.

Overall, the analyzed setting involves two main simplifications. First, some of our results are proven for tabular policies. Second, we assume access to exact gradients while, in practice, the large output space of language models prohibits exact computation of $\nabla\phi_{\text{RLHF}}(\theta)$. As Section 4 verifies empirically, the implications of our theory apply to standard language models trained via policy gradient methods that use sample-based estimates of the gradient (*e.g.*, RLOO and GRPO). Nonetheless, relaxing these simplifications can be an interesting challenge for future work.

## 3.2    Low Reward Variance Implies Slow Reward Maximization

In this section, we lower bound the time required for the expected reward, measured with respect to any reward function (including $r_{\text{RM}}$ and $r_{\text{G}}$), to increase by an additive constant. Theorem 1 shows that this time grows inversely with $\mathbb{E}_{\mathbf{x} \sim \mathcal{S}}[\text{Var}_{\mathbf{y} \sim \pi_{\theta(0)}(\cdot|\mathbf{x})}[r_{\text{RM}}(\mathbf{x}, \mathbf{y})]]$ — the average reward variance induced by $r_{\text{RM}}$ for the initial policy $\pi_{\theta(0)}$ and prompts in the training set $\mathcal{S}$. As a result, if $\text{Var}_{\mathbf{y} \sim \pi_{\theta(0)}(\cdot|\mathbf{x})}[r_{\text{RM}}(\mathbf{x}, \mathbf{y})]$ is low for prompts $\mathbf{x} \in \mathcal{S}$, which occurs when $r_{\text{RM}}$ does not sufficiently separate outputs that are probable under $\pi_{\theta(0)}$, then policy gradient suffers from slow optimization.

**Theorem 1** (Abridged version of Theorem 4). *Suppose that we maximize the RLHF objective (Equation (1)) with respect to a reward model $r_{\text{RM}}$, using a general autoregressive policy (Equation (2)). For any $\gamma > 0$, prompt $\mathbf{x} \in \mathcal{X}$, and reward function $r$ (e.g., $r$ can be the ground truth reward $r_{\text{G}}$ or the reward model $r_{\text{RM}}$), the time it takes until $\mathbb{E}_{\mathbf{y} \sim \pi_{\theta(t)}(\cdot|\mathbf{x})}[r(\mathbf{x}, \mathbf{y})] \geq \mathbb{E}_{\mathbf{y} \sim \pi_{\theta(0)}(\cdot|\mathbf{x})}[r(\mathbf{x}, \mathbf{y})] + \gamma$ is:*

$$\Omega\left(\mathbb{E}_{\mathbf{x}' \sim \mathcal{S}}\left[\text{Var}_{\mathbf{y} \sim \pi_{\theta(0)}(\cdot|\mathbf{x}')}[r_{\text{RM}}(\mathbf{x}', \mathbf{y})]\right]^{-\frac{1}{3}}\right),$$

*where $\Omega(\cdot)$ hides terms depending on $\gamma$, the KL regularization coefficient $\lambda$, the maximal output sequence length $L$, and the Jacobian of $f_\theta$ (Equation (2)).*

*Proof sketch (full proof of Theorem 4 is in Appendix D.4).* Low reward variance is known to imply a vanishing gradient for policies that produce next-token distributions via softmax (Theorem 1 in [64]). Specifically, $\|\nabla \mathbb{E}_{\mathbf{y} \sim \pi_\theta(\cdot|\mathbf{x})}[r_{\text{RM}}(\mathbf{x}, \mathbf{y})]\|$ decays with $\text{Var}_{\mathbf{y} \sim \pi_\theta(\cdot|\mathbf{x})}[r_{\text{RM}}(\mathbf{x}, \mathbf{y})]$ for any prompt $\mathbf{x}$ and policy $\pi_\theta$. However, this alone does not yield a satisfactory lower bound on the reward maximization rate because, without further knowledge, the gradient norm may increase rapidly during training. We show that this is not the case: when the reward variance is low, higher-order derivatives of the RLHF objective vanish alongside the gradient, preventing a rapid increase in gradient norm. This constrains the movement of policy parameters $\theta(t)$, leading to a lower bound on the rate of reward increase. $\square$

## 3.3    More Accurate Reward Models Are Not Necessarily Better Teachers

Section 3.2 showed that low reward variance hinders the efficiency of policy gradient. Notably, reward variance is independent of the measure typically used to evaluate reward models — accuracy [44, 93, 25, 47]. Accuracy depends only on how the reward model ranks different outputs, disregarding the degree of separation between their rewards, whereas reward variance is determined by this degree of separation. A key implication, established by Theorem 2, is that a reward model $r_{\text{RM}}$ can be accurate yet lead to arbitrarily slow ground truth reward maximization compared to a substantially less accurate model $r'_{\text{RM}}$, due to inducing low reward variance.

For clarity, Theorem 2 considers: *(i)* the extreme case where $r_{\mathrm{RM}}$ is perfectly accurate and $r'_{\mathrm{RM}}$ is almost completely inaccurate; and *(ii)* accuracy with respect to the uniform distribution over unordered output pairs, denoted by $\mathrm{acc}_{\mathbf{x}}(r_{\mathrm{RM}})$ for a prompt $\mathbf{x}$ and reward model $r_{\mathrm{RM}}$. The detailed version of the theorem (Theorem 5 in Appendix B) is much more general, applying to almost any accuracy values for $r_{\mathrm{RM}}$ and $r'_{\mathrm{RM}}$, as measured with respect to any distribution over output pairs.

**Theorem 2** (Abridged version of Theorem 5; proof sketch deferred to Appendix B.2)**.** *Suppose $\pi_\theta$ is a tabular policy (Equation (3)). Given a prompt $\mathbf{x} \in \mathcal{S}$, let $\gamma > 0$ and denote by $t_\gamma > 0$ the initial time at which $\mathbb{E}_{\mathbf{y} \sim \pi_{\theta(t)}(\cdot|\mathbf{x})}[r_{\mathrm{G}}(\mathbf{x}, \mathbf{y})] \geq \mathbb{E}_{\mathbf{y} \sim \pi_{\theta(0)}(\cdot|\mathbf{x})}[r_{\mathrm{G}}(\mathbf{x}, \mathbf{y})] + \gamma$. For any initial policy $\pi_{\theta(0)}$, there exist a perfectly accurate reward model $r_{\mathrm{RM}}$, i.e. $\mathrm{acc}_{\mathbf{x}}(r_{\mathrm{RM}}) = 1$, and an inaccurate reward model $r'_{\mathrm{RM}}$ with $\mathrm{acc}_{\mathbf{x}}(r'_{\mathrm{RM}}) \leq 2/|\mathcal{Y}|$ such that the following hold.*

- ***Slow ground truth reward increase under*** $r_{\mathrm{RM}}$***.*** *When using $r_{\mathrm{RM}}$ for maximizing the RLHF objective, $t_\gamma$ can be arbitrarily large.*

- ***Fast ground truth reward increase under*** $r'_{\mathrm{RM}}$***.*** *In contrast, $t_\gamma = \mathcal{O}\big(\pi_{\theta(0)}(\mathbf{y}^\gamma|\mathbf{x})^{-1}\big)$ when using $r'_{\mathrm{RM}}$, where $\mathbf{y}^\gamma \in \mathcal{Y}$ is an output satisfying $r_{\mathrm{G}}(\mathbf{x}, \mathbf{y}^\gamma) > \mathbb{E}_{\mathbf{y} \sim \pi_{\theta(0)}(\cdot|\mathbf{x})}[r_{\mathrm{G}}(\mathbf{x}, \mathbf{y})] + \gamma$ and $\mathcal{O}(\cdot)$ hides terms depending on $\gamma$.*

We emphasize that Theorem 2 does not suggest highly accurate reward models are inherently poor teachers. In fact, as established by its detailed version in Appendix B (Theorem 5), at almost any accuracy level some reward models lead to inefficient optimization while others perform well. Rather, Theorem 2 formalizes why accuracy on its own is an insufficient criterion for evaluating reward models in RLHF, agreeing with recent empirical evidence [37, 10, 80].

Accuracy is a desirable property nonetheless, since more accurate models are usually less susceptible to *reward hacking* [5, 70, 57, 26]. Namely, when training with an imperfect reward model, eventually the ground truth reward can start decreasing due to a mismatch between the two rewards. It is therefore common practice to run policy gradient for only a few epochs (see, *e.g.*, [55, 37, 2]). Theorem 2 captures this regime, in which a less accurate reward model can outperform a more accurate one by driving a faster increase in ground truth reward.

**Role of accuracy depends on the alignment method.** While accuracy alone does not guarantee the effectiveness of RLHF, its importance varies across alignment methods. For example, in Best-of-N sampling [52], it is straightforward to prove that a perfectly accurate reward model is always optimal (see Proposition 1 in Appendix C). Investigating the role of accuracy in additional methods [17, 32, 27, 18, 65] remains an important direction for future work.

### 3.4 For Different Initial Policies, Different Reward Models Are Better

Reward variance depends on both the reward model and the policy. In particular, a reward model inducing high reward variance for one policy may induce low reward variance for another. Consequently, the connection between reward variance and optimization implies that for different initial policies, different reward models are better — see Theorem 3. This highlights that faithfully evaluating reward models for RLHF requires taking into account the policy being aligned.

**Theorem 3** (Abridged version of Theorem 6; proof sketch deferred to Appendix B.3)**.** *Suppose $\pi_\theta$ is a tabular policy (Equation (3)). Given a prompt $\mathbf{x} \in \mathcal{S}$, let $\gamma > 0$ and denote by $t_\gamma > 0$ the initial time at which $\mathbb{E}_{\mathbf{y} \sim \pi_{\theta(t)}(\cdot|\mathbf{x})}[r_{\mathrm{G}}(\mathbf{x}, \mathbf{y})] \geq \mathbb{E}_{\mathbf{y} \sim \pi_{\theta(0)}(\cdot|\mathbf{x})}[r_{\mathrm{G}}(\mathbf{x}, \mathbf{y})] + \gamma$. There exist reward models $r_{\mathrm{RM}}, r'_{\mathrm{RM}}$ and initial policy families $\Pi, \Pi'$ such the following hold.*

- *$r_{\mathrm{RM}}$ **is a better teacher for initial policies in** $\Pi$. If $\pi_{\theta(0)} \in \Pi$, then $t_\gamma = \mathcal{O}(1)$ when using $r_{\mathrm{RM}}$ for maximizing the RLHF objective, whereas $t_\gamma$ can be arbitrarily large when using $r'_{\mathrm{RM}}$.*

- *$r'_{\mathrm{RM}}$ **is a better teacher for initial policies in** $\Pi'$. If $\pi_{\theta(0)} \in \Pi'$, then $t_\gamma = \mathcal{O}(1)$ when using $r'_{\mathrm{RM}}$ for maximizing the RLHF objective, whereas $t_\gamma$ can be arbitrarily large when using $r_{\mathrm{RM}}$.*

*In the above, $\mathcal{O}(\cdot)$ hides terms depending on $\gamma$.*

# 4 Experiments

We verify that conclusions drawn from our theoretical analysis (Section 3) hold in practice. Consistent with Sections 3.2 and 3.3, Section 4.1 shows that reward variance strongly correlates with the reward maximization rate during policy gradient. In particular, using more accurate reward models may result in worse performance within a given training budget if they induce low reward variance. Strikingly, this applies also to the ground truth reward: even if it is accessible, in some regimes a proxy reward model can be more effective. Then, in accordance with Section 3.4, Section 4.2 demonstrates that for different language models, different reward models lead to a higher ground truth reward.

Our experiments include language models from different families (Pythia [8] and Llama-3.2 [21]), reward models of up to 8B scale, and standard RLHF datasets (AlpacaFarm [22] and UltraFeedback [14]). For brevity, we defer to Appendices E and F some experiments and implementation details, respectively. Code for reproducing our results is available at `https://github.com/princeton-pli/what-makes-good-rm`.

## 4.1 More Accurate Reward Models Are Not Necessarily Better Teachers

### 4.1.1 Setting

We follow the RLHF pipeline outlined in Section 2.1.

**Ground truth reward.** Evaluating policies based on human preferences is expensive. Thus, as done in prior work [26, 13, 73, 7, 10, 80, 72], we simulate human preferences with a high-quality reward model. In our experiments, the ArmoRM [76] model serves as the ground truth reward.

**Data.** We partition the UltraFeedback [14] training set into two subsets: 80% of the samples are used for reward model training and the rest for the policy gradient step of RLHF. Output preferences in the reward modeling subset are relabeled using the ground truth reward.

**Initial policy.** We SFT the pretrained Pythia-2.8B language model on AlpacaFarm. Appendix E includes experiments where SFT is performed on the (UltraFeedback-based) reward model training set.

**Reward models.** Studying how reward variance and accuracy affect the outcome of RLHF requires reward models that vary in these properties. To that end, we use different output pair distributions for training. Specifically, we create five different training sets by associating either 100%, 75%, 50%, 25%, or 0% of the prompts with *on-policy* preference pairs, sampled from the initial policy and labeled according to the ground truth reward, while the remaining prompts are associated with their *off-policy* preferences from (the relabeled) UltraFeedback. We then train a separate reward model over each training set via the usual approach. Namely, each reward model is initialized based on the SFT model and trained with a Bradley-Terry log-likelihood loss (*cf.* [9, 55]). We additionally consider a perfectly accurate reward model that induces low reward variance, obtained by taking the ground truth reward and reducing how well it separates different outputs while preserving their rankings.

**Reward normalization.** For fair comparison, we normalize all reward models so that they produce rewards on the same scale (similarly to [26]; see Appendix F for further details). Table 1 reports, for each model, the reward variance it induces for the initial policy and its accuracy, both on-policy (*i.e.*, on outputs sampled from the initial policy) and off-policy (*i.e.*, on outputs from UltraFeedback).

**Policy gradient.** We use RLOO [42, 2] as it is substantially more resource efficient than PPO [68] and has shown competitive results [2]. Policies are evaluated by computing the average proxy and ground truth rewards over prompts from the policy gradient training set, where for each prompt 10 outputs are sampled from the policy. We also evaluated policies using prompts from the (UltraFeedback) test set. The results were nearly identical, so we report only rewards over the training set for conciseness.[5]

### 4.1.2 Results

For each reward model described in Section 4.1.1, Figure 2 presents the increase in proxy reward (used for training) and ground truth reward during policy gradient. As a baseline, we compare the reward models to running policy gradient directly with the ground truth reward.

---

[5]Rewards over the training and test sets were similar since, in accordance with common practice [55, 37, 2], policy gradient was carried out for only a few epochs (six in the experiments of Section 4.1).

Table 1: For each reward model described in Section 4.1.1 and the ground truth reward, we report: *(i)* the reward variance induced for the Pythia-2.8B initial policy; and *(ii)* accuracy, measured on-policy (*i.e.*, on outputs sampled from the initial policy) and off-policy (*i.e.*, on outputs from UltraFeedback). All quantities were averaged across prompts in the policy gradient training set (their values on the test set were nearly identical). As mentioned in Section 4.1.1, to ensure fair comparison of reward variance, the reward models and ground truth reward were normalized so that they produce rewards on the same scale.

| | RM On-Policy % | | | | | RM with Perfect Acc. but Low Reward Variance | Ground Truth Reward |
| | 100% | 75% | 50% | 25% | 0% | | |
|---|---|---|---|---|---|---|---|
| Reward Variance | 0.630 | 0.616 | 0.555 | 0.438 | 0.314 | 0.111 | 0.256 |
| On-Policy Acc. | 0.660 | 0.656 | 0.640 | 0.616 | 0.587 | 1.000 | 1.000 |
| Off-Policy Acc. | 0.630 | 0.732 | 0.758 | 0.754 | 0.762 | 1.000 | 1.000 |

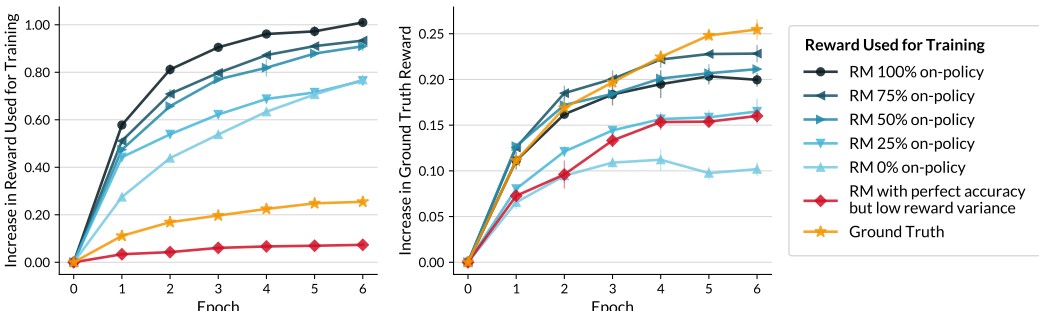

Figure 2: **More accurate reward models are not necessarily better teachers.** Using each reward model described in Section 4.1.1, we trained a Pythia-2.8B language model via policy gradient (specifically, RLOO [2]) on prompts from UltraFeedback; characteristics of these reward models are provided in Table 1. For comparison, we also ran policy gradient directly with the ground truth reward. Plotted is the increase in proxy reward (left), *i.e.*, the reward used for training, and ground truth reward (right) against the number of epochs. Markers indicate the mean across three runs, with error bars showing standard deviation. In agreement with Theorem 2, a perfectly accurate reward model that induces low reward variance (red markers) underperforms less accurate models. Moreover, in the first couple of epochs, a proxy reward model worked better than directly optimizing the ground truth reward; see Figure 7 for an experiment where this gap is even larger.

**Reward variance is indicative of reward maximization rate.** In line with Theorem 1, the reward variance induced for the initial policy strongly correlates with the increase in both proxy and ground truth rewards; see Table 2 for the correlation coefficients. For the ground truth reward, most indicative is a combination of reward variance and accuracy. This is expected, since when the reward model is inaccurate, improvements in proxy reward translate less to improvements in ground truth reward.

**Perfectly accurate reward models are not necessarily the best teachers.** Unlike reward variance, in our setting accuracy by itself is not indicative of reward increase (Table 2).[6] In line with Theorem 2, the perfectly accurate reward model with low reward variance leads to a significantly slower increase in ground truth reward compared to less accurate models (Figure 2). Moreover, using a proxy reward model can be more effective than training with the ground truth reward directly, at least in the first few epochs (Figures 2 and 7). However, if policy gradient is run for long enough with a proxy reward model, the ground truth reward saturates sooner and may decrease due to reward hacking. We further find that accurate reward models are typically more KL efficient. At a given ground truth reward value, they yield policies with lower KL divergence from the initial policy (Figure 4 in Appendix E).

**On-policy vs off-policy accuracy.** As discussed in Section 2.2, measuring accuracy on off-policy outputs can be problematic when they differ substantially from on-policy outputs. This is evident in Figure 2 and Table 1, as well as the additional experiments mentioned below: reward models with high accuracy on off-policy outputs from UltraFeedback but low on-policy accuracy are poor teachers and tend to cause reward hacking.

---

[6]Although accuracy is often negatively correlated with reward increase in our setting, which includes a perfectly accurate reward model that induces low reward variance, this is not always the case. The relationship between accuracy and reward increase can vary depending on the reward models considered (*cf.* [80, 25]).

Table 2: **Reward variance strongly correlates with reward increase, while accuracy on its own may not.**
For the experiments in Figure 2, we report the Pearson and Spearman correlations of different reward model properties (see Table 1) with reward increase after one epoch of policy gradient. "On- & Off-Policy Acc." refers to accuracy measured both on output pairs sampled from the initial policy and output pairs from UltraFeedback. "Reward Variance & Acc." averages this accuracy with reward variance (induced for the initial policy). Notably, the latter combined measure is more indicative of ground truth reward increase than each measure separately.

| | Increase in Reward Used for Training | | Increase in Ground Truth Reward | |
|---|---|---|---|---|
| | Pearson Corr. | Spearman Corr. | Pearson Corr. | Spearman Corr. |
| Reward Variance | **0.982** | **1.000** | 0.834 | 0.714 |
| On-Policy Acc. | −0.762 | 0.143 | −0.283 | 0.200 |
| Off-Policy Acc. | −0.933 | −0.943 | −0.507 | −0.543 |
| On- & Off-Policy Acc. | −0.870 | −0.486 | −0.397 | 0.200 |
| Reward Variance & Acc. | 0.676 | 0.886 | **0.940** | **0.828** |

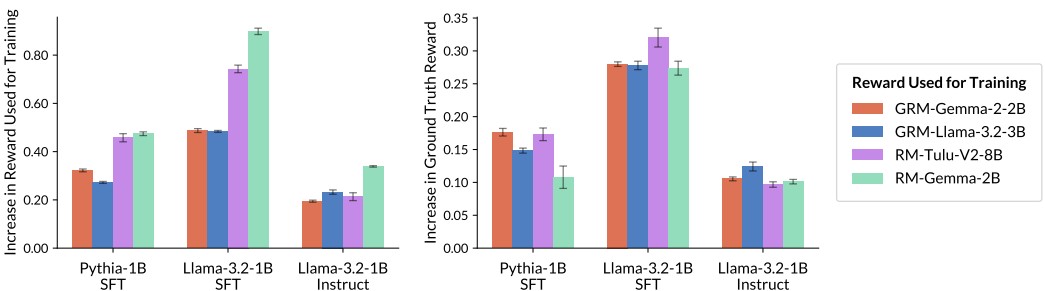

Figure 3: **For different language models, different reward models are better.** Using publicly available reward models, we trained different language models via policy gradient (specifically, RLOO [2]) on prompts from UltraFeedback; see Table 10 for characteristics of the reward models. Plotted is the increase in proxy reward (left), *i.e.*, the reward used for training, and ground truth reward (right) for each combination of initial policy and reward model, averaged across three runs (error bars denote standard deviation). In agreement with Theorem 3, the reward model that yields the highest ground truth reward varies across initial policies.

**Additional experiments.** Appendix E.1 includes analogous findings for experiments with: *(i)* a Pythia-1B language model; *(ii)* SFT carried out over the UltraFeedback reward model training set (as opposed to on AlpacaFarm); and *(iii)* a different ground truth reward model.

## 4.2   For Different Language Models, Different Reward Models Are Better

### 4.2.1   Setting

We follow the setup outlined in Section 4.1.1, but instead of using a single initial policy and training reward models ourselves, we consider multiple initial policies and publicly available reward models. The initial policies are based on the Pythia [8] and Llama-3.2 [21] language model families, with SFT performed over UltraFeedback. Additionally, we consider instruction-tuned Llama-3.2 language models (which we do not SFT). To ensure diversity in reward model qualities, we select four models varying in their RewardBench [44] scores: GRM-Llama-3.2-3B [87] (90.9), GRM-Gemma-2-2B [87] (88.4), RM-Tulu-V2-8B [37] (74.5), and RM-Gemma-2B (65.4). As in Section 4.1, for fair comparison, we normalize reward models so that they produce rewards on the same scale.

### 4.2.2   Results

Figure 3 corroborates Theorem 3 by showing that for different initial policies, different reward models are better in terms of ground truth reward increase. Furthermore, Table 10 in Appendix E displays the reward variance and accuracy (both on-policy and off-policy) for each combination of reward model and initial policy. The results demonstrate that low accuracy and low reward variance can each be detrimental to the success of RLHF. Specifically, RM-Gemma-2B induces the highest reward variance for all initial policies, and accordingly led to the largest increase in proxy reward. However, since it is inaccurate, this did not translate into a higher ground truth reward. On the other hand, as

was the case in the experiments of Section 4.1, the most accurate reward model (whether on-policy or off-policy) was not necessarily the most effective either.

**Additional experiments.** Appendix E.2 includes experiments using: *(i)* additional language models of up to 3B scale; and *(ii)* GRPO [69] instead of RLOO.

# 5 Conclusion

Although RLHF heavily relies on the reward model's quality, there is limited understanding as to how this quality should be measured (*cf.* [37, 10, 80]). In this work, we studied what makes a good reward model from an optimization perspective. Our theoretical analysis showed that, regardless of how accurate a reward model is, if it induces low *reward variance*, then RLHF suffers from slow reward maximization due to a flat objective landscape. This implies that alongside being accurate, reward models need to induce sufficient variance for enabling efficient optimization.

We then proved and empirically demonstrated two key implications of the connection between reward variance and optimization rate: *(i)* more accurate reward models are not necessarily better teachers; and *(ii)* a reward model that works well for one language model can perform poorly for another. These results formalize shortcomings of existing reward model benchmarks, which focus primarily on accuracy and produce universal rankings independent of any particular language model [44, 93, 25, 47]. As elaborated below, we hope insights from this work will inform improved methodologies for reward model training and evaluation that account for properties beyond accuracy.

## 5.1 Limitations and Future Work

**Improving reward variance.** We established that low reward variance undermines the effectiveness of RLHF (*i.e.*, of policy gradient optimization). Yet, how to best modify or train reward models to induce higher reward variance remains an open question. A naive solution is to multiply all rewards by a factor of $c > 1$, thereby scaling up reward variance by a factor of $c^2$, but such an artificial inflation can be problematic. The expected reward gradient vanishes when the reward variance is low (*cf.* [64]). Thus, sample-based estimates of the gradient may be dominated by noise, which will be amplified by reward scaling. In preliminary experiments, we indeed found reward scaling did not alter trends observed in Section 4.1. However, this does not preclude more sophisticated reward scaling techniques from being helpful, especially when coupled with improved gradient estimates (*e.g.*, by sampling more outputs for each prompt). An alternative approach, given that reward variance depends on the separation between rewards (see intuition following Definition 2), can be to encourage a larger margin during reward model training (*cf.* [74, 60, 51]).

**Reward model evaluation.** Our results emphasize the need for holistic evaluations that account for: *(i)* properties beyond accuracy, reinforcing empirical observations in [10, 80]; and *(ii)* the specific language model being aligned. We identified reward variance, arising from the interaction between the reward model and language model, as key for efficient policy gradient optimization. Characterizing additional factors influencing the outcome of RLHF and providing robust protocols for reward model evaluation is a valuable direction for future work.

**Verifiable rewards and general reinforcement learning environments.** Although our positioning focused on RLHF — training language models via reward models learned from human preferences — the analysis in Section 3 is agnostic to the source of rewards and can be straightforwardly extended to any reinforcement learning environment with policies that produce a distribution over outputs (*i.e.*, actions) via the softmax function. In particular, policy gradient methods are increasingly used for solving tasks in which outputs can be automatically verified as correct, such as math and coding [43, 33]. Investigating whether our insights can aid in designing data selection algorithms or verifiable rewards that improve optimization efficiency is a promising avenue for further research.

**Alignment methods beyond RLHF.** As shown in Section 3.3, the effectiveness of a reward model can vary depending on the alignment method. Namely, while in RLHF perfectly accurate reward models (*i.e.*, reward models that rank all outputs correctly) can underperform less accurate models, for Best-of-N [52] they are always optimal. An exciting next step is to study how properties of different alignment methods, including Best-of-N and those based on contrastive objectives [61, 27, 18, 65] or rejection sampling [17, 32], determine which aspects of a reward model are important.

## Acknowledgments and Disclosure of Funding

We thank Eshbal Hezroni for aid in preparing illustrative figures and Sadhika Malladi and Boris Hanin for providing feedback on this paper. NR is supported by Princeton Language and Intelligence (PLI) and the Zuckerman STEM Leadership Program. JDL acknowledges support of Open Philanthropy, NSF IIS 2107304, NSF CCF 2212262, NSF CAREER Award 2144994, and NSF CCF 2019844. SA acknowledges funding from ONR, Schmidt Science, and OpenAI.

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

# A    Related Work

**Theoretical analyses of RLHF.** To the best of our knowledge, this work is the first to characterize how properties of the reward model affect policy gradient optimization in RLHF. Existing analyses primarily considered specialized algorithms that deviate from the standard RLHF pipeline (*cf.* Section 2.1) and provided, under varying technical assumptions, guarantees on finding an optimal policy [85, 54, 56, 94, 83, 78, 89, 39, 20, 15, 38, 67, 29, 90, 82, 36, 4]. Furthermore, [46, 72] bounded the sample complexity of estimating the ground truth reward based on preference data.

**Transforming and ensembling reward models.** The question of what makes a good reward model also has roots in the reward shaping literature [53]. There, prior work characterized which reward transformations conserve optimal policies [53, 31] or analyzed the convergence of value-based algorithms [19, 34]. More recently, in the context of RLHF, methods for transforming [7, 79] and ensembling [13, 62, 23, 91] reward models were proposed. Though, how these modifications to the reward model influence policy gradient optimization was not accounted for. We note that, while reward ensembling was largely suggested for mitigating reward hacking [5, 70], our optimization viewpoint sheds light on an additional benefit it may bring. Namely, incorporating multiple reward models can reduce the chance of a flat RLHF objective landscape due to low reward variance.

**Optimization difficulties in policy gradient.** Since the objective that policy gradient maximizes is non-concave even for simple policy parameterizations (*e.g.*, tabular policies; Equation (3)), establishing guarantees on optimization rate has proven challenging [24, 48, 49, 1, 45, 50, 64, 63]. In particular, for policies that produce distributions over outputs via softmax (*e.g.*, language models), the gradient is known to vanish when the policy is near-deterministic [49, 48, 1]. This issue was empirically demonstrated to cause slow reward maximization in several reinforcement learning environments [3, 66, 35, 49, 48, 28]. However, its relevance to RLHF is limited — language models rarely produce near-deterministic distributions. More generally, [64] showed that the gradient vanishes whenever the reward variance is low. Our analysis makes use of this result for studying what makes a good reward model for RLHF (see proof sketch of Theorem 1 for technical details).

It is worth noting that, in theory, the poor optimization landscape of policy gradient can be solved via natural policy gradient [40, 77, 1, 88] — a preconditioning method — or entropy regularization [49, 16]. However, these approaches are currently impractical in the context of large langauge models. Specifically, natural policy gradient requires inverting a $P \times P$ matrix, where $P$ is the number of language model parameters, while entropy regularization often does not yield noticeable improvements for language generation due to the large output space [64].

**Stronger teachers do not always provide better supervision.** In the context of knowledge distillation, [58, 86] showed that stronger language models do not necessarily generate better labels for training a student language model. This phenomenon resembles, yet is distinct from, the finding that more accurate reward models are not necessarily better teachers for RLHF (Section 3.3). Notably, in the setting of [58], strong models inherently provide weak supervision. By contrast, in RLHF, high accuracy does not imply that a reward model is a poor teacher. Rather, factors beyond accuracy can determine its effectiveness (see discussion following Theorem 2).

# B    Detailed Statements of Theoretical Results

In this appendix, we deliver the full statements of our theoretical results. Appendices B.1 to B.3 cover the theorems discussed in Sections 3.2 to 3.4, respectively. We refer the reader to Section 3.1 for the technical setting of the analysis.

## B.1    Low Reward Variance Implies Slow Reward Maximization

Theorem 4 is the detailed version of Theorem 1 (Section 3.2). It lower bounds the time required for the expected reward, measured with respect to any reward function, to increase by an additive constant. As shown below, this time grows inversely with $\mathbb{E}_{\mathbf{x} \sim \mathcal{S}}[\mathrm{Var}_{\mathbf{y} \sim \pi_{\theta(0)}(\cdot|\mathbf{x})}[r_{\mathrm{RM}}(\mathbf{x}, \mathbf{y})]]$, regardless of whether KL regularization is applied or not. The proof of Theorem 4 is deferred to Appendix D.4.

**Theorem 4.** *Suppose gradient flow, over a general autoregressive policy (Equation (2)), is used to maximize the RLHF objective with respect to a reward model $r_{\mathrm{RM}} : \mathcal{X} \times \mathcal{Y} \to [-1, 1]$ over a set of prompts $\mathcal{S}$ (Equations (1) and (4)) . For any $\gamma > 0$, prompt $\mathbf{x} \in \mathcal{X}$, and reward function*

$r : \mathcal{X} \times \mathcal{Y} \to [-1, 1]$ *(e.g., $r$ can be the ground truth reward $r_{\mathrm{G}}$ or the reward model $r_{\mathrm{RM}}$), denote by $t_\gamma$ the initial time at which $\mathbb{E}_{\mathbf{y} \sim \pi_{\theta(t)}(\cdot|\mathbf{x})}[r(\mathbf{x}, \mathbf{y})] \geq \mathbb{E}_{\mathbf{y} \sim \pi_{\theta(0)}(\cdot|\mathbf{x})}[r(\mathbf{x}, \mathbf{y})] + \gamma$ (by convention, $t_\gamma = \infty$ if no such time exists). Then, for any initial policy $\pi_{\theta(0)}$:*

$$t_\gamma \geq \frac{\min\{1 - \exp(-\frac{\gamma}{9}), 1 - \exp(-\frac{7}{3\lambda})\}}{8L^2 J_{t_\gamma}^2} \cdot \frac{1}{\mathbb{E}_{\mathbf{x}' \sim \mathcal{S}}\left[\mathrm{Var}_{\mathbf{y} \sim \pi_{\theta(0)}(\cdot|\mathbf{x}')}[r_{\mathrm{RM}}(\mathbf{x}', \mathbf{y})]\right]^{\frac{1}{3}}} \cdot$$

*In the above, $J_{t_\gamma} := \sup_{\mathbf{x}' \in \mathcal{X}, \mathbf{y} \in \mathcal{Y}, l \in [L], t \in [0, t_\gamma]} \left\| \mathbf{J}_{f_{\theta(t)}(\mathbf{x}', \mathbf{y}_{<l})} \right\|_2$, where $\mathbf{J}_{f_\theta(\mathbf{x}', \mathbf{y}_{<l})}$ denotes the Jacobian of $f_\theta(\mathbf{x}', \mathbf{y}_{<l})$ with respect to $\theta$ and $\|\cdot\|_2$ denotes the spectral norm, $L$ is the maximal output sequence length, and by convention $\exp(-\frac{7}{3\lambda}) = 0$ if $\lambda = 0$.*

**Remark 1.** For some policy parameterizations (*e.g.*, tabular policies), the Jacobian of $f_\theta$ with respect to $\theta$ remains constant throughout optimization. By contrast, for more complex neural network parameterizations, it can vary. In theory, there may exist settings in which the Jacobian norm grows substantially, thereby alleviating the harm due to low reward variance. However, our experiments (Section 4) demonstrate that reward variance strongly correlates with reward increase. This indicates that changes in the Jacobian do not compensate for low reward variance in practical settings.

## B.2 More Accurate Reward Models Are Not Necessarily Better Teachers

Theorem 5 is a detailed and more general version of Theorem 2 (Section 3.3). For any distribution $\mathcal{D}$ over unordered output pairs and almost any accuracy values $\alpha, \alpha' \in [0, 1]$, Theorem 5 shows that there exist reward models $r_{\mathrm{RM}}, r'_{\mathrm{RM}}$ attaining these accuracy values such that $r_{\mathrm{RM}}$ leads to a substantially slower ground truth reward increase compared to $r'_{\mathrm{RM}}$, where the gap in optimization rates stems from $r_{\mathrm{RM}}$ inducing lower reward variance for the initial policy. Since this holds even when $\alpha > \alpha'$, Theorem 5 implies that more accurate reward models are not necessarily better teachers. The proof of Theorem 5 is deferred to Appendix D.8.

Note that not all values between zero and one correspond to a valid accuracy. For example, if the ground truth reward $r_{\mathrm{G}}$ assigns the same reward to three different outputs, a reward model cannot misorder only one pair among them — it must either rank them all correctly or make at least two mistakes. We therefore use the following notion of *attainable accuracy* in Theorem 5.

**Definition 3.** For any prompt $\mathbf{x} \in \mathcal{X}$, distribution $\mathcal{D}$ over unordered output pairs, and ground truth reward $r_{\mathrm{G}} : \mathcal{X} \times \mathcal{Y} \to [-1, 1]$, we say that $\alpha \in [0, 1]$ is an *attainable accuracy* if there exists a reward model $r_{\mathrm{RM}} : \mathcal{X} \times \mathcal{Y} \to [-1, 1]$ such that $\mathrm{acc}_{\mathbf{x}, \mathcal{D}}(r_{\mathrm{RM}}) = \alpha$ (recall $\mathrm{acc}_{\mathbf{x}, \mathcal{D}}(r_{\mathrm{RM}})$ denotes the accuracy of $r_{\mathrm{RM}}$; see Definition 1). Furthermore, for an output $\mathbf{y} \in \mathcal{Y}$, we say that $\alpha$ is an *attainable accuracy subject to $\mathbf{y}$ being ranked first* if there exists $r_{\mathrm{RM}} : \mathcal{X} \times \mathcal{Y} \to [-1, 1]$ such that $\mathrm{acc}_{\mathbf{x}, \mathcal{D}}(r_{\mathrm{RM}}) = \alpha$ and $\mathbf{y} \in \mathrm{argmax}_{\mathbf{y}' \in \mathcal{Y}} r_{\mathrm{RM}}(\mathbf{x}, \mathbf{y}')$.

**Theorem 5.** *Suppose gradient flow, over a tabular policy (Equation (3)), is used to maximize the RLHF objective over a set of prompts $\mathcal{S}$ (Equations (1) and (4)). Given a prompt $\mathbf{x} \in \mathcal{S}$, let $\gamma > 0$ be a desired expected ground truth reward increase and $\mathbf{y}^\gamma \in \mathcal{Y}$ be an output satisfying $r_{\mathrm{G}}(\mathbf{x}, \mathbf{y}^\gamma) > \mathbb{E}_{\mathbf{y} \sim \pi_{\theta(0)}(\cdot|\mathbf{x})}[r_{\mathrm{G}}(\mathbf{x}, \mathbf{y})] + \gamma$. Denote by $t_\gamma$ the initial time at which $\mathbb{E}_{\mathbf{y} \sim \pi_{\theta(t)}(\cdot|\mathbf{x})}[r_{\mathrm{G}}(\mathbf{x}, \mathbf{y})] \geq \mathbb{E}_{\mathbf{y} \sim \pi_{\theta(0)}(\cdot|\mathbf{x})}[r_{\mathrm{G}}(\mathbf{x}, \mathbf{y})] + \gamma$ (by convention, $t_\gamma = \infty$ if no such time exists). Then, for any $T \geq 1$, initial policy $\pi_{\theta(0)}$, distribution $\mathcal{D}$ over unordered output pairs, attainable accuracy $\alpha \in [0, 1]$, and attainable accuracy subject to $\mathbf{y}^\gamma$ being ranked first $\alpha' \in [0, 1]$, there exist reward models $r_{\mathrm{RM}}, r'_{\mathrm{RM}} : \mathcal{X} \times \mathcal{Y} \to [-1, 1]$ with $\mathrm{acc}_{\mathbf{x}, \mathcal{D}}(r_{\mathrm{RM}}) = \alpha$ and $\mathrm{acc}_{\mathbf{x}, \mathcal{D}}(r'_{\mathrm{RM}}) = \alpha'$ such that the following hold.*

- ***Slow ground truth reward increase under $r_{\mathrm{RM}}$.*** *When using $r_{\mathrm{RM}}$ for maximizing the RLHF objective:*

$$t_\gamma \geq 2|\mathcal{S}|\left(1 - \exp\left(-\frac{\gamma}{2}\right)\right) \cdot T = \Omega(T).$$

- ***Fast ground truth reward increase under $r'_{\mathrm{RM}}$.*** *In contrast, if the KL regularization coefficient $\lambda \geq 0$ is not too large, namely $\lambda \leq \frac{\pi_{\theta(0)}(\mathbf{y}^\gamma|\mathbf{x})(1-\rho)^4}{8\left(\frac{1}{2e} - 2 \ln \min_{\mathbf{y} \in \mathcal{Y}} \pi_{\theta(0)}(\mathbf{y}|\mathbf{x})\right)}$, when using $r'_{\mathrm{RM}}$ for maximizing the RLHF objective:*

$$t_\gamma \leq \frac{4|\mathcal{S}|}{(1-\rho)^4} \cdot \left(\frac{1}{\pi_{\theta(0)}(\mathbf{y}^\gamma|\mathbf{x})} - \frac{1}{\rho}\right) = \mathcal{O}\left(\frac{1}{\pi_{\theta(0)}(\mathbf{y}^\gamma|\mathbf{x})}\right),$$

*where $\rho := \frac{\mathbb{E}_{\mathbf{y} \sim \pi_{\theta(0)}(\cdot|\mathbf{x})}[r_G(\mathbf{x},\mathbf{y})]+\gamma+1}{r_G(\mathbf{x},\mathbf{y}^\gamma)+1} \in (0,1)$.*

*Proof sketch (full proof of Theorem 5 is in Appendix D.8).* Regardless of a reward model's accuracy, it can induce an arbitrarily low reward variance for $\pi_{\theta(0)}$. For example, $r_{\mathrm{RM}}$ can assign roughly the same reward to all outputs that have a non-negligible probability under $\pi_{\theta(0)}$ while ranking outputs according to any given order. Thus, the lower bound on $t_\gamma$ when using a perfectly accurate $r_{\mathrm{RM}}$ readily follows by Theorem 1.[7] On the other hand, since the RLHF objective is non-concave, upper bounding $t_\gamma$ requires a fine-grained analysis of how the policy evolves during training. We show that, if $r'_{\mathrm{RM}}(\mathbf{x}, \mathbf{y}^\gamma)$ is sufficiently higher than $r'_{\mathrm{RM}}(\mathbf{x}, \mathbf{y})$ for all $\mathbf{y} \neq \mathbf{y}^\gamma$, then $\pi_{\theta(t)}(\mathbf{y}^\gamma|\mathbf{x})$ grows rapidly. This holds even when $r'_{\mathrm{RM}}$ is almost completely inaccurate and implies that $t_\gamma = \mathcal{O}(\pi_{\theta(0)}(\mathbf{y}^\gamma|\mathbf{x})^{-1})$. Note that the dependence on $\pi_{\theta(0)}(\mathbf{y}^\gamma|\mathbf{x})$ falls in line with intuition, by which the initial policy needs to already be somewhat competent for policy gradient to be effective [92, 64, 12]. □

**Remark 2.** Theorem 2 in Section 3.3 focuses on a special case of Theorem 5. Specifically, Theorem 2 considers the case where $\mathcal{D}$ is the uniform distribution over unordered output pairs, $\alpha = 1$, and $\alpha'$ corresponds to the accuracy of a reward model that orders incorrectly all output pairs, except those that include $\mathbf{y}^\gamma$ and an output $\mathbf{y} \in \mathcal{Y}$ that satisfies $r_G(\mathbf{x}, \mathbf{y}) < r_G(\mathbf{x}, \mathbf{y}^\gamma)$. Notice that $\alpha' \leq 2/|\mathcal{Y}|$ since the number of unordered output pairs containing $\mathbf{y}^\gamma$ is $|\mathcal{Y}| - 1$ and the total number of unordered output pairs is $|\mathcal{Y}|(|\mathcal{Y}| - 1)/2$.

## B.3 For Different Initial Policies, Different Reward Models Are Better

Theorem 6 is the detailed version of Theorem 3 (Section 3.4). It establishes that for different initial policies, different reward models can lead to a faster increase in ground truth reward. Namely, there exist reward models $r_{\mathrm{RM}}, r'_{\mathrm{RM}}$ and initial policy families $\Pi, \Pi'$ such that $r_{\mathrm{RM}}$ is a better teacher for initial policies in $\Pi$, while $r'_{\mathrm{RM}}$ is a better teacher for initial policies in $\Pi'$. The proof of Theorem 6 is deferred to Appendix D.9.

**Theorem 6.** *Suppose gradient flow, over a tabular policy (Equation (3)), is used to maximize the RLHF objective over a set of prompts $\mathcal{S}$ (Equations (1) and (4)). Given a prompt $\mathbf{x} \in \mathcal{S}$, let $\gamma > 0$ be a desired expected ground truth reward increase for which there exist outputs $\mathbf{y}^\gamma, \mathbf{y}'^\gamma \in \mathcal{Y}$ satisfying $r_G(\mathbf{x}, \mathbf{y}^\gamma) > V_0 + \gamma$ and $r_G(\mathbf{x}, \mathbf{y}'^\gamma) > V_0 + \gamma$, where $V_0 \in (-1, 1)$. Denote by $t_\gamma$ the initial time at which $\mathbb{E}_{\mathbf{y} \sim \pi_{\theta(t)}(\cdot|\mathbf{x})}[r_G(\mathbf{x}, \mathbf{y})] \geq \mathbb{E}_{\mathbf{y} \sim \pi_{\theta(0)}(\cdot|\mathbf{x})}[r_G(\mathbf{x}, \mathbf{y})] + \gamma$ (by convention, $t_\gamma = \infty$ if no such time exists). Then, for any $T \geq 1$, constant $c \in (0, 1)$, and the initial policy families $\Pi, \Pi'$ defined by:*

$$\Pi := \left\{ \pi_\theta : \mathbb{E}_{\mathbf{y} \sim \pi_\theta(\cdot|\mathbf{x})}[r_G(\mathbf{x}, \mathbf{y})] = V_0 \,,\, \pi_\theta(\mathbf{y}'^\gamma|\mathbf{x}) \leq T^{-2} \,,\, \pi_\theta(\mathbf{y}^\gamma|\mathbf{x}) \geq c \right\},$$

$$\Pi' := \left\{ \pi_\theta : \mathbb{E}_{\mathbf{y} \sim \pi_\theta(\cdot|\mathbf{x})}[r_G(\mathbf{x}, \mathbf{y})] = V_0 \,,\, \pi_\theta(\mathbf{y}^\gamma|\mathbf{x}) \leq T^{-2} \,,\, \pi_\theta(\mathbf{y}'^\gamma|\mathbf{x}) \geq c \right\},$$

*there exist reward models $r_{\mathrm{RM}}, r'_{\mathrm{RM}} : \mathcal{X} \times \mathcal{Y} \to [-1, 1]$ such that the following hold.*

- $r_{\mathrm{RM}}$ *is a better teacher for initial policies in $\Pi$.* *If $\pi_{\theta(0)} \in \Pi$, when using $r_{\mathrm{RM}}$ for maximizing the RLHF objective with $\lambda \leq \frac{c(1-\rho)^3}{16\left(\frac{1}{2e} - 2\ln\min_{\mathbf{y} \in \mathcal{Y}} \pi_{\theta(0)}(\mathbf{y}|\mathbf{x})\right)}$:*

$$t_\gamma \leq \frac{2|\mathcal{S}|}{(1-\rho)^3} \cdot \left( \frac{1}{c} - \frac{1}{\rho} \right) = \mathcal{O}(1),$$

*where $\rho := \frac{\mathbb{E}_{\mathbf{y} \sim \pi_{\theta(0)}(\cdot|\mathbf{x})}[r_G(\mathbf{x},\mathbf{y})]+\gamma+1}{r_G(\mathbf{x},\mathbf{y}^\gamma)+1} \in (0,1)$, while when using $r'_{\mathrm{RM}}$:*

$$t_\gamma \geq \frac{|\mathcal{S}|}{\sqrt{2}} \left( 1 - \exp\left( -\frac{\gamma}{2} \right) \right) \cdot T = \Omega(T).$$

- $r'_{\mathrm{RM}}$ *is a better teacher for initial policies in $\Pi'$.* *If $\pi_{\theta(0)} \in \Pi'$, when using $r'_{\mathrm{RM}}$ for maximizing the RLHF objective with $\lambda \leq \frac{c(1-\rho')^3}{16\left(\frac{1}{2e} - 2\ln\min_{\mathbf{y} \in \mathcal{Y}} \pi_{\theta(0)}(\mathbf{y}|\mathbf{x})\right)}$:*

---

[7] More precisely, the lower bound on $t_\gamma$ follows from Proposition 3 (Appendix D.6), which is analogous to Theorem 1, but applies to tabular policies instead of general autoregressive policies.

$$t_\gamma \leq \frac{2|\mathcal{S}|}{(1-\rho')^3} \cdot \left(\frac{1}{c} - \frac{1}{\rho'}\right) = \mathcal{O}(1),$$

*where $\rho' := \frac{\mathbb{E}_{\mathbf{y}\sim\pi_{\theta(0)}(\cdot|\mathbf{x})}[r_G(\mathbf{x},\mathbf{y})]+\gamma+1}{r_G(\mathbf{x},\mathbf{y}'^\gamma)+1} \in (0,1)$, while when using $r_{\mathrm{RM}}$:*

$$t_\gamma \geq \frac{|\mathcal{S}|}{\sqrt{2}}\left(1 - \exp\left(-\frac{\gamma}{2}\right)\right) \cdot T = \Omega(T).$$

*Proof sketch (full proof of Theorem 6 is in Appendix D.9).* For outputs $\mathbf{y}^\gamma, \mathbf{y}'^\gamma \in \mathcal{Y}$ of sufficiently high ground truth reward, we let $r_{\mathrm{RM}}$ assign the same low reward to all outputs except $\mathbf{y}^\gamma$, which is assigned a high reward. We define $r'_{\mathrm{RM}}$ similarly, but instead of assigning high reward to $\mathbf{y}^\gamma$, it assigns high reward to $\mathbf{y}'^\gamma$. The initial policy families $\Pi, \Pi'$ are chosen so that $r_{\mathrm{RM}}$ and $r'_{\mathrm{RM}}$ induce low reward variance for policies in $\Pi'$ and $\Pi$, respectively. Specifically, policies in $\Pi$ give negligible probability to $\mathbf{y}'^\gamma$, whereas policies in $\Pi'$ give negligible probability to $\mathbf{y}^\gamma$. Theorem 1 then yields the lower bounds on $t_\gamma$.[7] For the upper bounds on $t_\gamma$, we follow arguments identical to those used in the proof of Theorem 2. Note that we account for potential differences in optimization due to initial policy quality by taking all policies in $\Pi$ and $\Pi'$ to have the same expected ground truth reward. $\square$

## C  Perfectly Accurate Reward Models Are Optimal for Best-of-N

An alternative method to RLHF for aligning language models is Best-of-N sampling [52]. Instead of using a reward model $r_{\mathrm{RM}}$ for policy gradient training, Best-of-N applies $r_{\mathrm{RM}}$ only at test-time for choosing the best output from multiple candidates. Specifically, given a prompt $\mathbf{x} \in \mathcal{X}$, Best-of-N samples $N \in \mathbb{N}$ independent candidate outputs $\mathbf{y}^1, \ldots, \mathbf{y}^N \in \mathcal{Y}$ from the policy $\pi$. Then, the output with highest reward according to $r_{\mathrm{RM}}$ is returned.

Theorem 2 (Section 3.3) showed that perfectly accurate reward models (*i.e.*, reward models that rank all outputs correctly) can underperform less accurate models in RLHF. In contrast, for Best-of-N, Proposition 1 establishes that perfectly accurate reward models are always optimal. This highlights that the role of accuracy — and, more broadly, what makes a good reward model — can vary depending on the choice of alignment method.

**Proposition 1.** *For any policy $\pi$ and reward model $r_{\mathrm{RM}} : \mathcal{X} \times \mathcal{Y} \to [-1,1]$, let $\pi^{\mathrm{BoN}}_{r_{\mathrm{RM}}}$ be the Best-of-N policy induced by $\pi$ and $r_{\mathrm{RM}}$, for $N \in \mathbb{N}$. Given a prompt $\mathbf{x} \in \mathcal{X}$, suppose that $r_{\mathrm{RM}}$ has perfect accuracy, i.e., $\mathrm{acc}_{\mathbf{x},\mathcal{D}}(r_{\mathrm{RM}}) = 1$ for any distribution $\mathcal{D}$ over unordered output pairs. Then, $r_{\mathrm{RM}}$ is optimal for Best-of-N in the sense that:*

$$\mathbb{E}_{\mathbf{y}\sim\pi^{\mathrm{BoN}}_{r_{\mathrm{RM}}}(\cdot|\mathbf{x})}[r_G(\mathbf{x},\mathbf{y})] = \max_{r'_{\mathrm{RM}}:\mathcal{X}\times\mathcal{Y}\to[-1,1]} \mathbb{E}_{\mathbf{y}\sim\pi^{\mathrm{BoN}}_{r'_{\mathrm{RM}}}(\cdot|\mathbf{x})}[r_G(\mathbf{x},\mathbf{y})].$$

*Proof.* For candidate outputs $\mathbf{y}^1, \ldots, \mathbf{y}^N \in \mathcal{Y}$, we let $\mathbf{y}^{\mathrm{BoN}}_{r_{\mathrm{RM}}}(\mathbf{y}^1, \ldots, \mathbf{y}^N)$ be an output with maximal proxy reward $r_{\mathrm{RM}}$, *i.e.*:

$$\mathbf{y}^{\mathrm{BoN}}_{r_{\mathrm{RM}}}(\mathbf{y}^1, \ldots, \mathbf{y}^N) \in \mathrm{argmax}_{\mathbf{y}\in\{\mathbf{y}^1,\ldots,\mathbf{y}^N\}} r_{\mathrm{RM}}(\mathbf{x},\mathbf{y}).$$

Since $r_{\mathrm{RM}}$ is perfectly accurate, the ordering of $\mathbf{y}^1, \ldots, \mathbf{y}^N$ according to $r_{\mathrm{RM}}$ and $r_G$ is the same. In particular:

$$\mathrm{argmax}_{\mathbf{y}\in\{\mathbf{y}^1,\ldots,\mathbf{y}^N\}} r_{\mathrm{RM}}(\mathbf{x},\mathbf{y}) = \mathrm{argmax}_{\mathbf{y}\in\{\mathbf{y}^1,\ldots,\mathbf{y}^N\}} r_G(\mathbf{x},\mathbf{y}).$$

This implies that:

$$r_G\left(\mathbf{x}, \mathbf{y}^{\mathrm{BoN}}_{r_{\mathrm{RM}}}(\mathbf{y}^1, \ldots, \mathbf{y}^N)\right) = \max_{\mathbf{y}\in\{\mathbf{y}^1,\ldots,\mathbf{y}^N\}} r_G(\mathbf{x},\mathbf{y}).$$

Thus, for any alternative reward model $r'_{\mathrm{RM}} : \mathcal{X} \times \mathcal{Y} \to [-1,1]$, it readily follows that:

$$\begin{aligned}
\mathbb{E}_{\mathbf{y}\sim\pi^{\mathrm{BoN}}_{r_{\mathrm{RM}}}(\cdot|\mathbf{x})}[r_G(\mathbf{x},\mathbf{y})] &= \mathbb{E}_{\mathbf{y}^1,\ldots,\mathbf{y}^N\overset{\mathrm{i.i.d.}}{\sim}\pi(\cdot|\mathbf{x})}\left[r_G\left(\mathbf{x}, \mathbf{y}^{\mathrm{BoN}}_{r_{\mathrm{RM}}}(\mathbf{y}^1, \ldots, \mathbf{y}^N)\right)\right] \\
&= \mathbb{E}_{\mathbf{y}^1,\ldots,\mathbf{y}^N\overset{\mathrm{i.i.d.}}{\sim}\pi(\cdot|\mathbf{x})}\left[\max_{\mathbf{y}\in\{\mathbf{y}^1,\ldots,\mathbf{y}^N\}} r_G(\mathbf{x},\mathbf{y})\right] \\
&\geq \mathbb{E}_{\mathbf{y}^1,\ldots,\mathbf{y}^N\overset{\mathrm{i.i.d.}}{\sim}\pi(\cdot|\mathbf{x})}\left[r_G\left(\mathbf{x}, \mathbf{y}^{\mathrm{BoN}}_{r'_{\mathrm{RM}}}(\mathbf{y}^1, \ldots, \mathbf{y}^N)\right)\right] \\
&= \mathbb{E}_{\mathbf{y}\sim\pi^{\mathrm{BoN}}_{r'_{\mathrm{RM}}}(\cdot|\mathbf{x})}[r_G(\mathbf{x},\mathbf{y})].
\end{aligned}$$

$\square$

# D  Deferred Proofs

In this appendix, we provide proofs for our main theoretical results: Theorems 4 to 6.

- Appendix D.1 introduces additional notation.
- Appendix D.2 establishes several auxiliary lemmas.
- For general autoregressive policies (Equation (2)), Appendix D.3 upper bounds the gradient norm of the KL-regularized RLHF objective by a quantity that depends on the reward variance. This generalizes Theorem 1 in [64], which considered only a non-KL-regularized objective.
- Appendix D.4 proves Theorem 4, relying on the upper bound from Appendix D.3.
- Moving to tabular policies (Equation (3)), Appendix D.5 includes auxiliary lemmas.
- We then establish two key results toward proving Theorems 5 and 6: *(i)* a lower bound on the reward increase rate, similar to that given in Theorem 4 for general autoregressive policies — see Appendix D.6; and *(ii)* sufficient conditions for fast ground truth reward increase — see Appendix D.7.
- Lastly, the proofs of Theorems 5 and 6 are delivered in Appendices D.8 and D.9, respectively.

## D.1  Notation

Recall the RLHF objective from Equation (1), defined for a policy $\pi_\theta$:

$$\phi_{\mathrm{RLHF}}(\theta) := \mathbb{E}_{\mathbf{x} \sim \mathcal{S}}\Big[\mathbb{E}_{\mathbf{y} \sim \pi_\theta(\cdot|\mathbf{x})}\big[r_{\mathrm{RM}}(\mathbf{x}, \mathbf{y})\big] - \lambda \cdot \mathrm{KL}\big(\pi_\theta(\cdot|\mathbf{x})||\pi_{\theta_{\mathrm{ref}}}(\cdot|\mathbf{x})\big)\Big].$$

For a prompt $\mathbf{x} \in \mathcal{X}$ and reward model $r_{\mathrm{RM}} : \mathcal{X} \times \mathcal{Y} \to [-1, 1]$, we denote:

$$V_{\mathrm{RM}}(\theta; \mathbf{x}) := \mathbb{E}_{\mathbf{y} \sim \pi_\theta(\cdot|\mathbf{x})}[r_{\mathrm{RM}}(\mathbf{x}, \mathbf{y})],$$

$$V_{\mathrm{G}}(\theta; \mathbf{x}) := \mathbb{E}_{\mathbf{y} \sim \pi_\theta(\cdot|\mathbf{x})}[r_{\mathrm{G}}(\mathbf{x}, \mathbf{y})].$$

Furthermore, we denote the RLHF objective for a single prompt $\mathbf{x} \in \mathcal{S}$ by:

$$\phi_{\mathrm{RLHF}}(\theta; \mathbf{x}) := V_{\mathrm{RM}}(\theta; \mathbf{x}) - \lambda \cdot \mathrm{KL}\big(\pi_\theta(\cdot|\mathbf{x})||\pi_{\theta_{\mathrm{ref}}}(\cdot|\mathbf{x})\big)$$

$$= \mathbb{E}_{\mathbf{y} \sim \pi_\theta(\cdot|\mathbf{x})}\left[r_{\mathrm{RM}}(\mathbf{x}, \mathbf{y}) - \lambda \cdot \ln \frac{\pi_\theta(\mathbf{y}|\mathbf{x})}{\pi_{\theta_{\mathrm{ref}}}(\mathbf{y}|\mathbf{x})}\right]$$

$$= \mathbb{E}_{\mathbf{y} \sim \pi_\theta(\cdot|\mathbf{x})}\big[r_{\mathrm{RM}}^{\mathrm{KL}}(\mathbf{x}, \mathbf{y}; \theta)\big],$$

where $r_{\mathrm{RM}}^{\mathrm{KL}}(\mathbf{x}, \mathbf{y}; \theta) := r_{\mathrm{RM}}(\mathbf{x}, \mathbf{y}) - \lambda \cdot \ln \frac{\pi_\theta(\mathbf{y}|\mathbf{x})}{\pi_{\theta_{\mathrm{ref}}}(\mathbf{y}|\mathbf{x})}$.

Lastly, we let $\mathbf{e}_k$ denote the $k$th standard basis vector, with dimension dependent on context, $\|\cdot\|$ denote the Euclidean norm, $\|\cdot\|_1$ denote the $\ell_1$ norm, and $\|\cdot\|_2$ denote the spectral norm of a matrix.

## D.2  Useful Lemmas: General

We make use of the following lemmas throughout the proofs.

**Lemma 1.** *Let $g : \mathbb{R} \to \mathbb{R}$ be a continuously differentiable function satisfying $g(t) > 0$ and $\frac{d}{dt}g(t) \leq c \cdot g(t)^p$ for all $t \geq 0$, where $c > 0$ and $p > 1$ are some fixed constants. Then, for all $t \in \big[0, 1/(c(p-1)g(0)^{p-1})\big)$ it holds that:*

$$g(t) \leq \frac{g(0)}{\big(1 - c(p-1)g(0)^{p-1} \cdot t\big)^{\frac{1}{p-1}}}.$$

*Proof.* For $s \in [0, t]$, dividing both sides of the inequality $c \cdot g(s)^p \geq \frac{d}{ds}g(s)$ by $g(s)^p$ and integrating gives:

$$c \cdot t \geq \int_0^t \frac{\frac{d}{ds}g(s)}{g(s)^p} ds = \left(-\frac{1}{p-1}g(s)^{-(p-1)}\right)\bigg|_{s=0}^t = \frac{1}{p-1}g(0)^{-(p-1)} - \frac{1}{p-1}g(t)^{-(p-1)}.$$

Rearranging the inequality above yields the desired upper bound on $g(t)$. $\qquad\square$

**Lemma 2.** *Let $g : \mathbb{R} \to \mathbb{R}$ be a continuously differentiable function satisfying $g(t) > 0$ and $\frac{d}{dt}g(t) \geq c \cdot g(t)^p$ for all $t \geq 0$, where $c > 0$ and $p > 1$ are some fixed constants. Then, for all $t \in \left[0, 1/(c(p-1)g(0)^{p-1})\right)$ it holds that:*

$$g(t) \geq \frac{g(0)}{\left(1 - c(p-1)g(0)^{p-1} \cdot t\right)^{\frac{1}{p-1}}} \,.$$

*Proof.* The proof is analogous to that of Lemma 1. □

**Lemma 3.** *Let $p, q$ be distributions over $\mathcal{Y}$, with $q(\mathbf{y}) > 0$ for all $\mathbf{y} \in \mathcal{Y}$. Then:*

$$\mathrm{KL}(p||q) \leq \ln \frac{1}{\min_{\mathbf{y} \in \mathcal{Y}} q(\mathbf{y})} \,.$$

*Proof.* We may write the KL divergence between $p$ and $q$ as:

$$\mathrm{KL}(p||q) = \sum_{\mathbf{y} \in \mathcal{Y}} p(\mathbf{y}) \ln \frac{p(\mathbf{y})}{q(\mathbf{y})} = \sum_{\mathbf{y} \in \mathcal{Y}} p(\mathbf{y}) \ln p(\mathbf{y}) + \sum_{\mathbf{y} \in \mathcal{Y}} p(\mathbf{y}) \ln \frac{1}{q(\mathbf{y})} \,.$$

The upper bound on $\mathrm{KL}(p||q)$ follows by noticing that $\sum_{\mathbf{y} \in \mathcal{Y}} p(\mathbf{y}) \ln p(\mathbf{y}) \leq 0$, since it is the negative (Shannon) entropy of $p$, and $\sum_{\mathbf{y} \in \mathcal{Y}} p(\mathbf{y}) \ln \frac{1}{q(\mathbf{y})} \leq \ln \frac{1}{\min_{\mathbf{y} \in \mathcal{Y}} q(\mathbf{y})}$. □

**Lemma 4.** *Let $p, q$ be distributions over $\mathcal{Y}$. Then, for any $\mathbf{y} \in \mathcal{Y}$ with $p(\mathbf{y}) > 0$ and $q(\mathbf{y}) > 0$ it holds that:*

$$-p(\mathbf{y})^2 \ln \frac{p(\mathbf{y})}{q(\mathbf{y})} \leq \frac{q(\mathbf{y})^2}{2e} \,.$$

*Proof.* Define $h : (0, 1] \to \mathbb{R}$ as follows:

$$h(u) := -u^2 \ln \frac{u}{q(\mathbf{y})} \,.$$

The derivative of $h$ is given by:

$$\frac{d}{du}h(u) = -2u \ln \frac{u}{q(\mathbf{y})} - u = -u\left(2 \ln \frac{u}{q(\mathbf{y})} + 1\right) \,.$$

Notice that

$$2 \ln \frac{q(\mathbf{y})e^{-1/2}}{q(\mathbf{y})} = -1 \,,$$

and so $\frac{d}{du}h(q(\mathbf{y})e^{-1/2}) = 0$. Furthermore, since $\ln(u/q(\mathbf{y}))$ is monotonically increasing with respect to $u$, then $\frac{d}{du}h(u) > 0$ for $u \in (0, q(\mathbf{y})e^{-1/2})$ and $\frac{d}{du}h(u) < 0$ for $u \in (q(\mathbf{y})e^{-1/2}, 1]$. This implies that the maximal value of $h(u)$ over $(0, 1]$ is attained at $u = q(\mathbf{y})e^{-1/2}$, *i.e.*, for all $u \in (0, 1]$:

$$h(u) = -u^2 \ln \frac{u}{q(\mathbf{y})} \leq \frac{q(\mathbf{y})^2}{2e} \,,$$

from which it follows that $-p(\mathbf{y})^2 \ln \frac{p(\mathbf{y})}{q(\mathbf{y})} \leq \frac{q(\mathbf{y})^2}{2e}$. □

### D.3 Low Reward Variance Implies Vanishing Gradient

For a general autoregressive policy $\pi_\theta$ (Equation (2)), this appendix generalizes the gradient norm upper bound from [64] to the KL-regularized RLHF objective (Equation (1)). Given a reward model $r_{\mathrm{RM}}$ and prompt $\mathbf{x}$, the bound depends on the reward variance with respect to the KL-regularized reward, *i.e.*, on

$$\mathrm{Var}_{\mathbf{y} \sim \pi_\theta(\cdot|\mathbf{x})}\left[r_{\mathrm{RM}}^{\mathrm{KL}}(\mathbf{x}, \mathbf{y}; \theta)\right] \,,$$

where $r_{\mathrm{RM}}^{\mathrm{KL}}(\mathbf{x}, \mathbf{y}; \theta) := r_{\mathrm{RM}}(\mathbf{x}, \mathbf{y}) - \lambda \cdot \ln \frac{\pi_\theta(\mathbf{y}|\mathbf{x})}{\pi_{\theta_{\mathrm{ref}}}(\mathbf{y}|\mathbf{x})}$.

**Proposition 2.** *Let $\pi_\theta$ be a general autoregressive policy (Equation (2)). For any prompt $\mathbf{x} \in \mathcal{X}$, reward model $r_{\mathrm{RM}} : \mathcal{X} \times \mathcal{Y} \to [-1, 1]$, and $M > 1$, if $r_{\mathrm{RM}}^{\mathrm{KL}}(\mathbf{x}, \mathbf{y}; \theta) \in [-M, M]$ for all $\mathbf{y} \in \mathcal{Y}$, then:*

$$\left\| \nabla \phi_{\mathrm{RLHF}}(\theta; \mathbf{x}) \right\| = \left\| \nabla \, \mathbb{E}_{\mathbf{y} \sim \pi_\theta(\cdot|\mathbf{x})} \left[ r_{\mathrm{RM}}^{\mathrm{KL}}(\mathbf{x}, \mathbf{y}; \theta) \right] \right\| \le 6 M^{\frac{1}{3}} L J_{\mathbf{x}} \cdot \mathrm{Var}_{\mathbf{y} \sim \pi_\theta(\cdot|\mathbf{x})} \left[ r_{\mathrm{RM}}^{\mathrm{KL}}(\mathbf{x}, \mathbf{y}; \theta) \right]^{\frac{1}{3}}.$$

*In the above, $J_{\mathbf{x}} := \sup_{\mathbf{y} \in \mathcal{Y}, l \in [L]} \left\| \mathbf{J}_{f_\theta(\mathbf{x}, \mathbf{y}_{<l})} \right\|_2$, where $\mathbf{J}_{f_\theta(\mathbf{x}, \mathbf{y}_{<l})}$ denotes the Jacobian of $f_\theta(\mathbf{x}, \mathbf{y}_{<l})$ with respect to $\theta$, and $L$ is the output sequence length.*

*Proof.* We begin by differentiating $\phi_{\mathrm{RLHF}}(\theta; \mathbf{x}) = \mathbb{E}_{\mathbf{y} \sim \pi_\theta(\cdot|\mathbf{x})} \left[ r_{\mathrm{RM}}^{\mathrm{KL}}(\mathbf{x}, \mathbf{y}; \theta) \right]$ with respect to $\theta$ using the log-derivative trick:

$$\nabla \phi_{\mathrm{RLHF}}(\theta; \mathbf{x}) = \nabla \, \mathbb{E}_{\mathbf{y} \sim \pi_\theta(\cdot|\mathbf{x})} \left[ r_{\mathrm{RM}}(\mathbf{x}, \mathbf{y}) - \lambda \cdot \ln \frac{\pi_\theta(\mathbf{y}|\mathbf{x})}{\pi_{\theta_{\mathrm{ref}}}(\mathbf{y}|\mathbf{x})} \right]$$

$$= \mathbb{E}_{\mathbf{y} \sim \pi_\theta(\cdot|\mathbf{x})} \left[ r_{\mathrm{RM}}^{\mathrm{KL}}(\mathbf{x}, \mathbf{y}; \theta) \nabla \ln \pi_\theta(\mathbf{y}|\mathbf{x}) \right] - \lambda \, \mathbb{E}_{\mathbf{y} \sim \pi_\theta(\cdot|\mathbf{x})} \left[ \nabla \ln \pi_\theta(\mathbf{y}|\mathbf{x}) \right].$$

Since $\mathbb{E}_{\mathbf{y} \sim \pi_\theta(\cdot|\mathbf{x})}[\nabla \ln \pi_\theta(\mathbf{y}|\mathbf{x})] = \sum_{\mathbf{y} \in \mathcal{Y}} \nabla \pi_\theta(\mathbf{y}|\mathbf{x}) = \nabla \sum_{\mathbf{y} \in \mathcal{Y}} \pi_\theta(\mathbf{y}|\mathbf{x}) = \nabla 1 = 0$, this implies that:

$$\nabla \phi_{\mathrm{RLHF}}(\theta; \mathbf{x}) = \mathbb{E}_{\mathbf{y} \sim \pi_\theta(\cdot|\mathbf{y})} \left[ r_{\mathrm{RM}}^{\mathrm{KL}}(\mathbf{x}, \mathbf{y}; \theta) \nabla \ln \pi_\theta(\mathbf{y}|\mathbf{x}) \right].$$

Recall, as stated in Section 3.1, that we take $\mathcal{Y} = \mathcal{V}^L$ without loss of generality, *i.e.*, all sequences in $\mathcal{Y}$ are of the maximal length $L$. Thus, by the autoregressive definition of $\pi_\theta(\mathbf{y}|\mathbf{x})$ (Equation (2)) and the chain rule, we arrive at:

$$\nabla \phi_{\mathrm{RLHF}}(\theta; \mathbf{x}) = \mathbb{E}_{\mathbf{y} \sim \pi_\theta(\cdot|\mathbf{x})} \left[ r_{\mathrm{RM}}^{\mathrm{KL}}(\mathbf{x}, \mathbf{y}; \theta) \sum_{l=1}^{L} \nabla \ln \pi_\theta(\mathbf{y}_l|\mathbf{x}, \mathbf{y}_{<l}) \right]$$

$$= \mathbb{E}_{\mathbf{y} \sim \pi_\theta(\cdot|\mathbf{x})} \left[ r_{\mathrm{RM}}^{\mathrm{KL}}(\mathbf{x}, \mathbf{y}; \theta) \sum_{l=1}^{L} \nabla \ln \mathrm{softmax}(f_\theta(\mathbf{x}, \mathbf{y}_{<l}))_{\mathbf{y}_l} \right]$$

$$= \sum_{\mathbf{y} \in \mathcal{Y}} \pi_\theta(\mathbf{y}|\mathbf{x}) r_{\mathrm{RM}}^{\mathrm{KL}}(\mathbf{x}, \mathbf{y}; \theta) \sum_{l=1}^{L} \mathbf{J}_{f_\theta(\mathbf{x}, \mathbf{y}_{<l})}^\top (\mathbf{e}_{\mathbf{y}_l} - \pi_\theta(\cdot|\mathbf{x}, \mathbf{y}_{<l})),$$

where, with slight abuse of notation, $\pi_\theta(\cdot|\mathbf{x}, \mathbf{y}_{<l}) = \mathrm{softmax}(f_\theta(\mathbf{x}, \mathbf{y}_{<l}))$.

Now, for a constant $c > 0$ to be determined later, denote by $\mathcal{Y}_c$ the set of outputs whose KL-regularized rewards deviate by more than $c$ from the expected KL-regularized reward, *i.e.*:

$$\mathcal{Y}_c := \left\{ \mathbf{y} \in \mathcal{Y} : \left| r_{\mathrm{RM}}^{\mathrm{KL}}(\mathbf{x}, \mathbf{y}; \theta) - \phi_{\mathrm{RLHF}}(\theta; \mathbf{x}) \right| > c \right\}.$$

Defining a modified reward function $\tilde{r} : \mathcal{X} \times \mathcal{Y} \to [-M, M]$ by:

$$\tilde{r}(\mathbf{x}, \mathbf{y}) := \begin{cases} r_{\mathrm{RM}}^{\mathrm{KL}}(\mathbf{x}, \mathbf{y}; \theta) & , \; \mathbf{y} \notin \mathcal{Y}_c \\ \phi_{\mathrm{RLHF}}(\theta; \mathbf{x}) & , \; \mathbf{y} \in \mathcal{Y}_c \end{cases},$$

we can write $\nabla \phi_{\mathrm{RLHF}}(\mathbf{x}; \theta)$ as follows:

$$\nabla \phi_{\mathrm{RLHF}}(\theta; \mathbf{x}) = \underbrace{\sum_{\mathbf{y} \in \mathcal{Y}} \pi_\theta(\mathbf{y}|\mathbf{x}) \tilde{r}(\mathbf{x}, \mathbf{y}) \sum_{l=1}^{L} \mathbf{J}_{f_\theta(\mathbf{x}, \mathbf{y}_{<l})}^\top (\mathbf{e}_{\mathbf{y}_l} - \pi_\theta(\cdot|\mathbf{x}, \mathbf{y}_{<l}))}_{(I)}$$

$$+ \underbrace{\sum_{\mathbf{y} \in \mathcal{Y}_c} \pi_\theta(\mathbf{y}|\mathbf{x}) \left[ r_{\mathrm{RM}}^{\mathrm{KL}}(\mathbf{x}, \mathbf{y}; \theta) - \tilde{r}(\mathbf{x}, \mathbf{y}) \right] \sum_{l=1}^{L} \mathbf{J}_{f_\theta(\mathbf{x}, \mathbf{y}_{<l})}^\top (\mathbf{e}_{y_l} - \pi_\theta(\cdot|\mathbf{x}, \mathbf{y}_{<l}))}_{(II)}.$$

We upper bound the Euclidean norms of $(I)$ and $(II)$ separately. Starting with $(II)$, by Chebyshev's inequality we know that:

$$\pi_\theta(\mathcal{Y}_c|\mathbf{x}) \le \frac{\mathrm{Var}_{\mathbf{y} \sim \pi_\theta(\cdot|\mathbf{x})} \left[ r_{\mathrm{RM}}^{\mathrm{KL}}(\mathbf{x}, \mathbf{y}; \theta) \right]}{c^2}.$$

Notice that $\|\mathbf{e}_{\mathbf{y}_l} - \pi_\theta(\cdot|\mathbf{x}, \mathbf{y}_{<l})\| \le \|\mathbf{e}_{\mathbf{y}_l} - \pi_\theta(\cdot|\mathbf{x}, \mathbf{y}_{<l})\|_1 \le 2$ for all $\mathbf{y} \in \mathcal{Y}$. Thus, for any $\mathbf{y} \in \mathcal{Y}_c$ and $l \in \{1, \ldots, L\}$:

$$\left\|\mathbf{J}_{f_\theta(\mathbf{x}, \mathbf{y}_{<l})}^\top (\mathbf{e}_{\mathbf{y}_l} - \pi_\theta(\cdot|\mathbf{x}, \mathbf{y}_{<l}))\right\| \le \left\|\mathbf{J}_{f_\theta(\mathbf{x}, \mathbf{y}_{<l})}\right\|_2 \cdot \left\|(\mathbf{e}_{\mathbf{y}_l} - \pi_\theta(\cdot|\mathbf{x}, \mathbf{y}_{<l}))\right\| \le 2 J_{\mathbf{x}}.$$

Since the KL-regularized rewards lie in $[-M, M]$, by the triangle inequality this implies that:

$$\|(II)\| \le 4MLJ_{\mathbf{x}} \cdot \pi_\theta(\mathcal{Y}_c|\mathbf{x}) \le 4MLJ_{\mathbf{x}} \cdot \frac{\mathrm{Var}_{\mathbf{y} \sim \pi_\theta(\cdot|\mathbf{x})}\left[r^{\mathrm{KL}}(\mathbf{x}, \mathbf{y})\right]}{c^2}. \tag{5}$$

As for $(I)$, denoting for $\mathbf{y}_{<l} \in \mathcal{V}^{l-1}$:

$$\mathbf{a}^{(\mathbf{y}_{<l})} := \sum\nolimits_{\mathbf{y}_{\ge l} \in \mathcal{V}^{L-l+1}} \pi_\theta(\mathbf{y}_{\ge l}|\mathbf{x}, \mathbf{y}_{<l})\tilde{r}(\mathbf{x}, \mathbf{y})(\mathbf{e}_{\mathbf{y}_l} - \pi_\theta(\cdot|\mathbf{x}, \mathbf{y}_{<l})) \in \mathbb{R}^{|\mathcal{V}|},$$

where $\mathbf{y}_{\ge l} := (\mathbf{y}_l, \ldots, \mathbf{y}_L)$, we have that:

$$(I) = \sum\nolimits_{l=1}^L \sum\nolimits_{\mathbf{y}_{<l} \in \mathcal{V}^{l-1}} \pi_\theta(\mathbf{y}_{<l}|\mathbf{x}) \mathbf{J}_{f_\theta(\mathbf{x}, \mathbf{y}_{<l})}^\top \mathbf{a}^{(\mathbf{y}_{<l})}.$$

The $y$th entry of $\mathbf{a}^{(\mathbf{y}_{<l})}$ is given by:

$$\begin{aligned}
a_y^{(\mathbf{y}_{<l})} &= \sum\nolimits_{\mathbf{y}_{\ge l+1} \in \mathcal{V}^{L-l}} \pi_\theta(\mathbf{y}_{\ge l+1}|\mathbf{x}, \mathbf{y}_{<l}, y)\pi_\theta(y|\mathbf{x}, \mathbf{y}_{<l})\tilde{r}(\mathbf{x}, \mathbf{y}_{<l}, y, \mathbf{y}_{\ge l+1}) \\
&\quad - \sum\nolimits_{\mathbf{y}_{\ge l} \in \mathcal{V}^{L-l+1}} \pi_\theta(\mathbf{y}_{\ge l}|\mathbf{x}, \mathbf{y}_{<l})\pi_\theta(y|\mathbf{x}, \mathbf{y}_{<l})\tilde{r}(\mathbf{x}, \mathbf{y}) \\
&= \pi_\theta(y|\mathbf{x}, \mathbf{y}_{<l})\Big( \mathbb{E}_{\mathbf{y}_{\ge l+1} \sim \pi_\theta(\cdot|\mathbf{x}, \mathbf{y}_{<l}, y)}[\tilde{r}(\mathbf{x}, \mathbf{y}_{<l}, y, \mathbf{y}_{\ge l+1})] \\
&\quad - \mathbb{E}_{\mathbf{y}_{\ge l} \sim \pi_\theta(\cdot|\mathbf{x}, \mathbf{y}_{<l})}[\tilde{r}(\mathbf{x}, \mathbf{y})]\Big).
\end{aligned}$$

Since $\tilde{r}$ satisfies by construction $|\tilde{r}(\mathbf{x}, \mathbf{y}) - \phi_{\mathrm{RLHF}}(\theta; \mathbf{x})| \le c$ for all $\mathbf{y} \in \mathcal{Y}$, we can bound the difference of expectations in the equation above through adding and subtracting $\phi_{\mathrm{RLHF}}(\theta; \mathbf{x})$ and the triangle inequality:

$$\begin{aligned}
&\left|\mathbb{E}_{\mathbf{y}_{\ge l+1} \sim \pi_\theta(\cdot|\mathbf{x}, \mathbf{y}_{<l}, y)}[\tilde{r}(\mathbf{x}, \mathbf{y}_{<l}, y, \mathbf{y}_{\ge l+1})] - \mathbb{E}_{\mathbf{y}_{\ge l} \sim \pi_\theta(\cdot|\mathbf{x}, \mathbf{y}_{<l})}[\tilde{r}(\mathbf{x}, \mathbf{y})]\right| \\
&\le \left|\mathbb{E}_{\mathbf{y}_{\ge l+1} \sim \pi_\theta(\cdot|\mathbf{x}, \mathbf{y}_{<l}, y)}[\tilde{r}(\mathbf{x}, \mathbf{y}_{<l}, y, \mathbf{y}_{\ge l+1})] - \phi_{\mathrm{RLHF}}(\theta; \mathbf{x})\right| \\
&\quad + \left|\phi_{\mathrm{RLHF}}(\theta; \mathbf{x}) - \mathbb{E}_{\mathbf{y}_{\ge l} \sim \pi_\theta(\cdot|\mathbf{x}, \mathbf{y}_{<l})}[\tilde{r}(\mathbf{x}, \mathbf{y})]\right| \le 2c.
\end{aligned}$$

Thus:

$$\left\|\mathbf{a}^{(\mathbf{y}_{<l})}\right\| \le \left\|\mathbf{a}^{(\mathbf{y}_{<l})}\right\|_1 \le \sum\nolimits_{y \in \mathcal{V}} \pi_\theta(y|\mathbf{x}, \mathbf{y}_{<l}) \cdot 2c = 2c.$$

Recalling that:

$$(I) = \sum\nolimits_{l=1}^L \sum\nolimits_{\mathbf{y}_{<l} \in \mathcal{V}^{l-1}} \pi_\theta(\mathbf{y}_{<l}|\mathbf{x}) \mathbf{J}_{f_\theta(\mathbf{x}, \mathbf{y}_{<l})}^\top \mathbf{a}^{(\mathbf{y}_{<l})},$$

taking the norm of both sides and applying the triangle inequality yields:

$$\begin{aligned}
\|(I)\| &\le \sum\nolimits_{l=1}^L \sum\nolimits_{\mathbf{y}_{<l} \in \mathcal{V}^{l-1}} \pi_\theta(\mathbf{y}_{<l}|\mathbf{x}) J_{\mathbf{x}} \|\mathbf{a}^{(\mathbf{y}_{<l})}\| \\
&\le \sum\nolimits_{l=1}^L 2J_{\mathbf{x}} \cdot c \sum\nolimits_{\mathbf{y}_{<l} \in \mathcal{V}^{l-1}} \pi_\theta(\mathbf{y}_{<l}|\mathbf{x}) \\
&= 2LJ_{\mathbf{x}} \cdot c.
\end{aligned} \tag{6}$$

Combining the bounds on the norms of $(I)$ and $(II)$ (Equations (5) and (6)) then leads to:

$$\|\nabla\phi_{\mathrm{RLHF}}(\theta; \mathbf{x})\| \le 4MLJ_{\mathbf{x}} \cdot \frac{\mathrm{Var}_{\mathbf{y} \sim \pi_\theta(\cdot|\mathbf{x})}\left[r_{\mathrm{RM}}^{\mathrm{KL}}(\mathbf{x}, \mathbf{y}; \theta)\right]}{c^2} + 2LJ_{\mathbf{x}} \cdot c.$$

Lastly, choosing $c = M^{\frac{1}{3}} \mathrm{Var}_{\mathbf{y} \sim \pi_\theta(\cdot|\mathbf{x})}\left[r_{\mathrm{RM}}^{\mathrm{KL}}(\mathbf{x}, \mathbf{y}; \theta)\right]^{\frac{1}{3}}$ concludes the proof:

$$\|\nabla\phi_{\mathrm{RLHF}}(\theta; \mathbf{x})\| \le 6M^{\frac{1}{3}} LJ_{\mathbf{x}} \cdot \mathrm{Var}_{\mathbf{y} \sim \pi_\theta(\cdot|\mathbf{x})}\left[r_{\mathrm{RM}}^{\mathrm{KL}}(\mathbf{x}, \mathbf{y}; \theta)\right]^{\frac{1}{3}}.$$

$\square$

## D.4  Proof of Theorem 4

The proof assumes familiarity with the notation introduced in Appendix D.1.

We start by bounding how far the policy parameters $\theta(t)$ and KL-regularized rewards can deviate from their initial values until some time $T \geq 0$ of gradient flow. Specifically, Lemma 5 establishes that, if the reward variance that $r_{\mathrm{RM}}$ induces for $\pi_{\theta(0)} = \pi_{\theta_{\mathrm{ref}}}$ is low and the KL-regularized rewards are bounded, then the parameters stay close to their initial values. This follows by showing that, when the reward variance is low, the RLHF objective gradient vanishes (Proposition 2) and does not rapidly increase in norm. Lemma 6 then establishes that the KL-regularized rewards remain bounded as required in Lemma 5 for a sufficiently long time. Lastly, we combine both lemmas and translate the upper bound on the change in policy parameters to a lower bound on $t_\gamma$, *i.e.*, to a lower bound on the time until $\mathbb{E}_{\mathbf{y} \sim \pi_{\theta(t)}(\cdot|\mathbf{x})}[r(\mathbf{x}, \mathbf{y})] \geq \mathbb{E}_{\mathbf{y} \sim \pi_{\theta(0)}(\cdot|\mathbf{x})}[r(\mathbf{x}, \mathbf{y})] + \gamma$.

**Lemma 5.** *Under the setting of Theorem 4, let*

$$
T \in \left[0, \frac{1}{8L^2 J_T^2 \, \mathbb{E}_{\mathbf{x} \sim \mathcal{S}}\left[\mathrm{Var}_{\mathbf{y} \sim \pi_{\theta(0)}(\cdot|\mathbf{x})}[r_{\mathrm{RM}}(\mathbf{x}, \mathbf{y})]\right]^{1/3}}\right),
$$

*where* $J_T := \sup_{\mathbf{x} \in \mathcal{X}, \mathbf{y} \in \mathcal{Y}, l \in [L], t \in [0,T]} \left\| \mathbf{J}_{f_{\theta(t)}(\mathbf{x}, \mathbf{y}_{<l})} \right\|_2$. *If* $r_{\mathrm{RM}}^{\mathrm{KL}}(\mathbf{x}, \mathbf{y}; \theta(t)) \in [-8, 8]$ *for all* $\mathbf{x} \in \mathcal{S}$, $\mathbf{y} \in \mathcal{Y}$, *and* $t \in [0, T]$, *then:*

$$
\|\theta(T) - \theta(0)\| \leq \int_0^T \left\| \frac{d}{dt}\theta(t) \right\| dt
$$

$$
\leq \frac{3}{2LJ_T} \ln\left( \frac{1}{1 - 8L^2 J_T^2 \, \mathbb{E}_{\mathbf{x} \sim \mathcal{S}}\left[\mathrm{Var}_{\mathbf{y} \sim \pi_{\theta(0)}(\cdot|\mathbf{x})}[r_{\mathrm{RM}}(\mathbf{x}, \mathbf{y})]\right]^{1/3} \cdot T} \right).
$$

*Proof.* For all $t \geq 0$ and $\mathbf{x} \in \mathcal{S}$, we shorthand $\mathrm{Var}_{\mathbf{x}}^{\mathrm{KL}}(\theta(t)) := \mathrm{Var}_{\mathbf{y} \sim \pi_{\theta(t)}(\cdot|\mathbf{x})}\left[r_{\mathrm{RM}}^{\mathrm{KL}}(\mathbf{x}, \mathbf{y}; \theta(t))\right]$ and let $\mathrm{Var}^{\mathrm{KL}}(\theta(t)) := \mathbb{E}_{\mathbf{x} \sim \mathcal{S}}[\mathrm{Var}_{\mathbf{x}}^{\mathrm{KL}}(\theta(t))]$.

First, if at some $t \geq 0$ it holds that $\mathrm{Var}^{\mathrm{KL}}(\theta(t)) = \mathbb{E}_{\mathbf{x} \sim \mathcal{S}}[\mathrm{Var}_{\mathbf{x}}^{\mathrm{KL}}(\theta(t))] = 0$, then $\mathrm{Var}_{\mathbf{x}}^{\mathrm{KL}}(\theta(t)) = 0$ for all $\mathbf{x} \in \mathcal{S}$ at that time since the (KL-regularized) reward variance for a prompt is always non-negative. By Proposition 2, this implies that $\nabla \phi_{\mathrm{RLHF}}(\theta(t); \mathbf{x}) = 0$ for all $\mathbf{x} \in \mathcal{S}$, and so gradient flow is at a critical point at time $t$, *i.e.* $\nabla \phi_{\mathrm{RLHF}}(\theta(t)) = \mathbb{E}_{\mathbf{x} \sim \mathcal{S}}[\nabla \phi_{\mathrm{RLHF}}(\theta(t); \mathbf{x})] = 0$. Due to the uniqueness of the gradient flow solution and the existence of a solution that remains at $\theta(t)$ through time, in this case $\theta(0) = \theta(t)$ for all $t \geq 0$. Hence, the upper bound on $\|\theta(T) - \theta(0)\|$ trivially holds.

Now, consider the more interesting case where $\mathrm{Var}^{\mathrm{KL}}(\theta(t)) > 0$ for all $t \geq 0$. By the fundamental theorem of calculus and the triangle inequality:

$$
\|\theta(T) - \theta(0)\| = \left\| \int_0^T \tfrac{d}{dt}\theta(t)dt \right\|
$$

$$
\leq \int_0^T \left\| \tfrac{d}{dt}\theta(t) \right\| dt
$$

$$
= \int_0^T \|\nabla \phi_{\mathrm{RLHF}}(\theta(t))\| dt \,.
$$

For all $t \in [0, T]$, applying the triangle inequality, Proposition 2, and Jensen's inequality leads to:

$$
\begin{aligned}
\|\nabla \phi_{\mathrm{RLHF}}(\theta(t))\| &= \|\mathbb{E}_{\mathbf{x} \sim \mathcal{S}}[\nabla \phi_{\mathrm{RLHF}}(\theta(t); \mathbf{x})]\| \\
&\leq \mathbb{E}_{\mathbf{x} \sim \mathcal{S}}[\|\nabla \phi_{\mathrm{RLHF}}(\theta(t); \mathbf{x})\|] \\
&\leq 12 L J_T \cdot \mathbb{E}_{\mathbf{x} \sim \mathcal{S}}\left[\mathrm{Var}_{\mathbf{x}}^{\mathrm{KL}}(\theta(t))^{\frac{1}{3}}\right] \\
&\leq 12 L J_T \cdot \mathrm{Var}^{\mathrm{KL}}(\theta(t))^{\frac{1}{3}} \,.
\end{aligned} \tag{7}
$$

Hence:

$$
\|\theta(T) - \theta(0)\| \leq \int_0^T \|\tfrac{d}{dt}\theta(t)\| dt = \int_0^T \|\nabla \phi_{\mathrm{RLHF}}(\theta(t))\| dt \leq 12 L J_T \int_0^T \mathrm{Var}^{\mathrm{KL}}(\theta(t))^{\frac{1}{3}} dt \,.
$$

$$
\tag{8}
$$

We now bound the rate at which $\mathrm{Var}^{\mathrm{KL}}(\theta(t))$ can increase. In turn, this will yield an upper bound on the distance between $\theta(T)$ and $\theta(0)$. Differentiating $\mathrm{Var}^{\mathrm{KL}}(\theta(t))$ with respect to time results in:

$$\frac{d}{dt}\mathrm{Var}^{\mathrm{KL}}(\theta(t)) = \left\langle \nabla\mathrm{Var}^{\mathrm{KL}}(\theta(t)), \tfrac{d}{dt}\theta(t) \right\rangle = \left\langle \nabla\mathrm{Var}^{\mathrm{KL}}(\theta(t)), \nabla\phi_{\mathrm{RLHF}}(\theta(t)) \right\rangle . \qquad (9)$$

Let us compute the gradient of the variance given a prompt $\mathbf{x} \in \mathcal{S}$, *i.e.*, the gradient with respect to $\theta$ of $\mathrm{Var}^{\mathrm{KL}}_{\mathbf{x}}(\theta(t)) = \sum_{\mathbf{y}\in\mathcal{Y}} \pi_{\theta(t)}(\mathbf{y}|\mathbf{x}) \cdot \left[ r^{\mathrm{KL}}_{\mathrm{RM}}(\mathbf{x},\mathbf{y};\theta(t)) - \phi_{\mathrm{RLHF}}(\theta(t);\mathbf{x}) \right]^2$:

$$\nabla\mathrm{Var}^{\mathrm{KL}}_{\mathbf{x}}(\theta(t)) = \underbrace{\sum_{\mathbf{y}\in\mathcal{Y}} \left[ r^{\mathrm{KL}}_{\mathrm{RM}}(\mathbf{x},\mathbf{y};\theta(t)) - \phi_{\mathrm{RLHF}}(\theta(t);\mathbf{x}) \right]^2 \cdot \nabla\pi_{\theta(t)}(\mathbf{y}|\mathbf{x})}_{\text{denote by } \mathbf{a}}$$

$$\underbrace{- 2\sum_{\mathbf{y}\in\mathcal{Y}} \pi_{\theta(t)}(\mathbf{y}|\mathbf{x}) \left[ r^{\mathrm{KL}}_{\mathrm{RM}}(\mathbf{x},\mathbf{y};\theta(t)) - \phi_{\mathrm{RLHF}}(\theta(t);\mathbf{x}) \right] \cdot \nabla\phi_{\mathrm{RLHF}}(\theta(t);\mathbf{x})}_{\text{denote by } \mathbf{b}}$$

$$\underbrace{- 2\lambda\sum_{\mathbf{y}\in\mathcal{Y}} \pi_{\theta(t)}(\mathbf{y}|\mathbf{x}) \left[ r^{\mathrm{KL}}_{\mathrm{RM}}(\mathbf{x},\mathbf{y};\theta(t)) - \phi_{\mathrm{RLHF}}(\theta(t);\mathbf{x}) \right] \cdot \nabla\ln\pi_{\theta(t)}(\mathbf{y}|\mathbf{x})}_{\text{denote by } \mathbf{c}} .$$

We first consider the term $\mathbf{a}$. Since

$$\nabla\pi_{\theta(t)}(\mathbf{y}|\mathbf{x}) = \pi_{\theta(t)}(\mathbf{y}|\mathbf{x})\nabla\ln\pi_{\theta(t)}(\mathbf{y}|\mathbf{x})$$

$$= \pi_{\theta(t)}(\mathbf{y}|\mathbf{x})\nabla\sum_{l=1}^{L} \ln\pi_{\theta(t)}(\mathbf{y}_l|\mathbf{x},\mathbf{y}_{<l})$$

$$= \pi_{\theta(t)}(\mathbf{y}|\mathbf{x})\sum_{l=1}^{L} \mathbf{J}^{\top}_{f_{\theta(t)}(\mathbf{x},\mathbf{y}_{<l})}\left( \mathbf{e}_{\mathbf{y}_l} - \pi_{\theta(t)}(\cdot|\mathbf{x},\mathbf{y}_{<l}) \right),$$

where, with slight abuse of notation, $\pi_{\theta(t)}(\cdot|\mathbf{x},\mathbf{y}_{<l}) = \mathrm{softmax}(f_{\theta(t)}(\mathbf{x},\mathbf{y}_{<l}))$, we have that:

$$\langle \mathbf{a}, \nabla\phi_{\mathrm{RLHF}}(\theta(t)) \rangle$$

$$= \sum_{\mathbf{y}\in\mathcal{Y}} \pi_{\theta(t)}(\mathbf{y}|\mathbf{x}) \left[ r^{\mathrm{KL}}_{\mathrm{RM}}(\mathbf{x},\mathbf{y};\theta(t)) - \phi_{\mathrm{RLHF}}(\theta(t);\mathbf{x}) \right]^2$$

$$\cdot \sum_{l=1}^{L} \left\langle \mathbf{J}^{\top}_{f_{\theta(t)}(\mathbf{x},\mathbf{y}_{<l})}\left( \mathbf{e}_{\mathbf{y}_l} - \pi_{\theta(t)}(\cdot|\mathbf{x},\mathbf{y}_{<l}) \right), \nabla\phi_{\mathrm{RLHF}}(\theta(t)) \right\rangle$$

$$\leq \sum_{\mathbf{y}\in\mathcal{Y}} \pi_{\theta(t)}(\mathbf{y}|\mathbf{x}) \left[ r^{\mathrm{KL}}_{\mathrm{RM}}(\mathbf{x},\mathbf{y};\theta(t)) - \phi_{\mathrm{RLHF}}(\theta(t);\mathbf{x}) \right]^2 J_T$$

$$\cdot \sum_{l=1}^{L} \left\| \mathbf{e}_{\mathbf{y}_l} - \pi_{\theta(t)}(\cdot|\mathbf{x},\mathbf{y}_{<l}) \right\| \left\| \nabla\phi_{\mathrm{RLHF}}(\theta(t)) \right\|$$

$$\leq 24L^2 J_T^2 \cdot \mathrm{Var}^{\mathrm{KL}}(\theta(t))^{\frac{1}{3}} \sum_{\mathbf{y}\in\mathcal{Y}} \pi_{\theta(t)}(\mathbf{y}|\mathbf{x}) \left[ r^{\mathrm{KL}}_{\mathrm{RM}}(\mathbf{x},\mathbf{y};\theta(t)) - \phi_{\mathrm{RLHF}}(\theta(t);\mathbf{x}) \right]^2$$

$$= 24L^2 J_T^2 \cdot \mathrm{Var}^{\mathrm{KL}}(\theta(t))^{\frac{1}{3}} \mathrm{Var}^{\mathrm{KL}}_{\mathbf{x}}(\theta(t)),$$

where the second transition is due to the Cauchy-Schwarz inequality and the third is by Equation (7) and the fact that $\left\| \mathbf{e}_{\mathbf{y}_l} - \pi_{\theta(t)}(\cdot|\mathbf{x},\mathbf{y}_{<l}) \right\| \leq \left\| \mathbf{e}_{\mathbf{y}_l} - \pi_{\theta(t)}(\cdot|\mathbf{x},\mathbf{y}_{<l}) \right\|_1 \leq 2$.

Moving on to the term $\mathbf{b}$, notice that it is equal to zero:

$$\mathbf{b} = 2\nabla\phi_{\mathrm{RLHF}}(\theta(t);\mathbf{x}) \cdot \sum_{\mathbf{y}\in\mathcal{Y}} \pi_{\theta(t)}(\mathbf{y}|\mathbf{x}) \left[ r^{\mathrm{KL}}_{\mathrm{RM}}(\mathbf{x},\mathbf{y};\theta(t)) - \phi_{\mathrm{RLHF}}(\theta(t);\mathbf{x}) \right]$$

$$= 2\nabla\phi_{\mathrm{RLHF}}(\theta(t);\mathbf{x}) \cdot \left[ \underbrace{\sum_{\mathbf{y}\in\mathcal{Y}} \pi_{\theta(t)}(\mathbf{y}|\mathbf{x}) r^{\mathrm{KL}}_{\mathrm{RM}}(\mathbf{x},\mathbf{y};\theta(t))}_{=\phi_{\mathrm{RLHF}}(\theta(t);\mathbf{x})} - \phi_{\mathrm{RLHF}}(\theta(t);\mathbf{x}) \cdot \underbrace{\sum_{\mathbf{y}\in\mathcal{Y}} \pi_{\theta(t)}(\mathbf{y}|\mathbf{x})}_{=1} \right]$$

$$= 0,$$

and so $\langle \mathbf{b}, \nabla \phi_{\mathrm{RLHF}}(\theta(t)) \rangle = 0$. Lastly:

$$\mathbf{c} = 2\lambda \nabla \phi_{\mathrm{RLHF}}(\theta(t); \mathbf{x}) - 2\lambda \phi_{\mathrm{RLHF}}(\theta(t); \mathbf{x}) \cdot \underbrace{\sum_{\mathbf{y} \in \mathcal{Y}} \pi_{\theta(t)}(\mathbf{y}|\mathbf{x}) \cdot \nabla \ln \pi_{\theta(t)}(\mathbf{y}|\mathbf{x})}_{=\sum_{\mathbf{y} \in \mathcal{Y}} \nabla \pi_{\theta(t)}(\mathbf{y}|\mathbf{x}) = \nabla \sum_{\mathbf{y} \in \mathcal{Y}} \pi_{\theta(t)}(\mathbf{y}|\mathbf{x}) = 0}$$

$$= 2\lambda \nabla \phi_{\mathrm{RLHF}}(\theta(t); \mathbf{x}).$$

Thus, going back to Equation (9) we obtain:

$$\frac{d}{dt} \mathrm{Var}^{\mathrm{KL}}(\theta(t))$$

$$= \underset{\mathbf{x} \sim \mathcal{S}}{\mathbb{E}}[\langle \mathbf{a}, \nabla \phi_{\mathrm{RLHF}}(\theta(t)) \rangle] - \underset{\mathbf{x} \sim \mathcal{S}}{\mathbb{E}}[\langle \mathbf{b}, \nabla \phi_{\mathrm{RLHF}}(\theta(t)) \rangle] - \underset{\mathbf{x} \sim \mathcal{S}}{\mathbb{E}}[\langle \mathbf{c}, \nabla \phi_{\mathrm{RLHF}}(\theta(t)) \rangle]$$

$$= \underset{\mathbf{x} \sim \mathcal{S}}{\mathbb{E}}[\langle \mathbf{a}, \nabla \phi_{\mathrm{RLHF}}(\theta(t)) \rangle] - 0 - 2\lambda \|\nabla \phi_{\mathrm{RLHF}}(\theta(t))\|^2$$

$$\leq 24 L^2 J_T^2 \cdot \mathrm{Var}^{\mathrm{KL}}(\theta(t))^{\frac{1}{3}} \underset{\mathbf{x} \sim \mathcal{S}}{\mathbb{E}}[\mathrm{Var}_{\mathbf{x}}^{\mathrm{KL}}(\theta(t))] - 2\lambda \|\nabla \phi_{\mathrm{RLHF}}(\theta(t))\|^2$$

$$= 24 L^2 J_T^2 \cdot \mathrm{Var}^{\mathrm{KL}}(\theta(t))^{\frac{4}{3}} - 2\lambda \|\nabla \phi_{\mathrm{RLHF}}(\theta(t))\|^2$$

$$\leq 24 L^2 J_T^2 \cdot \mathrm{Var}^{\mathrm{KL}}(\theta(t))^{\frac{4}{3}}.$$

By Lemma 1, this implies that for all $t < \frac{1}{8} L^{-2} J_T^{-2} \mathrm{Var}^{\mathrm{KL}}(\theta(0))^{-\frac{1}{3}}$:

$$\mathrm{Var}^{\mathrm{KL}}(\theta(t)) \leq \frac{\mathrm{Var}^{\mathrm{KL}}(\theta(0))}{\left(1 - 8 L^2 J_T^2 \mathrm{Var}^{\mathrm{KL}}(\theta(0))^{\frac{1}{3}} \cdot t\right)^3}.$$

Plugging this into Equation (8) yields:

$$\|\theta(T) - \theta(0)\|$$

$$\leq \int_0^T \left\| \frac{d}{dt} \theta(t) \right\| dt$$

$$\leq 12 L J_T \int_0^T \mathrm{Var}^{\mathrm{KL}}(\theta(t))^{\frac{1}{3}} dt$$

$$\leq 12 L J_T \int_0^T \frac{\mathrm{Var}^{\mathrm{KL}}(\theta(0))^{\frac{1}{3}}}{1 - 8 L^2 J_T^2 \mathrm{Var}^{\mathrm{KL}}(\theta(0))^{\frac{1}{3}} \cdot t} dt$$

$$= 12 L J_T \mathrm{Var}^{\mathrm{KL}}(\theta(0))^{\frac{1}{3}} \left( -\frac{1}{8 L^2 J_T^2 \mathrm{Var}^{\mathrm{KL}}(\theta(0))^{\frac{1}{3}}} \cdot \ln\left(1 - 8 L^2 J_T^2 \mathrm{Var}^{\mathrm{KL}}(\theta(0))^{\frac{1}{3}} \cdot t\right) \right) \Big|_{t=0}^T$$

$$= \frac{3}{2 L J_T} \ln\left( \frac{1}{1 - 8 L^2 J_T^2 \mathrm{Var}^{\mathrm{KL}}(\theta(0))^{\frac{1}{3}} \cdot T} \right).$$

The proof concludes by noticing that:

$$\mathrm{Var}^{\mathrm{KL}}(\theta(0)) = \mathbb{E}_{\mathbf{x} \sim \mathcal{S}}\left[ \mathrm{Var}_{\mathbf{y} \sim \pi_{\theta(0)}(\cdot|\mathbf{x})}\left[ r_{\mathrm{RM}}^{\mathrm{KL}}(\mathbf{x}, \mathbf{y}; \theta(0)) \right] \right] = \mathbb{E}_{\mathbf{x} \sim \mathcal{S}}\left[ \mathrm{Var}_{\mathbf{y} \sim \pi_{\theta(0)}(\cdot|\mathbf{x})}[r_{\mathrm{RM}}(\mathbf{x}, \mathbf{y})] \right],$$

since $r_{\mathrm{RM}}^{\mathrm{KL}}(\mathbf{x}, \mathbf{y}; \theta(0)) = r_{\mathrm{RM}}(\mathbf{x}, \mathbf{y}) - \lambda \cdot \ln \frac{\pi_{\theta(0)}(\mathbf{y}|\mathbf{x})}{\pi_{\theta(0)}(\mathbf{y}|\mathbf{x})} = r_{\mathrm{RM}}(\mathbf{x}, \mathbf{y})$ for all $\mathbf{x} \in \mathcal{S}, \mathbf{y} \in \mathcal{Y}$. $\qquad \square$

Lemma 5 relies on an assumption that the KL-regularized reward $r_{\mathrm{RM}}^{\mathrm{KL}}(\mathbf{x}, \mathbf{y}; \theta(t))$ remains bounded, for all $\mathbf{x} \in \mathcal{S}$ and $\mathbf{y} \in \mathcal{Y}$. In general, however, the KL-regularized reward can be unbounded if $\pi_{\theta(t)}(\mathbf{y}|\mathbf{x})$ goes to zero. Fortunately, by tracking the evolution of the KL-regularization term in $r_{\mathrm{RM}}^{\mathrm{KL}}(\mathbf{x}, \mathbf{y}; \theta(t))$, *i.e.*, of $\lambda \cdot \ln(\pi_{\theta(t)}(\mathbf{y}|\mathbf{x})/\pi_{\theta_{\mathrm{ref}}}(\mathbf{y}|\mathbf{x}))$, it is possible to prove that it remains bounded for a sufficiently long time.

**Lemma 6.** *Under the setting of Theorem 4, let*

$$T \in \left[ 0, \frac{1 - \exp\left(-\frac{7}{3\lambda}\right)}{8 L^2 J_T^2 \mathbb{E}_{\mathbf{x} \sim \mathcal{S}}[\mathrm{Var}_{\mathbf{y} \sim \pi_{\theta(0)}(\cdot|\mathbf{x})}(r_{\mathrm{RM}}(\mathbf{x}, \mathbf{y}))]^{1/3}} \right),$$

where $J_T := \sup_{\mathbf{x} \in \mathcal{X}, \mathbf{y} \in \mathcal{Y}, l \in [L], t \in [0,T]} \left\| \mathbf{J}_{f_{\theta(t)}(\mathbf{x}, \mathbf{y}_{<l})} \right\|_2$ and, by convention, $\exp\left(-\frac{7}{3\lambda}\right) = 0$ if $\lambda = 0$. Then, for all $\mathbf{x} \in \mathcal{S}$, $\mathbf{y} \in \mathcal{Y}$, and $t \in [0,T]$, it holds that $r_{\mathrm{RM}}^{\mathrm{KL}}(\mathbf{x}, \mathbf{y}; \theta(t)) \in [-8, 8]$.

*Proof.* If $\lambda = 0$, the claim is trivial since $r_{\mathrm{RM}}^{\mathrm{KL}}(\mathbf{x}', \mathbf{y}'; \theta(t)) = r_{\mathrm{RM}}(\mathbf{x}', \mathbf{y}') \in [-1, 1]$ for all $\mathbf{x}' \in \mathcal{S}$, $\mathbf{y}' \in \mathcal{Y}$, and $t \geq 0$.

Otherwise, if $\lambda > 0$, assume by way of contradiction that there exist $\mathbf{x} \in \mathcal{S}, \mathbf{y} \in \mathcal{Y}$, and $t \in [0, T]$ for which $r_{\mathrm{RM}}^{\mathrm{KL}}(\mathbf{x}, \mathbf{y}; \theta(t)) \notin [-8, 8]$. Since $r_{\mathrm{RM}}^{\mathrm{KL}}(\mathbf{x}, \mathbf{y}; \theta(t))$ is continuous in $t$ and initially $r_{\mathrm{RM}}^{\mathrm{KL}}(\mathbf{x}', \mathbf{y}'; \theta(0)) = r_{\mathrm{RM}}(\mathbf{x}', \mathbf{y}') \in [-1, 1]$ for all $\mathbf{x}' \in \mathcal{S}$ and $\mathbf{y}' \in \mathcal{Y}$, the KL-regularized reward $r_{\mathrm{RM}}^{\mathrm{KL}}(\mathbf{x}, \mathbf{y}; \theta(t))$ must be equal to either $-8$ or $8$ before escaping the interval $[-8, 8]$. We denote by $t_\lambda$ the initial time at which this occurs for some $\mathbf{x} \in \mathcal{S}$ and $\mathbf{y} \in \mathcal{Y}$, *i.e.*, $t_\lambda$ is the initial time at which $|r_{\mathrm{RM}}^{\mathrm{KL}}(\mathbf{x}, \mathbf{y}; \theta(t))| = 8$. By the definition of $t_\lambda$ it holds that: *(i)* $t_\lambda < T$ and *(ii)* $r_{\mathrm{RM}}^{\mathrm{KL}}(\mathbf{x}', \mathbf{y}'; \theta(t)) \in [-8, 8]$ for all $\mathbf{x}' \in \mathcal{S}$, $\mathbf{y}' \in \mathcal{Y}$, and $t \leq t_\lambda$.

Now, we bound how far the KL-regularized reward for $\mathbf{x}, \mathbf{y}$ deviates from its initial value as follows:

$$
\begin{aligned}
\left| r_{\mathrm{RM}}^{\mathrm{KL}}(\mathbf{x}, \mathbf{y}; \theta(t_\lambda)) - r_{\mathrm{RM}}^{\mathrm{KL}}(\mathbf{x}, \mathbf{y}; \theta(0)) \right| &= \left| \lambda \cdot \ln \frac{\pi_{\theta(t_\lambda)}(\mathbf{y}|\mathbf{x})}{\pi_{\theta_{\mathrm{ref}}}(\mathbf{y}|\mathbf{x})} - \lambda \cdot \ln \frac{\pi_{\theta(0)}(\mathbf{y}|\mathbf{x})}{\pi_{\theta_{\mathrm{ref}}}(\mathbf{y}|\mathbf{x})} \right| \\
&= \lambda \cdot \left| \ln \pi_{\theta(t_\lambda)}(\mathbf{y}|\mathbf{x}) - \ln \pi_{\theta(0)}(\mathbf{y}|\mathbf{x}) \right| \\
&= \lambda \cdot \left| \int_0^{t_\lambda} \tfrac{d}{dt} \ln \pi_{\theta(t)}(\mathbf{y}|\mathbf{x}) dt \right| \\
&\leq \lambda \cdot \int_0^{t_\lambda} \left| \left\langle \nabla \ln \pi_{\theta(t)}(\mathbf{y}|\mathbf{x}), \tfrac{d}{dt}\theta(t) \right\rangle \right| dt \\
&\leq \lambda \cdot \int_0^{t_\lambda} \left\| \nabla \ln \pi_{\theta(t)}(\mathbf{y}|\mathbf{x}) \right\| \left\| \tfrac{d}{dt}\theta(t) \right\| dt \,,
\end{aligned}
$$

where the penultimate transition is by the triangle inequality and the chain rule and the final transition is by the Cauchy-Schwarz inequality. Note that $\|\nabla \ln \pi_{\theta(t)}(\mathbf{y}|\mathbf{x})\| \leq 2LJ_T$ because:

$$
\begin{aligned}
\left\| \nabla \ln \pi_{\theta(t)}(\mathbf{y}|\mathbf{x}) \right\| &= \left\| \nabla \sum_{l=1}^{L} \ln \pi_{\theta(t)}(\mathbf{y}_l|\mathbf{x}, \mathbf{y}_{<l}) \right\| \\
&= \left\| \sum_{l=1}^{L} \mathbf{J}_{f_\theta(\mathbf{x}, \mathbf{y}_{<l})}^\top (\mathbf{e}_{\mathbf{y}_l} - \pi_{\theta(t)}(\cdot|\mathbf{x}, \mathbf{y}_{<l})) \right\| \\
&\leq J_T \sum_{l=1}^{L} \left\| \mathbf{e}_{\mathbf{y}_l} - \pi_{\theta(t)}(\cdot|\mathbf{x}, \mathbf{y}_{<l}) \right\| \\
&\leq J_T \sum_{l=1}^{L} \left\| \mathbf{e}_{\mathbf{y}_l} - \pi_{\theta(t)}(\cdot|\mathbf{x}, \mathbf{y}_{<l}) \right\|_1 \\
&\leq 2LJ_T \,,
\end{aligned}
$$

where, with slight abuse of notation, $\pi_{\theta(t)}(\cdot|\mathbf{x}, \mathbf{y}_{<l}) = \mathrm{softmax}(f_{\theta(t)}(\mathbf{x}, \mathbf{y}_{<l}))$. Therefore, invoking Lemma 5 with $T$ therein being set to $t_\lambda$ yields:

$$
\begin{aligned}
&\left| r_{\mathrm{RM}}^{\mathrm{KL}}(\mathbf{x}, \mathbf{y}; \theta(t_\lambda)) - r_{\mathrm{RM}}^{\mathrm{KL}}(\mathbf{x}, \mathbf{y}; \theta(0)) \right| \\
&\leq \lambda \cdot \int_0^{t_\lambda} \left\| \nabla \ln \pi_{\theta(t)}(\mathbf{y}|\mathbf{x}) \right\| \left\| \tfrac{d}{dt}\theta(t) \right\| dt \\
&\leq 2LJ_T\lambda \cdot \int_0^{t_\lambda} \left\| \tfrac{d}{dt}\theta(t) \right\| dt \\
&\leq 3\lambda \cdot \ln \left( \frac{1}{1 - 8L^2 J_T^2 \, \mathbb{E}_{\mathbf{x}' \sim \mathcal{S}} \left[ \mathrm{Var}_{\mathbf{y}' \sim \pi_{\theta(0)}(\cdot|\mathbf{x}')} [r_{\mathrm{RM}}(\mathbf{x}', \mathbf{y}')] \right]^{1/3} \cdot t_\lambda} \right) \,.
\end{aligned}
$$

Since

$$
t_\lambda < T < \frac{1 - \exp\left(-\frac{7}{3\lambda}\right)}{8L^2 J_T^2 \, \mathbb{E}_{\mathbf{x}' \sim \mathcal{S}} [\mathrm{Var}_{\mathbf{y}' \sim \pi_{\theta(0)}(\cdot|\mathbf{x}')} (r_{\mathrm{RM}}(\mathbf{x}', \mathbf{y}'))]^{1/3}} \,,
$$

we have that

$$\left| r_{\mathrm{RM}}^{\mathrm{KL}}(\mathbf{x}, \mathbf{y}; \theta(t_\lambda)) - r_{\mathrm{RM}}^{\mathrm{KL}}(\mathbf{x}, \mathbf{y}; \theta(0)) \right| < 7\,.$$

Thus, $|r_{\mathrm{RM}}^{\mathrm{KL}}(\mathbf{x}, \mathbf{y}; \theta(t_\lambda))| \leq |r_{\mathrm{RM}}^{\mathrm{KL}}(\mathbf{x}, \mathbf{y}; \theta(t_\lambda)) - r_{\mathrm{RM}}^{\mathrm{KL}}(\mathbf{x}, \mathbf{y}; \theta(0))| + |r_{\mathrm{RM}}^{\mathrm{KL}}(\mathbf{x}, \mathbf{y}; \theta(0))| < 8$, in contradiction to our assumption that $|r_{\mathrm{RM}}^{\mathrm{KL}}(\mathbf{x}, \mathbf{y}; \theta(t_\lambda))| = 8$. $\qquad\square$

With Lemmas 5 and 6 in place, we are now in a position to complete the proof of Theorem 4. To simplify notation, we denote the expected reward that $\pi_\theta$ achieves with respect to $r$ and the prompt $\mathbf{x} \in \mathcal{X}$ by $V(\theta; \mathbf{x}) := \mathbb{E}_{\mathbf{y} \sim \pi_\theta(\cdot|\mathbf{x})}[r(\mathbf{x}, \mathbf{y})]$.

Notice that if

$$t_\gamma \geq \frac{1 - \exp\left(-\frac{7}{3\lambda}\right)}{8 L^2 J_{t_\gamma}^2 \, \mathbb{E}_{\mathbf{x}' \sim \mathcal{S}}[\mathrm{Var}_{\mathbf{y} \sim \pi_{\theta(0)}(\cdot|\mathbf{x}')}(r_{\mathrm{RM}}(\mathbf{x}', \mathbf{y}))]^{1/3}}\,,$$

then we are done. Otherwise, Lemma 6 implies that $r_{\mathrm{RM}}^{\mathrm{KL}}(\mathbf{x}', \mathbf{y}'; \theta(t)) \in [-8, 8]$ for all $\mathbf{x}' \in \mathcal{S}, \mathbf{y}' \in \mathcal{Y}$, and time $t \in [0, t_\gamma]$. By the fundamental theorem of calculus, the triangle inequality, and the Cauchy-Schwarz inequality:

$$
\begin{aligned}
|V(\theta(t_\gamma); \mathbf{x}) - V(\theta(0); \mathbf{x})| &= \left| \int_0^{t_\gamma} \tfrac{d}{dt} V(\theta(t); \mathbf{x}) dt \right| \\
&\leq \int_0^{t_\gamma} \left| \tfrac{d}{dt} V(\theta(t); \mathbf{x}) \right| dt \\
&= \int_0^{t_\gamma} \left| \langle \nabla V(\theta(t); \mathbf{x}), \tfrac{d}{dt}\theta(t) \rangle \right| dt \\
&\leq \int_0^{t_\gamma} \left\| \nabla V(\theta(t); \mathbf{x}) \right\| \left\| \tfrac{d}{dt}\theta(t) \right\| dt\,.
\end{aligned}
$$

Applying Proposition 2, we can upper bound $\|\nabla V(\theta(t); \mathbf{x})\|$ as follows:

$$
\begin{aligned}
\|\nabla V(\theta(t); \mathbf{x})\| &\leq 6 L J_{t_\gamma} \cdot \mathrm{Var}_{\mathbf{y} \sim \pi_{\theta(t)}(\cdot|\mathbf{x})}[r_{\mathrm{RM}}(\mathbf{x}, \mathbf{y})]^{\frac{1}{3}} \\
&\leq 6 L J_{t_\gamma}\,,
\end{aligned}
$$

where the last inequality is due to the variance of a random variable bounded within $[-1, 1]$ being at most 1 (recall $r(\mathbf{x}, \mathbf{y}) \in [-1, 1]$ for all $\mathbf{y} \in \mathcal{Y}$). Thus, Lemma 5 yields (note that $t_\gamma$ indeed satisfies the requirement in Lemma 5):

$$
\begin{aligned}
|V(\theta(t_\gamma); \mathbf{x}) - V(\theta(0); \mathbf{x})| &\leq 6 L J_{t_\gamma} \int_0^{t_\gamma} \left\| \tfrac{d}{dt}\theta(t) \right\| dt \\
&\leq 9 \ln\left( \frac{1}{1 - 8 L^2 J_{t_\gamma}^2 \, \mathbb{E}_{\mathbf{x}' \sim \mathcal{S}}\left[ \mathrm{Var}_{\mathbf{y} \sim \pi_{\theta(0)}(\cdot|\mathbf{x}')}[r_{\mathrm{RM}}(\mathbf{x}', \mathbf{y})] \right]^{1/3} \cdot T} \right).
\end{aligned}
$$

For $V(\theta(t_\gamma); \mathbf{x}) - V(\theta(0); \mathbf{x}) \geq \gamma$ to hold, it therefore must be the case that:

$$\gamma \leq 9 \ln\left( \frac{1}{1 - 8 L^2 J_{t_\gamma}^2 \, \mathbb{E}_{\mathbf{x}' \sim \mathcal{S}}\left[ \mathrm{Var}_{\mathbf{y} \sim \pi_{\theta(0)}(\cdot|\mathbf{x}')}[r_{\mathrm{RM}}(\mathbf{x}', \mathbf{y})] \right]^{1/3} \cdot T} \right).$$

Rearranging the inequality concludes the proof:

$$t_\gamma \geq \frac{1 - \exp\left(-\frac{\gamma}{9}\right)}{8 L^2 J_{t_\gamma}^2} \cdot \frac{1}{\mathbb{E}_{\mathbf{x}' \sim \mathcal{S}}\left[ \mathrm{Var}_{\mathbf{y} \sim \pi_{\theta(0)}(\cdot|\mathbf{x}')}[r_{\mathrm{RM}}(\mathbf{x}', \mathbf{y})] \right]^{1/3}}\,.$$

$\qquad\square$

### D.5 Useful Lemmas: Tabular Policies

We establish preliminary lemmas toward analyzing the gradient flow trajectory for tabular policies (Equation (3)) in the proofs of Theorems 5 and 6. All policies in the lemmas below are parameterized according to Equation (3).

**Lemma 7.** *For any two distinct prompts* $\mathbf{x}, \mathbf{x}' \in \mathcal{X}$, *reward model* $r_{\mathrm{RM}} : \mathcal{X} \times \mathcal{Y} \to [-1, 1]$, *and policy* $\pi_\theta$:

$$\nabla_{\theta_{:,\mathbf{x}}} \phi_{\mathrm{RLHF}}(\theta; \mathbf{x}) = \pi_\theta(\cdot|\mathbf{x}) \odot \left[ r_{\mathrm{RM}}^{\mathrm{KL}}(\mathbf{x}, \cdot; \theta) - \mathbb{E}_{\mathbf{y} \sim \pi_\theta(\cdot|\mathbf{x})} \left[ r_{\mathrm{RM}}^{\mathrm{KL}}(\mathbf{x}, \mathbf{y}; \theta) \right] \cdot \mathbf{1} \right],$$

$$\nabla_{\theta_{:,\mathbf{x}'}} \phi_{\mathrm{RLHF}}(\theta; \mathbf{x}) = 0,$$

*where recall that* $\phi_{\mathrm{RLHF}}(\theta; \mathbf{x}) := \mathbb{E}_{\mathbf{y} \sim \pi_\theta(\cdot|\mathbf{x})} \left[ r_{\mathrm{RM}}^{\mathrm{KL}}(\mathbf{x}, \mathbf{y}; \theta) \right]$, *with* $r_{\mathrm{RM}}^{\mathrm{KL}}(\mathbf{x}, \mathbf{y}; \theta) := r_{\mathrm{RM}}(\mathbf{x}, \mathbf{y}) - \lambda \cdot \ln(\pi_\theta(\mathbf{y}|\mathbf{x})/\pi_{\theta_{\mathrm{ref}}}(\mathbf{y}|\mathbf{x}))$ *denoting the KL-regularized reward,* $\pi(\cdot|\mathbf{x}) = \mathrm{softmax}(\theta_{:,\mathbf{x}}) \in \mathbb{R}^{|\mathcal{Y}|}$ *is a vector holding the probabilities that* $\pi_\theta$ *assigns to all outputs,* $r_{\mathrm{RM}}^{\mathrm{KL}}(\mathbf{x}, \cdot; \theta) \in \mathbb{R}^{|\mathcal{Y}|}$ *is a vector holding the KL-regularized rewards of all outputs,* $\odot$ *denotes the Hadamard (element-wise) product, and* $\mathbf{1} \in \mathbb{R}^{|\mathcal{Y}|}$ *denotes the all-ones vector.*

*Proof.* First, $\nabla_{\theta_{:,\mathbf{x}'}} \phi_{\mathrm{RLHF}}(\theta; \mathbf{x}) = 0$ follows from the fact that, under a tabular policy, $\phi_{\mathrm{RLHF}}(\theta; \mathbf{x})$ does not depend on $\theta_{:,\mathbf{x}'}$. As for the gradient with respect to $\theta_{:,\mathbf{x}}$, by the log-derivative trick we have:

$$\nabla_{\theta_{:,\mathbf{x}}} \phi_{\mathrm{RLHF}}(\theta; \mathbf{x}) = \nabla_{\theta_{:,\mathbf{x}}} \mathbb{E}_{\mathbf{y} \sim \pi_\theta(\cdot|\mathbf{x})} \left[ r_{\mathrm{RM}}(\mathbf{x}, \mathbf{y}) - \lambda \cdot \ln \frac{\pi_\theta(\mathbf{y}|\mathbf{x})}{\pi_{\theta_{\mathrm{ref}}}(\mathbf{y}|\mathbf{x})} \right]$$

$$= \mathbb{E}_{\mathbf{y} \sim \pi_\theta(\cdot|\mathbf{x})} \left[ r_{\mathrm{RM}}^{\mathrm{KL}}(\mathbf{x}, \mathbf{y}) \nabla_{\theta_{:,\mathbf{x}}} \ln \pi_\theta(\mathbf{y}|\mathbf{x}) \right] - \lambda \, \mathbb{E}_{\mathbf{y} \sim \pi_\theta(\cdot|\mathbf{x})} \left[ \nabla_{\theta_{:,\mathbf{x}}} \ln \pi_\theta(\mathbf{y}|\mathbf{x}) \right].$$

Since

$$\mathbb{E}_{\mathbf{y} \sim \pi_\theta(\cdot|\mathbf{x})} \left[ \nabla_{\theta_{:,\mathbf{x}}} \ln \pi_\theta(\mathbf{y}|\mathbf{x}) \right] = \sum\nolimits_{\mathbf{y} \in \mathcal{Y}} \nabla_{\theta_{:,\mathbf{x}}} \pi_\theta(\mathbf{y}|\mathbf{x}) = \nabla_{\theta_{:,\mathbf{x}}} \sum\nolimits_{\mathbf{y} \in \mathcal{Y}} \pi_\theta(\mathbf{y}|\mathbf{x}) = \nabla 1 = 0,$$

we may write:

$$\nabla_{\theta_{:,\mathbf{x}}} \phi_{\mathrm{RLHF}}(\theta; \mathbf{x}) = \mathbb{E}_{\mathbf{y} \sim \pi_\theta(\cdot|\mathbf{x})} \left[ r_{\mathrm{RM}}^{\mathrm{KL}}(\mathbf{x}, \mathbf{y}; \theta) \nabla_{\theta_{:,\mathbf{x}}} \ln \pi_\theta(\mathbf{y}|\mathbf{x}) \right]$$

$$= \sum\nolimits_{\mathbf{y} \in \mathcal{Y}} \pi_\theta(\mathbf{y}|\mathbf{x}) r_{\mathrm{RM}}^{\mathrm{KL}}(\mathbf{x}, \mathbf{y}; \theta) \cdot \nabla_{\theta_{:,\mathbf{x}}} \ln \pi_\theta(\mathbf{y}|\mathbf{x})$$

$$= \sum\nolimits_{\mathbf{y} \in \mathcal{Y}} \pi_\theta(\mathbf{y}|\mathbf{x}) r_{\mathrm{RM}}^{\mathrm{KL}}(\mathbf{x}, \mathbf{y}; \theta) \cdot \nabla_{\theta_{:,\mathbf{x}}} \left( \theta_{\mathbf{y},\mathbf{x}} - \ln \sum\nolimits_{\mathbf{y}' \in \mathcal{Y}} \exp(\theta_{\mathbf{y}',\mathbf{x}}) \right)$$

$$= \sum\nolimits_{\mathbf{y} \in \mathcal{Y}} \pi_\theta(\mathbf{y}|\mathbf{x}) r_{\mathrm{RM}}^{\mathrm{KL}}(\mathbf{x}, \mathbf{y}; \theta) \cdot (\mathbf{e}_\mathbf{y} - \pi_\theta(\cdot|\mathbf{x})).$$

Noticing that

$$\sum\nolimits_{\mathbf{y} \in \mathcal{Y}} \pi_\theta(\mathbf{y}|\mathbf{x}) r_{\mathrm{RM}}^{\mathrm{KL}}(\mathbf{x}, \mathbf{y}; \theta) \cdot \mathbf{e}_\mathbf{y} = \pi_\theta(\cdot|\mathbf{x}) \odot r_{\mathrm{RM}}^{\mathrm{KL}}(\mathbf{x}, \cdot; \theta)$$

and

$$\sum\nolimits_{\mathbf{y} \in \mathcal{Y}} \pi_\theta(\mathbf{y}|\mathbf{x}) r_{\mathrm{RM}}^{\mathrm{KL}}(\mathbf{x}, \mathbf{y}; \theta) \cdot \pi_\theta(\cdot|\mathbf{x}) = \pi_\theta(\cdot|\mathbf{x}) \odot \left( \mathbb{E}_{\mathbf{y} \sim \pi_\theta(\cdot|\mathbf{x})} \left[ r_{\mathrm{RM}}^{\mathrm{KL}}(\mathbf{x}, \mathbf{y}; \theta) \right] \cdot \mathbf{1} \right)$$

concludes the proof. $\qquad\qquad\square$

**Lemma 8.** *For any prompt* $\mathbf{x} \in \mathcal{X}$, *reward model* $r_{\mathrm{RM}} : \mathcal{X} \times \mathcal{Y} \to [-1, 1]$, *and policy* $\pi_\theta$, *it holds that:*

$$\left\| \nabla_{\theta_{:,\mathbf{x}}} \phi_{\mathrm{RLHF}}(\theta; \mathbf{x}) \right\|_1 = \left\| \nabla_{\theta_{:,\mathbf{x}}} \mathbb{E}_{\mathbf{y} \sim \pi_\theta(\cdot|\mathbf{x})} \left[ r_{\mathrm{RM}}^{\mathrm{KL}}(\mathbf{x}, \mathbf{y}; \theta) \right] \right\|_1 \leq \sqrt{\mathrm{Var}_{\mathbf{y} \sim \pi_\theta(\cdot|\mathbf{x})} \left[ r_{\mathrm{RM}}^{\mathrm{KL}}(\mathbf{x}, \mathbf{y}; \theta) \right]},$$

*where* $\|\cdot\|_1$ *denotes the* $\ell_1$ *norm.*

*Proof.* By Lemma 7:

$$\left\| \nabla_{\theta_{:,\mathbf{x}}} \phi_{\mathrm{RLHF}}(\theta; \mathbf{x}) \right\|_1 = \left\| \pi_\theta(\cdot|\mathbf{x}) \odot \left[ r_{\mathrm{RM}}^{\mathrm{KL}}(\mathbf{x}, \cdot; \theta) - \mathbb{E}_{\mathbf{y}' \sim \pi_\theta(\cdot|\mathbf{x})} \left[ r_{\mathrm{RM}}^{\mathrm{KL}}(\mathbf{x}, \mathbf{y}'; \theta) \right] \cdot \mathbf{1} \right] \right\|_1$$

$$= \sum\nolimits_{\mathbf{y} \in \mathcal{Y}} \pi_\theta(\mathbf{y}|\mathbf{x}) \cdot \left| r_{\mathrm{RM}}^{\mathrm{KL}}(\mathbf{x}, \mathbf{y}; \theta) - \mathbb{E}_{\mathbf{y}' \sim \pi_\theta(\cdot|\mathbf{x})} \left[ r_{\mathrm{RM}}^{\mathrm{KL}}(\mathbf{x}, \mathbf{y}'; \theta) \right] \right|$$

$$= \sum\nolimits_{\mathbf{y} \in \mathcal{Y}} \pi_\theta(\mathbf{y}|\mathbf{x}) \cdot \sqrt{\left( r_{\mathrm{RM}}^{\mathrm{KL}}(\mathbf{x}, \mathbf{y}; \theta) - \mathbb{E}_{\mathbf{y}' \sim \pi_\theta(\cdot|\mathbf{x})} \left[ r_{\mathrm{RM}}^{\mathrm{KL}}(\mathbf{x}, \mathbf{y}'; \theta) \right] \right)^2}.$$

Jensen's inequality then concludes the proof:

$$\left\|\nabla_{\theta_{:,\mathbf{x}}}\phi_{\mathrm{RLHF}}(\theta;\mathbf{x})\right\|_1 \leq \sqrt{\sum\nolimits_{\mathbf{y}\in\mathcal{Y}}\pi_\theta(\mathbf{y}|\mathbf{x})\cdot\left(r_{\mathrm{RM}}^{\mathrm{KL}}(\mathbf{x},\mathbf{y};\theta)-\mathbb{E}_{\mathbf{y}'\sim\pi_\theta(\cdot|\mathbf{x})}\left[r_{\mathrm{RM}}^{\mathrm{KL}}(\mathbf{x},\mathbf{y}';\theta)\right]\right)^2}$$

$$= \sqrt{\mathrm{Var}_{\mathbf{y}\sim\pi_\theta(\cdot|\mathbf{x})}\left[r_{\mathrm{RM}}^{\mathrm{KL}}(\mathbf{x},\mathbf{y};\theta)\right]}.$$

$\square$

**Lemma 9.** *Suppose gradient flow, over a tabular policy (Equation* (3)*), is used to maximize the RLHF objective over a set of prompts* $\mathcal{S}$ *(Equations* (1) *and* (4)*). For any prompt* $\mathbf{x}\in\mathcal{S}$*, output* $\mathbf{y}\in\mathcal{Y}$*, and time* $t\geq 0$ *it holds that:*

$$\frac{d}{dt}\pi_{\theta(t)}(\mathbf{y}|\mathbf{x}) = \frac{1}{|\mathcal{S}|}\cdot\pi_{\theta(t)}(\mathbf{y}|\mathbf{x})^2\left[r_{\mathrm{RM}}(\mathbf{x},\mathbf{y})-V_{\mathrm{RM}}(\theta(t);\mathbf{x})\right]$$

$$-\frac{1}{|\mathcal{S}|}\cdot\pi_{\theta(t)}(\mathbf{y}|\mathbf{x})\sum\nolimits_{\mathbf{y}'\in\mathcal{Y}}\pi_{\theta(t)}(\mathbf{y}'|\mathbf{x})^2\left[r_{\mathrm{RM}}(\mathbf{x},\mathbf{y}')-V_{\mathrm{RM}}(\theta(t);\mathbf{x})\right]$$

$$-\frac{\lambda}{|\mathcal{S}|}\cdot\pi_{\theta(t)}(\mathbf{y}|\mathbf{x})^2\left[\ln\frac{\pi_{\theta(t)}(\mathbf{y}|\mathbf{x})}{\pi_{\theta(0)}(\mathbf{y}|\mathbf{x})}-\mathrm{KL}(\theta(t))\right]$$

$$+\frac{\lambda}{|\mathcal{S}|}\cdot\pi_{\theta(t)}(\mathbf{y}|\mathbf{x})\sum\nolimits_{\mathbf{y}'\in\mathcal{Y}}\pi_{\theta(t)}(\mathbf{y}'|\mathbf{x})^2\left[\ln\frac{\pi_{\theta(t)}(\mathbf{y}'|\mathbf{x})}{\pi_{\theta(0)}(\mathbf{y}'|\mathbf{x})}-\mathrm{KL}(\theta(t))\right],$$

*where* $V_{\mathrm{RM}}(\theta(t);\mathbf{x}):=\mathbb{E}_{\mathbf{y}'\sim\pi_{\theta(t)}(\cdot|\mathbf{x})}[r_{\mathrm{RM}}(\mathbf{x},\mathbf{y}')]$ *and* $\mathrm{KL}(\theta(t)):=\mathrm{KL}(\pi_{\theta(t)}(\cdot|\mathbf{x})||\pi_{\theta(0)}(\cdot|\mathbf{x}))$.

*Proof.* Noticing that $\frac{d}{dt}\pi_{\theta(t)}(\mathbf{y}|\mathbf{x})=\pi_{\theta(t)}(\mathbf{y}|\mathbf{x})\cdot\frac{d}{dt}\ln\pi_{\theta(t)}(\mathbf{y}|\mathbf{x})$, by the chain rule and the fact that only the parameters $\theta_{:,\mathbf{x}}(t)$ affect $\ln\pi_{\theta(t)}(\mathbf{y}|\mathbf{x})$, we get:

$$\frac{d}{dt}\pi_{\theta(t)}(\mathbf{y}|\mathbf{x}) = \pi_{\theta(t)}(\mathbf{y}|\mathbf{x})\cdot\left\langle\nabla_{\theta_{:,\mathbf{x}}}\ln\pi_{\theta(t)}(\mathbf{y}|\mathbf{x}),\tfrac{d}{dt}\theta_{:,\mathbf{x}}(t)\right\rangle$$

$$= \pi_{\theta(t)}(\mathbf{y}|\mathbf{x})\cdot\left\langle\mathbf{e_y}-\pi_{\theta(t)}(\cdot|\mathbf{x}),\nabla_{\theta_{:,\mathbf{x}}}\phi_{\mathrm{RLHF}}(\theta(t))\right\rangle$$

$$= \pi_{\theta(t)}(\mathbf{y}|\mathbf{x})\cdot\left\langle\mathbf{e_y}-\pi_{\theta(t)}(\cdot|\mathbf{x}),\frac{1}{|\mathcal{S}|}\nabla_{\theta_{:,\mathbf{x}}}\phi_{\mathrm{RLHF}}(\theta(t);\mathbf{x})\right\rangle.$$

Plugging in the expression for $\nabla_{\theta_{:,\mathbf{x}}}\phi_{\mathrm{RLHF}}(\theta(t);\mathbf{x})$ from Lemma 7 then yields:

$$\frac{d}{dt}\pi_{\theta(t)}(\mathbf{y}|\mathbf{x}) = \frac{\pi_{\theta(t)}(\mathbf{y}|\mathbf{x})}{|\mathcal{S}|}\left\langle\mathbf{e_y}-\pi_{\theta(t)}(\cdot|\mathbf{x}),\pi_{\theta(t)}(\cdot|\mathbf{x})\odot[r_{\mathrm{RM}}(\mathbf{x},\cdot)-V_{\mathrm{RM}}(\theta(t);\mathbf{x})\cdot\mathbf{1}]\right\rangle$$

$$-\lambda\frac{\pi_{\theta(t)}(\mathbf{y}|\mathbf{x})}{|\mathcal{S}|}\left\langle\mathbf{e_y}-\pi_{\theta(t)}(\cdot|\mathbf{x}),\pi_{\theta(t)}(\cdot|\mathbf{x})\odot\left[\ln\frac{\pi_{\theta(t)}(\cdot|\mathbf{x})}{\pi_{\theta(0)}(\cdot|\mathbf{x})}-\mathrm{KL}(\theta(t))\cdot\mathbf{1}\right]\right\rangle,$$

from which the desired result readily follows by computing the inner products. $\square$

**Lemma 10.** *Suppose gradient flow, over a tabular policy (Equation* (3)*), is used to maximize the RLHF objective over a set of prompts* $\mathcal{S}$ *(Equations* (1) *and* (4)*). For any prompt* $\mathbf{x}\in\mathcal{S}$*, subset of outputs* $\mathcal{A}\subseteq\mathcal{Y}$*, and time* $T\geq 0$*, it holds that:*

$$\pi_{\theta(T)}(\mathcal{A}|\mathbf{x}) \leq \pi_{\theta(0)}(\mathcal{A}|\mathbf{x})\cdot\exp\left(\frac{1}{|\mathcal{S}|}\left(4+2\lambda\ln\frac{1}{\min_{\mathbf{y}\in\mathcal{Y}}\pi_{\theta(0)}(\mathbf{y}|\mathbf{x})}\right)\cdot T\right).$$

*Proof.* For any $\mathbf{y} \in \mathcal{A}$ and time $t \geq 0$, by Lemma 9:

$$\frac{d}{dt}\pi_{\theta(t)}(\mathbf{y}|\mathbf{x}) = \underbrace{\frac{1}{|\mathcal{S}|} \cdot \pi_{\theta(t)}(\mathbf{y}|\mathbf{x})^2\big[r_{\mathrm{RM}}(\mathbf{x}, \mathbf{y}) - V_{\mathrm{RM}}(\theta(t); \mathbf{x})\big]}_{(I)}$$

$$\underbrace{-\frac{1}{|\mathcal{S}|} \cdot \pi_{\theta(t)}(\mathbf{y}|\mathbf{x}) \sum\nolimits_{\mathbf{y}' \in \mathcal{Y}} \pi_{\theta(t)}(\mathbf{y}'|\mathbf{x})^2\big[r_{\mathrm{RM}}(\mathbf{x}, \mathbf{y}') - V_{\mathrm{RM}}(\theta(t); \mathbf{x})\big]}_{(II)}$$

$$\underbrace{-\frac{\lambda}{|\mathcal{S}|} \cdot \pi_{\theta(t)}(\mathbf{y}|\mathbf{x})^2\left[\ln\frac{\pi_{\theta(t)}(\mathbf{y}|\mathbf{x})}{\pi_{\theta(0)}(\mathbf{y}|\mathbf{x})} - \mathrm{KL}(\theta(t))\right]}_{(III)}$$

$$\underbrace{+\frac{\lambda}{|\mathcal{S}|} \cdot \pi_{\theta(t)}(\mathbf{y}|\mathbf{x}) \sum\nolimits_{\mathbf{y}' \in \mathcal{Y}} \pi_{\theta(t)}(\mathbf{y}'|\mathbf{x})^2\left[\ln\frac{\pi_{\theta(t)}(\mathbf{y}'|\mathbf{x})}{\pi_{\theta(0)}(\mathbf{y}'|\mathbf{x})} - \mathrm{KL}(\theta(t))\right]}_{(IV)},$$

where $V_{\mathrm{RM}}(\theta(t); \mathbf{x}) := \mathbb{E}_{\mathbf{y}' \sim \pi_{\theta(t)}(\cdot|\mathbf{x})}[r_{\mathrm{RM}}(\mathbf{x}, \mathbf{y}')]$ and $\mathrm{KL}(\theta(t)) := \mathrm{KL}(\pi_{\theta(t)}(\cdot|\mathbf{x})||\pi_{\theta(0)}(\cdot|\mathbf{x}))$.

We upper bound $(I) + (II)$ and $(III) + (IV)$ separately. Starting with $(I) + (II)$, since $r_{\mathrm{RM}}$ is bounded within $[-1, 1]$, for all $\mathbf{y}' \in \mathcal{Y}$ we have that:

$$|r_{\mathrm{RM}}(\mathbf{x}, \mathbf{y}') - V_{\mathrm{RM}}(\theta(t); \mathbf{x})| \leq 2\,.$$

Thus, $(I) + (II)$ can be bounded as follows:

$$(I) + (II) \leq \frac{2}{|\mathcal{S}|} \cdot \pi_{\theta(t)}(\mathbf{y}|\mathbf{x})^2 + \frac{2}{|\mathcal{S}|} \cdot \pi_{\theta(t)}(\mathbf{y}|\mathbf{x}) \leq \frac{4}{|\mathcal{S}|}\pi_{\theta(t)}(\mathbf{y}|\mathbf{x})\,.$$

As for $(III) + (IV)$, let $t_0 \in [0, T]$ be the latest time in $[0, T]$ at which $\pi_{\theta(t)}(\mathbf{y}|\mathbf{x}) \leq \pi_{\theta(0)}(\mathbf{y}|\mathbf{x})$. Formally:

$$t_0 := \max\{t \in [0, T] : \pi_{\theta(t)}(\mathbf{y}|\mathbf{x}) \leq \pi_{\theta(0)}(\mathbf{y}|\mathbf{x})\}\,.$$

Note that necessarily $\pi_{\theta(t_0)}(\mathbf{y}|\mathbf{x}) = \pi_{\theta(0)}(\mathbf{y}|\mathbf{x})$ since $\pi_{\theta(t)}(\mathbf{y}|\mathbf{x})$ is continuous in $t$. Now, for any time $t \geq t_0$ we have that $-\ln(\pi_{\theta(t)}(\mathbf{y}|\mathbf{x})/\pi_{\theta(0)}(\mathbf{y}|\mathbf{x})) \leq 0$. Along with the fact that:

$$0 \leq \mathrm{KL}(\theta(t)) \leq -\ln\pi_{\theta(0)}(\mathbf{y}^m|\mathbf{x})\,,$$

where $\mathbf{y}^m \in \arg\min_{\mathbf{y}' \in \mathcal{Y}} \pi_{\theta(0)}(\mathbf{y}'|\mathbf{x})$ (see Lemma 3 for the upper bound), we get that:

$$(III) + (IV) \leq \frac{\lambda}{|\mathcal{S}|}\pi_{\theta(t)}(\mathbf{y}|\mathbf{x})^2 \ln\frac{1}{\pi_{\theta(0)}(\mathbf{y}^m|\mathbf{x})} + \frac{\lambda}{|\mathcal{S}|}\pi_{\theta(t)}(\mathbf{y}|\mathbf{x}) \sum_{\mathbf{y}' \in \mathcal{Y}} \pi_{\theta(t)}(\mathbf{y}'|\mathbf{x})^2 \ln\frac{\pi_{\theta(t)}(\mathbf{y}'|\mathbf{x})}{\pi_{\theta(0)}(\mathbf{y}'|\mathbf{x})}\,.$$

Using the fact $\ln\frac{\pi_{\theta(t)}(\mathbf{y}'|\mathbf{x})}{\pi_{\theta(0)}(\mathbf{y}'|\mathbf{x})} \leq \ln\frac{1}{\pi_{\theta(0)}(\mathbf{y}'|\mathbf{x})} \leq \ln\frac{1}{\pi_{\theta(0)}(\mathbf{y}^m|\mathbf{x})}$ and $\pi_{\theta(t)}(\mathbf{y}'|\mathbf{x})^2 \leq \pi_{\theta(t)}(\mathbf{y}'|\mathbf{x})$ for all $\mathbf{y}' \in \mathcal{Y}$, we arrive at:

$$(III) + (IV) \leq \frac{\lambda}{|\mathcal{S}|}\pi_{\theta(t)}(\mathbf{y}|\mathbf{x})^2 \ln\frac{1}{\pi_{\theta(0)}(\mathbf{y}^m|\mathbf{x})} + \frac{\lambda}{|\mathcal{S}|}\pi_{\theta(t)}(\mathbf{y}|\mathbf{x}) \ln\frac{1}{\pi_{\theta(0)}(\mathbf{y}^m|\mathbf{x})}$$

$$\leq \frac{1}{|\mathcal{S}|}\left(2\lambda \ln\frac{1}{\pi_{\theta(0)}(\mathbf{y}^m|\mathbf{x})}\right)\pi_{\theta(t)}(\mathbf{y}|\mathbf{x})\,.$$

Combining our bounds on $(I) + (II)$ and $(III) + (IV)$, we obtain that at any time $t \in [t_0, T]$:

$$\frac{d}{dt}\pi_{\theta(t)}(\mathbf{y}|\mathbf{x}) \leq \frac{1}{|\mathcal{S}|}\left(4 + 2\lambda \ln\frac{1}{\pi_{\theta(0)}(\mathbf{y}^m|\mathbf{x})}\right) \cdot \pi_{\theta(t)}(\mathbf{y}|\mathbf{x})\,.$$

Lastly, applying Grönwall's inequality leads to:

$$\pi_{\theta(T)}(\mathbf{y}|\mathbf{x}) \leq \pi_{\theta(t_0)}(\mathbf{y}|\mathbf{x}) \cdot \exp\left(\frac{1}{|\mathcal{S}|}\left(4 + 2\lambda \ln\frac{1}{\pi_{\theta(0)}(\mathbf{y}^m|\mathbf{x})}\right) \cdot T\right)$$

$$= \pi_{\theta(0)}(\mathbf{y}|\mathbf{x}) \cdot \exp\left(\frac{1}{|\mathcal{S}|}\left(4 + 2\lambda \ln\frac{1}{\pi_{\theta(0)}(\mathbf{y}^m|\mathbf{x})}\right) \cdot T\right)\,.$$

Summing the inequality over all $\mathbf{y} \in \mathcal{A}$ completes the proof. $\qquad\square$

**Lemma 11.** *For any prompt* $\mathbf{x} \in \mathcal{X}$, *output* $\mathbf{y} \in \mathcal{Y}$, *policy* $\pi_\theta$, *and reference policy* $\pi_{\theta_{\mathrm{ref}}}$:

$$\pi_\theta(\mathbf{y}|\mathbf{x})^2 \left[\ln \frac{\pi_\theta(\mathbf{y}|\mathbf{x})}{\pi_{\theta_{\mathrm{ref}}}(\mathbf{y}|\mathbf{x})} - \mathrm{KL}(\theta)\right] - \pi_\theta(\mathbf{y}|\mathbf{x}) \sum_{\mathbf{y}' \in \mathcal{Y}} \pi_\theta(\mathbf{y}'|\mathbf{x})^2 \left[\ln \frac{\pi_\theta(\mathbf{y}'|\mathbf{x})}{\pi_{\theta_{\mathrm{ref}}}(\mathbf{y}'|\mathbf{x})} - \mathrm{KL}(\theta)\right]$$

$$\leq \pi_\theta(\mathbf{y}|\mathbf{x}) \left(2 \ln \frac{1}{\min_{\mathbf{y}' \in \mathcal{Y}} \pi_{\theta_{\mathrm{ref}}}(\mathbf{y}'|x)} + \frac{1}{2e}\right),$$

*where* $\mathrm{KL}(\theta) := \mathrm{KL}(\pi_\theta(\cdot|\mathbf{x})||\pi_{\theta_{\mathrm{ref}}}(\cdot|\mathbf{x}))$.

*Proof.* By Lemma 3 we have $0 \leq \mathrm{KL}(\theta) \leq - \ln \pi_{\theta_{\mathrm{ref}}}(\mathbf{y}^m|\mathbf{x})$, where $\mathbf{y}^m \in \mathrm{argmin}_{\mathbf{y}' \in \mathcal{Y}} \pi_{\theta_{\mathrm{ref}}}(\mathbf{y}'|\mathbf{x})$. Thus:

$$\pi_\theta(\mathbf{y}|\mathbf{x})^2 \left[\ln \frac{\pi_\theta(\mathbf{y}|\mathbf{x})}{\pi_{\theta_{\mathrm{ref}}}(\mathbf{y}|\mathbf{x})} - \mathrm{KL}(\theta)\right] - \pi_\theta(\mathbf{y}|\mathbf{x}) \sum_{\mathbf{y}' \in \mathcal{Y}} \pi_\theta(\mathbf{y}'|\mathbf{x})^2 \left[\ln \frac{\pi_\theta(\mathbf{y}'|\mathbf{x})}{\pi_{\theta_{\mathrm{ref}}}(\mathbf{y}'|\mathbf{x})} - \mathrm{KL}(\theta)\right]$$

$$\leq \pi_\theta(\mathbf{y}|\mathbf{x})^2 \ln \frac{\pi_\theta(\mathbf{y}|\mathbf{x})}{\pi_{\theta_{\mathrm{ref}}}(\mathbf{y}|\mathbf{x})} - \pi_\theta(\mathbf{y}|\mathbf{x}) \sum_{\mathbf{y}' \in \mathcal{Y}} \pi_\theta(\mathbf{y}'|\mathbf{x})^2 \left[\ln \frac{\pi_\theta(\mathbf{y}'|\mathbf{x})}{\pi_{\theta_{\mathrm{ref}}}(\mathbf{y}'|\mathbf{x})} - \ln \frac{1}{\pi_{\theta_{\mathrm{ref}}}(\mathbf{y}^m|\mathbf{x})}\right]$$

$$\leq \pi_\theta(\mathbf{y}|\mathbf{x})^2 \ln \frac{\pi_\theta(\mathbf{y}|\mathbf{x})}{\pi_{\theta_{\mathrm{ref}}}(\mathbf{y}|\mathbf{x})} + \pi_\theta(\mathbf{y}|\mathbf{x}) \ln \frac{1}{\pi_{\theta_{\mathrm{ref}}}(\mathbf{y}^m|\mathbf{x})} - \pi_\theta(\mathbf{y}|\mathbf{x}) \sum_{\mathbf{y}' \in \mathcal{Y}} \pi_\theta(\mathbf{y}'|\mathbf{x})^2 \ln \frac{\pi_\theta(\mathbf{y}'|\mathbf{x})}{\pi_{\theta_{\mathrm{ref}}}(\mathbf{y}'|\mathbf{x})}$$

$$\leq 2\pi_\theta(\mathbf{y}|\mathbf{x}) \ln \frac{1}{\pi_{\theta_{\mathrm{ref}}}(\mathbf{y}^m|\mathbf{x})} - \pi_\theta(\mathbf{y}|\mathbf{x}) \sum_{\mathbf{y}' \in \mathcal{Y}} \pi_\theta(\mathbf{y}'|\mathbf{x})^2 \ln \frac{\pi_\theta(\mathbf{y}'|\mathbf{x})}{\pi_{\theta_{\mathrm{ref}}}(\mathbf{y}'|\mathbf{x})},$$

where the penultimate transition is due to $\sum_{\mathbf{y}' \in \mathcal{Y}} \pi_\theta(\mathbf{y}'|\mathbf{x})^2 \leq 1$ and the last transition is by

$$\ln \frac{\pi_\theta(\mathbf{y}|\mathbf{x})}{\pi_{\theta_{\mathrm{ref}}}(\mathbf{y}|\mathbf{x})} \leq \ln \frac{1}{\pi_{\theta_{\mathrm{ref}}}(\mathbf{y}^m|\mathbf{x})}$$

and $\pi_\theta(\mathbf{y}|\mathbf{x})^2 \leq \pi_\theta(\mathbf{y}|\mathbf{x})$. Now, from Lemma 4 we know that:

$$-\pi_\theta(\mathbf{y}'|\mathbf{x})^2 \ln \frac{\pi_\theta(\mathbf{y}'|\mathbf{x})}{\pi_{\theta_{\mathrm{ref}}}(\mathbf{y}'|\mathbf{x})} \leq \frac{\pi_{\theta_{\mathrm{ref}}}(\mathbf{y}'|\mathbf{x})^2}{2e},$$

for all $\mathbf{y}' \in \mathcal{Y}$. Since $\sum_{\mathbf{y}' \in \mathcal{Y}} \frac{\pi_{\theta_{\mathrm{ref}}}(\mathbf{y}'|\mathbf{x})^2}{2e} \leq \sum_{\mathbf{y}' \in \mathcal{Y}} \frac{\pi_{\theta_{\mathrm{ref}}}(\mathbf{y}'|\mathbf{x})}{2e} = \frac{1}{2e}$, this implies that:

$$\pi_\theta(\mathbf{y}|\mathbf{x})^2 \left[\ln \frac{\pi_\theta(\mathbf{y}|\mathbf{x})}{\pi_{\theta_{\mathrm{ref}}}(\mathbf{y}|\mathbf{x})} - \mathrm{KL}(\theta)\right] - \pi_\theta(\mathbf{y}|\mathbf{x}) \sum_{\mathbf{y}' \in \mathcal{Y}} \pi_\theta(\mathbf{y}'|\mathbf{x})^2 \left[\ln \frac{\pi_\theta(\mathbf{y}'|\mathbf{x})}{\pi_{\theta_{\mathrm{ref}}}(\mathbf{y}'|\mathbf{x})} - \mathrm{KL}(\theta)\right]$$

$$\leq 2\pi_\theta(\mathbf{y}|\mathbf{x}) \ln \frac{1}{\pi_{\theta_{\mathrm{ref}}}(\mathbf{y}^m|\mathbf{x})} + \pi_\theta(\mathbf{y}|\mathbf{x}) \cdot \frac{1}{2e}$$

$$= \pi_\theta(\mathbf{y}|\mathbf{x}) \left(2 \ln \frac{1}{\min_{\mathbf{y}' \in \mathcal{Y}} \pi_{\theta_{\mathrm{ref}}}(\mathbf{y}'|\mathbf{x})} + \frac{1}{2e}\right).$$

$\square$

### D.6 Improved Lower Bound on Reward Increase Rate for Tabular Policies

For general autoregressive policies (Equation (2)), Theorem 4 (Appendix B.1) showed that the time required for the expected reward, measured with respect to any reward function $r$, to increase by an additive constant is:

$$\Omega\left(\mathbb{E}_{\mathbf{x} \sim \mathcal{S}}\left[\mathrm{Var}_{\mathbf{y} \sim \pi_{\theta(0)}(\cdot|\mathbf{x})}[r_{\mathrm{RM}}(\mathbf{x}, \mathbf{y})]\right]^{-\frac{1}{3}}\right).$$

In this appendix, we prove a stronger lower bound for tabular policies (Equation (3)), which is used in the proofs of Theorems 5 and 6. Specifically, for any prompt $\mathbf{x} \in \mathcal{S}$, Proposition 3 establishes that the time required for the expected reward (with respect to any reward function $r$) to increase by an additive constant is:

$$\Omega\left(\mathrm{Var}_{\mathbf{y} \sim \pi_{\theta(0)}(\cdot|\mathbf{x})}[r_{\mathrm{RM}}(\mathbf{x}, \mathbf{y})]^{-\frac{1}{2}}\right).$$

There are two main differences between the lower bounds of Theorem 4 and Proposition 3. First, the bound in Theorem 4 depends on the average reward variance over the training set $\mathcal{S}$, whereas that in Proposition 3 depends only on the reward variance for $\mathbf{x} \in \mathcal{S}$. This difference stems from the fact that under tabular policies, for any two prompts $\mathbf{x}, \mathbf{x}' \in \mathcal{X}$, the parameters governing $\pi_\theta(\cdot|\mathbf{x})$ and $\pi_\theta(\cdot|\mathbf{x}')$ are distinct (while for general autoregressive policies they can be shared). Second, is the different reward variance exponents, *i.e.*, $-1/3$ in Theorem 4 compared to $-1/2$ in Proposition 3. These exponents arise from the gradient norm upper bounds in Proposition 2 and Lemma 8. We note that the improved exponent in Proposition 3 is not critical for our arguments in Theorems 5 and 6.

**Proposition 3.** *Suppose gradient flow, over a tabular policy (Equation (3)), is used to maximize the RLHF objective with respect to a reward model $r_{\mathrm{RM}} : \mathcal{X} \times \mathcal{Y} \to [-1, 1]$ over a set of prompts $\mathcal{S}$ (Equations (1) and (4)) . For any $\gamma > 0$, prompt $\mathbf{x} \in \mathcal{S}$, and reward function $r : \mathcal{X} \times \mathcal{Y} \to [-1, 1]$ (e.g., $r$ can be the ground truth reward $r_{\mathrm{G}}$ or the reward model $r_{\mathrm{RM}}$), denote by $t_\gamma$ the initial time at which $\mathbb{E}_{\mathbf{y} \sim \pi_{\theta(t)}(\cdot|\mathbf{x})}[r(\mathbf{x}, \mathbf{y})] \geq \mathbb{E}_{\mathbf{y} \sim \pi_{\theta(0)}(\cdot|\mathbf{x})}[r(\mathbf{x}, \mathbf{y})] + \gamma$ (by convention, $t_\gamma = \infty$ if no such time exists). Then, for any initial policy $\pi_{\theta(0)}$:*

$$t_\gamma \geq 2|\mathcal{S}|\left(1 - \exp\left(-\frac{\gamma}{2}\right)\right) \cdot \frac{1}{\sqrt{\mathrm{Var}_{\mathbf{y} \sim \pi_{\theta(0)}(\cdot|\mathbf{x})}[r_{\mathrm{RM}}(\mathbf{x}, \mathbf{y})]}} \, .$$

*Proof.* Analogously to the proof of Theorem 4 (Appendix D.4), we start by bounding how far the policy parameters $\theta_{:,\mathbf{x}}(t)$ can deviate from their initial values until some time $T \geq 0$ of gradient flow. We will then translate this bound on the change in policy parameters to a lower bound on $t_\gamma$.

For all $t \geq 0$, we shorthand $\mathrm{Var}_{\mathbf{x}}^{\mathrm{KL}}(\theta(t)) := \mathrm{Var}_{\mathbf{y} \sim \pi_{\theta(t)}(\cdot|\mathbf{x})}\left[r_{\mathrm{RM}}^{\mathrm{KL}}(\mathbf{x}, \mathbf{y}; \theta(t))\right]$, where:

$$r_{\mathrm{RM}}^{\mathrm{KL}}(\mathbf{x}, \mathbf{y}; \theta(t)) := r_{\mathrm{RM}}(\mathbf{x}, \mathbf{y}) - \lambda \cdot \ln \frac{\pi_{\theta(t)}(\mathbf{y}|\mathbf{x})}{\pi_{\theta(0)}(\mathbf{y}|\mathbf{x})} \, .$$

Since $\pi_{\theta(t)}$ is a tabular policy, $\theta_{:,\mathbf{x}}(t)$ evolves independently of $\theta_{:,\mathbf{x}'}(t)$ for all $\mathbf{x}' \neq \mathbf{x}$. In particular, if $\mathrm{Var}_{\mathbf{x}}^{\mathrm{KL}}(\theta(t')) = 0$ at some $t' \geq 0$, then Lemma 8 implies that:

$$\frac{d}{dt}\theta_{:,\mathbf{x}}(t') = \frac{1}{|\mathcal{S}|}\nabla_{\theta_{:,\mathbf{x}}}\phi_{\mathrm{RLHF}}(\theta(t'); \mathbf{x}) = 0 \, .$$

In other words, gradient flow is at a critical point with respect to $\theta_{:,\mathbf{x}}$ at time $t'$. Due to the uniqueness of the gradient flow solution and the existence of a solution for which $\theta_{:,\mathbf{x}}(t) = \theta_{:,\mathbf{x}}(t')$ for all $t \geq 0$, we get that in this case the lower bound on $t_\gamma$ trivially holds — the expected reward for $\mathbf{x}$ never increases by $\gamma > 0$ from its initial value. Namely, $\theta_{:,\mathbf{x}}(t) = \theta_{:,\mathbf{x}}(0)$, and so $\pi_{\theta(t)}(\cdot|\mathbf{x}) = \pi_{\theta(0)}(\cdot|\mathbf{x})$ and $\mathbb{E}_{\mathbf{y} \sim \pi_{\theta(t)}(\cdot|\mathbf{x})}[r(\mathbf{x}, \mathbf{y})] = \mathbb{E}_{\mathbf{y} \sim \pi_{\theta(0)}(\cdot|\mathbf{x})}[r(\mathbf{x}, \mathbf{y})]$, for all $t \geq 0$.

Now, let us consider the case where $\mathrm{Var}_{\mathbf{x}}^{\mathrm{KL}}(\theta(t)) > 0$ for all $t \geq 0$. For any time $T \geq 0$, by the fundamental theorem of calculus and the triangle inequality:

$$\begin{aligned}
\|\theta_{:,\mathbf{x}}(T) - \theta_{:,\mathbf{x}}(0)\| &= \left\|\int_0^T \tfrac{d}{dt}\theta_{:,\mathbf{x}}(t)dt\right\| \\
&\leq \int_0^T \left\|\tfrac{d}{dt}\theta_{:,\mathbf{x}}(t)\right\|dt \\
&= \frac{1}{|\mathcal{S}|}\int_0^T \left\|\nabla_{\theta_{:,\mathbf{x}}}\phi_{\mathrm{RLHF}}(\theta(t); \mathbf{x})\right\|dt \, .
\end{aligned}$$

Lemma 8 then gives:

$$\|\theta_{:,\mathbf{x}}(T) - \theta_{:,\mathbf{x}}(0)\| \leq \frac{1}{|\mathcal{S}|}\int_0^T \sqrt{\mathrm{Var}_{\mathbf{x}}^{\mathrm{KL}}(\theta(t))}dt \, . \tag{10}$$

We now bound the rate at which $\mathrm{Var}_{\mathbf{x}}^{\mathrm{KL}}(\theta(t))$ can increase. In turn, this will yield an upper bound on $\|\theta_{:,\mathbf{x}}(T) - \theta_{:,\mathbf{x}}(0)\|$. Differentiating $\mathrm{Var}_{\mathbf{x}}^{\mathrm{KL}}(\theta(t))$ with respect to time leads to:

$$\begin{aligned}
\frac{d}{dt}\mathrm{Var}_{\mathbf{x}}^{\mathrm{KL}}(\theta(t)) &= \left\langle\nabla_{\theta_{:,\mathbf{x}}}\mathrm{Var}_{\mathbf{x}}^{\mathrm{KL}}(\theta(t)), \tfrac{d}{dt}\theta_{:,\mathbf{x}}(t)\right\rangle \\
&= \frac{1}{|\mathcal{S}|}\left\langle\nabla_{\theta_{:,\mathbf{x}}}\mathrm{Var}_{\mathbf{x}}^{\mathrm{KL}}(\theta(t)), \nabla_{\theta_{:,\mathbf{x}}}\phi_{\mathrm{RLHF}}(\theta(t); \mathbf{x})\right\rangle \, .
\end{aligned} \tag{11}$$

The gradient of $\mathrm{Var}_{\mathbf{x}}^{\mathrm{KL}}(\theta(t)) = \sum_{\mathbf{y}\in\mathcal{Y}} \pi_{\theta(t)}(\mathbf{y}|\mathbf{x}) \cdot \left[r_{\mathrm{RM}}^{\mathrm{KL}}(\mathbf{x},\mathbf{y};\theta(t)) - \phi_{\mathrm{RLHF}}(\theta(t);\mathbf{x})\right]^2$ with respect to $\theta_{:,\mathbf{x}}(t)$ can be written as follows:

$$\nabla_{\theta_{:,\mathbf{x}}} \mathrm{Var}_{\mathbf{x}}^{\mathrm{KL}}(\theta(t))$$
$$= \underbrace{\sum_{\mathbf{y}\in\mathcal{Y}} \left[r_{\mathrm{RM}}^{\mathrm{KL}}(\mathbf{x},\mathbf{y};\theta(t)) - \phi_{\mathrm{RLHF}}(\theta(t);\mathbf{x})\right]^2 \cdot \nabla_{\theta_{:,\mathbf{x}}} \pi_{\theta(t)}(\mathbf{y}|\mathbf{x})}_{\text{denote by } \mathbf{a}}$$
$$\underbrace{- 2\sum_{\mathbf{y}\in\mathcal{Y}} \pi_{\theta(t)}(\mathbf{y}|\mathbf{x})\left[r_{\mathrm{RM}}^{\mathrm{KL}}(\mathbf{x},\mathbf{y};\theta(t)) - \phi_{\mathrm{RLHF}}(\theta(t);\mathbf{x})\right] \cdot \nabla_{\theta_{:,\mathbf{x}}} \phi_{\mathrm{RLHF}}(\theta(t);\mathbf{x})}_{\text{denote by } \mathbf{b}}$$
$$\underbrace{- 2\lambda\sum_{\mathbf{y}\in\mathcal{Y}} \pi_{\theta(t)}(\mathbf{y}|\mathbf{x})\left[r_{\mathrm{RM}}^{\mathrm{KL}}(\mathbf{x},\mathbf{y};\theta(t)) - \phi_{\mathrm{RLHF}}(\theta(t);\mathbf{x})\right] \cdot \nabla_{\theta_{:,\mathbf{x}}} \ln \pi_{\theta(t)}(\mathbf{y}|\mathbf{x})}_{\text{denote by } \mathbf{c}} .$$

We first consider the term $\mathbf{a}$. Since

$$\nabla_{\theta_{:,\mathbf{x}}} \pi_{\theta(t)}(\mathbf{y}|\mathbf{x}) = \pi_{\theta(t)}(\mathbf{y}|\mathbf{x}) \nabla_{\theta_{:,\mathbf{x}}} \ln \pi_{\theta(t)}(\mathbf{y}|\mathbf{x})$$
$$= \pi_{\theta(t)}(\mathbf{y}|\mathbf{x})\left(\mathbf{e}_{\mathbf{y}} - \pi_{\theta(t)}(\cdot|\mathbf{x})\right),$$

where, with slight abuse of notation, $\pi_{\theta(t)}(\cdot|\mathbf{x}) = \mathrm{softmax}(\theta_{:,\mathbf{x}}(t))$, we have that:

$$\langle \mathbf{a}, \nabla_{\theta_{:,\mathbf{x}}} \phi_{\mathrm{RLHF}}(\theta(t);\mathbf{x}) \rangle$$
$$= \sum_{\mathbf{y}\in\mathcal{Y}} \pi_{\theta(t)}(\mathbf{y}|\mathbf{x})\left[r_{\mathrm{RM}}^{\mathrm{KL}}(\mathbf{x},\mathbf{y};\theta(t)) - \phi_{\mathrm{RLHF}}(\theta(t);\mathbf{x})\right]^2 \langle \mathbf{e}_{\mathbf{y}} - \pi_{\theta(t)}(\cdot|\mathbf{x}), \nabla_{\theta_{:,\mathbf{x}}} \phi_{\mathrm{RLHF}}(\theta(t);\mathbf{x}) \rangle .$$

By the duality of the $\ell_\infty$ and $\ell_1$ norms, Lemma 8, and the fact that $\|\mathbf{e}_{\mathbf{y}} - \pi_{\theta(t)}(\cdot|\mathbf{x})\|_\infty \leq 1$, we get:

$$\langle \mathbf{e}_{\mathbf{y}} - \pi_{\theta(t)}(\cdot|\mathbf{x}), \nabla_{\theta_{:,\mathbf{x}}} \phi_{\mathrm{RLHF}}(\theta(t);\mathbf{x}) \rangle \leq \left\|\mathbf{e}_{\mathbf{y}} - \pi_{\theta(t)}(\cdot|\mathbf{x})\right\|_\infty \left\|\nabla_{\theta_{:,\mathbf{x}}} \phi_{\mathrm{RLHF}}(\theta(t);\mathbf{x})\right\|_1$$
$$\leq \sqrt{\mathrm{Var}_{\mathbf{x}}^{\mathrm{KL}}(\theta(t))} .$$

Hence:

$$\langle \mathbf{a}, \nabla_{\theta_{:,\mathbf{x}}} \phi_{\mathrm{RLHF}}(\theta(t);\mathbf{x}) \rangle \leq \sqrt{\mathrm{Var}_{\mathbf{x}}^{\mathrm{KL}}(\theta(t))} \sum_{\mathbf{y}\in\mathcal{Y}} \pi_{\theta(t)}(\mathbf{y}|\mathbf{x})\left[r_{\mathrm{RM}}^{\mathrm{KL}}(\mathbf{x},\mathbf{y};\theta(t)) - \phi_{\mathrm{RLHF}}(\theta(t);\mathbf{x})\right]^2$$
$$= \mathrm{Var}_{\mathbf{x}}^{\mathrm{KL}}(\theta(t))^{3/2} .$$

As for the term $\mathbf{b}$, notice that it is equal to zero:

$$\mathbf{b} = 2\nabla_{\theta_{:,\mathbf{x}}} \phi_{\mathrm{RLHF}}(\theta(t);\mathbf{x}) \sum_{\mathbf{y}\in\mathcal{Y}} \pi_{\theta(t)}(\mathbf{y}|\mathbf{x})\left[r_{\mathrm{RM}}^{\mathrm{KL}}(\mathbf{x},\mathbf{y};\theta(t)) - \phi_{\mathrm{RLHF}}(\theta(t);\mathbf{x})\right]$$
$$= 2\nabla_{\theta_{:,\mathbf{x}}} \phi_{\mathrm{RLHF}}(\theta(t);\mathbf{x})\left[\underbrace{\sum_{\mathbf{y}\in\mathcal{Y}} \pi_{\theta(t)}(\mathbf{y}|\mathbf{x}) r_{\mathrm{RM}}^{\mathrm{KL}}(\mathbf{x},\mathbf{y};\theta(t))}_{= \phi_{\mathrm{RLHF}}(\theta(t);\mathbf{x})} - \phi_{\mathrm{RLHF}}(\theta(t);\mathbf{x}) \cdot \underbrace{\sum_{\mathbf{y}\in\mathcal{Y}} \pi_{\theta(t)}(\mathbf{y}|\mathbf{x})}_{=1}\right]$$
$$= 0 ,$$

and so $\langle \mathbf{b}, \nabla_{\theta_{:,\mathbf{x}}} \phi_{\mathrm{RLHF}}(\theta(t);\mathbf{x}) \rangle = 0$. Lastly:

$$\mathbf{c} = 2\lambda\nabla_{\theta_{:,\mathbf{x}}} \phi_{\mathrm{RLHF}}(\theta(t);\mathbf{x}) - 2\lambda\phi_{\mathrm{RLHF}}(\theta(t);\mathbf{x}) \cdot \underbrace{\sum_{\mathbf{y}\in\mathcal{Y}} \pi_{\theta(t)}(\mathbf{y}|\mathbf{x}) \cdot \nabla_{\theta_{:,\mathbf{x}}} \ln \pi_{\theta(t)}(\mathbf{y}|\mathbf{x})}_{= \nabla_{\theta_{:,\mathbf{x}}} \sum_{\mathbf{y}\in\mathcal{Y}} \pi_{\theta(t)}(\mathbf{y}|\mathbf{x})=0}$$
$$= 2\lambda\nabla_{\theta_{:,\mathbf{x}}} \phi_{\mathrm{RLHF}}(\theta(t);\mathbf{x}) .$$

Thus, going back to Equation (11) we obtain:

$$\frac{d}{dt}\operatorname{Var}^{\mathrm{KL}}_{\mathbf{x}}(\theta(t)) = \frac{1}{|\mathcal{S}|}\left\langle \mathbf{a}, \nabla_{\theta_{:,\mathbf{x}}}\phi_{\mathrm{RLHF}}(\theta(t);\mathbf{x})\right\rangle$$

$$- \frac{1}{|\mathcal{S}|}\left\langle \mathbf{b}, \nabla_{\theta_{:,\mathbf{x}}}\phi_{\mathrm{RLHF}}(\theta(t);\mathbf{x})\right\rangle$$

$$- \frac{1}{|\mathcal{S}|}\left\langle \mathbf{c}, \nabla_{\theta_{:,\mathbf{x}}}\phi_{\mathrm{RLHF}}(\theta(t);\mathbf{x})\right\rangle$$

$$\leq \frac{1}{|\mathcal{S}|}\left(\operatorname{Var}^{\mathrm{KL}}_{\mathbf{x}}(\theta(t))^{3/2} - 2\lambda \cdot \left\|\nabla_{\theta_{:,\mathbf{x}}}\phi_{\mathrm{RLHF}}(\theta(t);\mathbf{x})\right\|^2\right)$$

$$\leq \frac{1}{|\mathcal{S}|}\operatorname{Var}^{\mathrm{KL}}_{\mathbf{x}}(\theta(t))^{3/2}.$$

By Lemma 1, this implies that for all $0 \leq t < 2|\mathcal{S}|/\sqrt{\operatorname{Var}^{\mathrm{KL}}_{\mathbf{x}}(\theta(0))}$:

$$\operatorname{Var}^{\mathrm{KL}}_{\mathbf{x}}(\theta(t)) \leq \frac{\operatorname{Var}^{\mathrm{KL}}_{\mathbf{x}}(\theta(0))}{\left(1 - \frac{1}{2|\mathcal{S}|}\sqrt{\operatorname{Var}^{\mathrm{KL}}_{\mathbf{x}}(\theta(0))}\cdot t\right)^2}.$$

Notice that since $r^{\mathrm{KL}}_{\mathrm{RM}}(\mathbf{x},\mathbf{y};\theta(0)) = r_{\mathrm{RM}}(\mathbf{x},\mathbf{y}) - \lambda \cdot \ln\frac{\pi_{\theta(0)}(\mathbf{y}|\mathbf{x})}{\pi_{\theta(0)}(\mathbf{y}|\mathbf{x})} = r_{\mathrm{RM}}(\mathbf{x},\mathbf{y})$ for all $\mathbf{y} \in \mathcal{Y}$:

$$\operatorname{Var}^{\mathrm{KL}}_{\mathbf{x}}(\theta(0)) = \operatorname{Var}_{\mathbf{y}\sim\pi_{\theta(0)}(\cdot|\mathbf{x})}\left[r^{\mathrm{KL}}_{\mathrm{RM}}(\mathbf{x},\mathbf{y};\theta(0))\right] = \operatorname{Var}_{\mathbf{y}\sim\pi_{\theta(0)}(\cdot|\mathbf{x})}\left[r_{\mathrm{RM}}(\mathbf{x},\mathbf{y})\right].$$

Thus, we have that for all $0 \leq t < 2|\mathcal{S}|/\sqrt{\operatorname{Var}_{\mathbf{x}}(\theta(0))}$:

$$\operatorname{Var}^{\mathrm{KL}}_{\mathbf{x}}(\theta(t)) \leq \frac{\operatorname{Var}_{\mathbf{x}}(\theta(0))}{\left(1 - \frac{1}{2|\mathcal{S}|}\sqrt{\operatorname{Var}_{\mathbf{x}}(\theta(0))}\cdot t\right)^2},$$

where $\operatorname{Var}_{\mathbf{x}}(\theta(t)) := \operatorname{Var}_{\mathbf{y}\sim\pi_{\theta(t)}(\cdot|\mathbf{x})}\left[r_{\mathrm{RM}}(\mathbf{x},\mathbf{y})\right]$. Plugging this into Equation (10), for all $0 \leq T < 2|\mathcal{S}|/\sqrt{\operatorname{Var}_{\mathbf{x}}(\theta(0))}$ we get that:

$$\|\theta_{:,\mathbf{x}}(T) - \theta_{:,\mathbf{x}}(0)\| \leq \frac{1}{|\mathcal{S}|}\int_0^T \sqrt{\operatorname{Var}^{\mathrm{KL}}_{\mathbf{x}}(\theta(t))}dt$$

$$\leq \frac{1}{|\mathcal{S}|}\int_0^T \frac{\sqrt{\operatorname{Var}_{\mathbf{x}}(\theta(0))}}{1 - \frac{1}{2|\mathcal{S}|}\sqrt{\operatorname{Var}_{\mathbf{x}}(\theta(0))}\cdot t}dt$$

$$= \frac{\sqrt{\operatorname{Var}_{\mathbf{x}}(\theta(0))}}{|\mathcal{S}|}\left(-\frac{2|\mathcal{S}|}{\sqrt{\operatorname{Var}_{\mathbf{x}}(\theta(0))}}\cdot\ln\left(1 - \frac{1}{2|\mathcal{S}|}\sqrt{\operatorname{Var}_{\mathbf{x}}(\theta(0))}\cdot t\right)\right)\Bigg|_{t=0}^T$$

$$= 2\ln\left(\frac{1}{1 - \frac{1}{2|\mathcal{S}|}\sqrt{\operatorname{Var}_{\mathbf{x}}(\theta(0))}\cdot T}\right).$$

It remains to translate the upper bound on $\|\theta_{:,\mathbf{x}}(T) - \theta_{:,\mathbf{x}}(0)\|$ into a lower bound on $t_\gamma$. To simplify notation, let $V(\theta(t);\mathbf{x}) := \mathbb{E}_{\mathbf{x}\sim\pi_{\theta(t)}(\cdot|\mathbf{x})}\left[r(\mathbf{x},\mathbf{y})\right]$. Since $\pi_{\theta(t)}$ is a tabular policy (Equation (3)), we can view $V(\theta(t);\mathbf{x})$ as a function of only $\theta_{:,\mathbf{x}}(t)$, *i.e.*, it does not depend on $\theta_{:,\mathbf{x}'}(t)$ for all $\mathbf{x}' \neq \mathbf{x}$. Thus, by Lemma 8 and the fact that the variance of a random variable bounded within $[-1,1]$ is at most 1, we have that:

$$\left\|\nabla_{\theta_{:,\mathbf{x}}}V(\theta(t);\mathbf{x})\right\| \leq \sqrt{\operatorname{Var}_{\mathbf{y}\sim\pi_{\theta(t)}(\cdot|\mathbf{x})}\left[r(\mathbf{x},\mathbf{y})\right]} \leq 1.$$

This implies that $V(\theta(t);\mathbf{x})$ is 1-Lipschitz as a function of $\theta_{:,\mathbf{x}}(t)$, and so:

$$|V(\theta(T);\mathbf{x}) - V(\theta(0);\mathbf{x})| \leq \|\theta_{:,\mathbf{x}}(T) - \theta_{:,\mathbf{x}}(0)\| \leq 2\ln\left(\frac{1}{1 - \frac{1}{2|\mathcal{S}|}\sqrt{\operatorname{Var}_{\mathbf{x}}(\theta(0))}\cdot T}\right).$$

Now, if $t_\gamma \geq 2|\mathcal{S}|/\sqrt{\mathrm{Var}_\mathbf{x}(\theta(0))}$ then we are done. Otherwise, for $V(\theta(t_\gamma); \mathbf{x}) - V(\theta(0); \mathbf{x}) \geq \gamma$ to hold, it must be the case that:

$$\gamma \leq 2\ln\left(\frac{1}{1 - \frac{1}{2|\mathcal{S}|}\sqrt{\mathrm{Var}_\mathbf{x}(\theta(0)) \cdot t_\gamma}}\right).$$

Rearranging the inequality concludes the proof:

$$t_\gamma \geq 2|\mathcal{S}|\left(1 - \exp\left(-\frac{\gamma}{2}\right)\right) \cdot \frac{1}{\sqrt{\mathrm{Var}_\mathbf{x}(\theta(0))}}.$$

$\square$

### D.7 Sufficient Conditions for Fast Ground Truth Reward Increase

In this appendix, we establish conditions on the reward model, initial policy, and KL regularization coefficient that ensure a fast increase in ground truth reward during gradient flow. These sufficient conditions are used in the proofs of Theorems 5 and 6.

**Proposition 4.** *Suppose gradient flow, over a tabular policy (Equation (3)), is used to maximize the RLHF objective with respect to a reward model $r_{\mathrm{RM}} : \mathcal{X} \times \mathcal{Y} \to [-1, 1]$ over a set of prompts $\mathcal{S}$ (Equations (1) and (4)). Given a prompt $\mathbf{x} \in \mathcal{S}$, let $\gamma > 0$ be a desired expected ground truth reward increase and $\mathcal{Y}^+ \subseteq \mathcal{Y}$ be a set of outputs such that every $\mathbf{y}' \in \mathcal{Y}^+$ satisfies $r_{\mathrm{G}}(\mathbf{x}, \mathbf{y}') > V_{\mathrm{G}}(\theta(0); \mathbf{x}) + \gamma$ and $r_{\mathrm{RM}}(\mathbf{x}, \mathbf{y}') > V_{\mathrm{RM}}(\theta(0); \mathbf{x})$, where $V_{\mathrm{G}}(\theta(t); \mathbf{x}) := \mathbb{E}_{\mathbf{y} \sim \pi_{\theta(t)}(\cdot|\mathbf{x})}[r_{\mathrm{G}}(\mathbf{x}, \mathbf{y})]$ and $V_{\mathrm{RM}}(\theta(t); \mathbf{x}) := \mathbb{E}_{\mathbf{y} \sim \pi_{\theta(t)}(\cdot|\mathbf{x})}[r_{\mathrm{RM}}(\mathbf{x}, \mathbf{y})]$. Denote*

$$\rho := \frac{V_{\mathrm{G}}(\theta(0); \mathbf{x}) + \gamma + 1}{\min_{\mathbf{y} \in \mathcal{Y}^+} r_{\mathrm{G}}(\mathbf{x}, \mathbf{y}) + 1} \in (0, 1)$$

*and assume that the following conditions hold.*[8]

- *The reward model $r_{\mathrm{RM}}$ assigns to outputs in $\mathcal{Y}^+$ rewards that are higher than $V_{\mathrm{RM}}(\theta(0); \mathbf{x})$, yet not too spread out:*

$$\delta := (1 - \rho)\underbrace{\left(\max_{\mathbf{y} \in \mathcal{Y}^+} r_{\mathrm{RM}}(\mathbf{x}, \mathbf{y}) - V_{\mathrm{RM}}(\theta(0); \mathbf{x})\right)}_{\text{separation from initial expected (proxy) reward}} - \underbrace{\max_{\mathbf{y}, \mathbf{y}' \in \mathcal{Y}^+}|r_{\mathrm{RM}}(\mathbf{x}, \mathbf{y}) - r_{\mathrm{RM}}(\mathbf{x}, \mathbf{y}')|}_{\text{spread}} > 0.$$

- *The KL regularization coefficient $\lambda \geq 0$ is not too large:*

$$\lambda \leq \frac{\pi_{\theta(0)}(\mathcal{Y}^+|\mathbf{x})(1 - \rho)\delta}{8|\mathcal{Y}^+|\left(2\ln\frac{1}{\min_{\mathbf{y} \in \mathcal{Y}} \pi_{\theta(0)}(\mathbf{y}|\mathbf{x})} + \frac{1}{2e}\right)}.$$

- *The initial probability assigned to $\mathcal{Y}_{\mathrm{bad}} := \{\mathbf{y} \in \mathcal{Y} : r_{\mathrm{RM}}(\mathbf{x}, \mathbf{y}) > V_{\mathrm{RM}}(\theta(0); \mathbf{x})\} \setminus \mathcal{Y}^+$ is small:*

$$\pi_{\theta(0)}(\mathcal{Y}_{\mathrm{bad}}|\mathbf{x}) \leq \frac{\pi_{\theta(0)}(\mathcal{Y}^+|\mathbf{x})^2(1 - \rho)\delta}{16|\mathcal{Y}^+|}\exp\left(-\frac{20|\mathcal{Y}^+|}{(1 - \rho)\delta}\left(\frac{1}{\pi_{\theta(0)}(\mathcal{Y}^+|\mathbf{x})} - \frac{1}{\rho}\right)\right).$$

*Then, denoting by $t_\gamma$ the initial time at which $V_{\mathrm{G}}(\theta(t); \mathbf{x}) \geq V_{\mathrm{G}}(\theta(0); \mathbf{x}) + \gamma$, it holds that:*

$$t_\gamma \leq \frac{4|\mathcal{S}||\mathcal{Y}^+|}{(1 - \rho)\delta}\left(\frac{1}{\pi_{\theta(0)}(\mathcal{Y}^+|\mathbf{x})} - \frac{1}{\rho}\right) = \mathcal{O}\left(\frac{1}{\pi_{\theta(0)}(\mathcal{Y}^+|\mathbf{x})}\right).$$

*Proof.* If $\pi_{\theta(t)}(\mathcal{Y}^+|\mathbf{x}) \geq \rho$ at some $t \geq 0$, then $V_{\mathrm{G}}(\theta(t); \mathbf{x}) \geq V_{\mathrm{G}}(\theta(0); \mathbf{x}) + \gamma$ since:

$$
\begin{aligned}
V_{\mathrm{G}}(\theta(t); \mathbf{x}) &\geq \pi_{\theta(t)}(\mathcal{Y}^+|\mathbf{x}) \cdot \min_{\mathbf{y} \in \mathcal{Y}^+} r_{\mathrm{G}}(\mathbf{x}, \mathbf{y}) - \left(1 - \pi_{\theta(t)}(\mathcal{Y}^+|\mathbf{x})\right) \\
&= \pi_{\theta(t)}(\mathcal{Y}^+|\mathbf{x})\left(\min_{\mathbf{y} \in \mathcal{Y}^+} r_{\mathrm{G}}(\mathbf{x}, \mathbf{y}) + 1\right) - 1 \qquad (12) \\
&\geq V_{\mathrm{G}}(\theta(0); \mathbf{x}) + \gamma,
\end{aligned}
$$

---

[8]We defined $\rho$ such that $\pi_{\theta(t)}(\mathcal{Y}^+|\mathbf{x}) \geq \rho$ implies $V_{\mathrm{G}}(\theta(t); \mathbf{x}) \geq V_{\mathrm{G}}(\theta(0); \mathbf{x}) + \gamma$ (see Equation (12)).

where in the first inequality we used the fact that the minimal possible reward is $-1$. This implies that $\pi_{\theta(t)}(\mathcal{Y}^+|\mathbf{x}) < \rho$ for all $t \in [0, t_\gamma)$. For convenience, let us denote:

$$T := \frac{4|\mathcal{S}||\mathcal{Y}^+|}{(1-\rho)\delta}\left(\frac{1}{\pi_{\theta(0)}(\mathcal{Y}^+|\mathbf{x})} - \frac{1}{\rho}\right).$$

Seeking a contradiction, assume $t_\gamma > T$. We will show that, over the time interval $[0, T]$, the rate at which $\pi_{\theta(t)}(\mathcal{Y}^+|\mathbf{x})$ increases is sufficiently high such that $\pi_{\theta(t)}(\mathcal{Y}^+|\mathbf{x}) \geq \rho$ must occur until time $T$.

For any $t \in [0, T]$ and $\mathbf{y} \in \mathcal{Y}^+$, by Lemma 9 we know that:

$$\frac{d}{dt}\pi_{\theta(t)}(\mathbf{y}|\mathbf{x}) = \frac{1}{|\mathcal{S}|}\cdot\pi_{\theta(t)}(\mathbf{y}|\mathbf{x})^2\left[r_{\mathrm{RM}}(\mathbf{x},\mathbf{y}) - V_{\mathrm{RM}}(\theta(t);\mathbf{x})\right]$$

$$- \frac{1}{|\mathcal{S}|}\cdot\pi_{\theta(t)}(\mathbf{y}|\mathbf{x})\sum\nolimits_{\mathbf{y}'\in\mathcal{Y}}\pi_{\theta(t)}(\mathbf{y}'|\mathbf{x})^2\left[r_{\mathrm{RM}}(\mathbf{x},\mathbf{y}') - V_{\mathrm{RM}}(\theta(t);\mathbf{x})\right]$$

$$- \frac{\lambda}{|\mathcal{S}|}\cdot\pi_{\theta(t)}(\mathbf{y}|\mathbf{x})^2\left[\ln\frac{\pi_{\theta(t)}(\mathbf{y}|\mathbf{x})}{\pi_{\theta(0)}(\mathbf{y}|\mathbf{x})} - \mathrm{KL}(\theta(t))\right]$$

$$+ \frac{\lambda}{|\mathcal{S}|}\cdot\pi_{\theta(t)}(\mathbf{y}|\mathbf{x})\sum\nolimits_{\mathbf{y}'\in\mathcal{Y}}\pi_{\theta(t)}(\mathbf{y}'|\mathbf{x})^2\left[\ln\frac{\pi_{\theta(t)}(\mathbf{y}'|\mathbf{x})}{\pi_{\theta(0)}(\mathbf{y}'|\mathbf{x})} - \mathrm{KL}(\theta(t))\right],$$

where $\mathrm{KL}(\theta(t)) := \mathrm{KL}(\pi_{\theta(t)}(\cdot|\mathbf{x})||\pi_{\theta(0)}(\cdot|\mathbf{x}))$. Using Lemma 11 we can lower bound the contribution of the KL regularization terms:

$$\frac{d}{dt}\pi_{\theta(t)}(\mathbf{y}|\mathbf{x}) \geq \frac{1}{|\mathcal{S}|}\cdot\pi_{\theta(t)}(\mathbf{y}|\mathbf{x})^2\left[r_{\mathrm{RM}}(\mathbf{x},\mathbf{y}) - V_{\mathrm{RM}}(\theta(t);\mathbf{x})\right]$$

$$- \frac{1}{|\mathcal{S}|}\cdot\pi_{\theta(t)}(\mathbf{y}|\mathbf{x})\sum\nolimits_{\mathbf{y}'\in\mathcal{Y}}\pi_{\theta(t)}(\mathbf{y}'|\mathbf{x})^2\left[r_{\mathrm{RM}}(\mathbf{x},\mathbf{y}') - V_{\mathrm{RM}}(\theta(t);\mathbf{x})\right]$$

$$- \frac{\lambda}{|\mathcal{S}|}\cdot\pi_{\theta(t)}(\mathbf{y}|\mathbf{x})\left(2\ln\frac{1}{\pi_{\theta(0)}(\mathbf{y}^m|\mathbf{x})} + \frac{1}{2e}\right),$$

where $\mathbf{y}^m \in \operatorname{argmin}_{\mathbf{y}'\in\mathcal{Y}}\pi_{\theta(0)}(\mathbf{y}'|\mathbf{x})$. Summing over all outputs in $\mathcal{Y}^+$ then yields:

$$\frac{d}{dt}\pi_{\theta(t)}(\mathcal{Y}^+|\mathbf{x}) \geq \frac{1}{|\mathcal{S}|}\sum\nolimits_{\mathbf{y}\in\mathcal{Y}^+}\pi_{\theta(t)}(\mathbf{y}|\mathbf{x})^2\left[r_{\mathrm{RM}}(\mathbf{x},\mathbf{y}) - V_{\mathrm{RM}}(\theta(t);\mathbf{x})\right]$$

$$- \frac{1}{|\mathcal{S}|}\cdot\pi_{\theta(t)}(\mathcal{Y}^+|\mathbf{x})\sum\nolimits_{\mathbf{y}'\in\mathcal{Y}}\pi_{\theta(t)}(\mathbf{y}'|\mathbf{x})^2\left[r_{\mathrm{RM}}(\mathbf{x},\mathbf{y}') - V_{\mathrm{RM}}(\theta(t);\mathbf{x})\right]$$

$$- \frac{\lambda}{|\mathcal{S}|}\cdot\pi_{\theta(t)}(\mathcal{Y}^+|\mathbf{x})\left(2\ln\frac{1}{\pi_{\theta(0)}(\mathbf{y}^m|\mathbf{x})} + \frac{1}{2e}\right)$$

$$= \frac{1}{|\mathcal{S}|}\left(1 - \pi_{\theta(t)}(\mathcal{Y}^+|\mathbf{x})\right)\sum\nolimits_{\mathbf{y}\in\mathcal{Y}^+}\pi_{\theta(t)}(\mathbf{y}|\mathbf{x})^2\underbrace{\left[r_{\mathrm{RM}}(\mathbf{x},\mathbf{y}) - V_{\mathrm{RM}}(\theta(t);\mathbf{x})\right]}_{a(\mathbf{y})}$$

$$- \frac{1}{|\mathcal{S}|}\pi_{\theta(t)}(\mathcal{Y}^+|\mathbf{x})\sum\nolimits_{\mathbf{y}'\in\mathcal{Y}\backslash\mathcal{Y}^+}\pi_{\theta(t)}(\mathbf{y}'|\mathbf{x})^2\underbrace{\left[r_{\mathrm{RM}}(\mathbf{x},\mathbf{y}') - V_{\mathrm{RM}}(\theta(t);\mathbf{x})\right]}_{b(\mathbf{y}')}$$

$$- \frac{\lambda}{|\mathcal{S}|}\pi_{\theta(t)}(\mathcal{Y}^+|\mathbf{x})\left(2\ln\frac{1}{\pi_{\theta(0)}(\mathbf{y}^m|\mathbf{x})} + \frac{1}{2e}\right).$$

$$(13)$$

Towards lower bounding $\frac{d}{dt}\pi_{\theta(t)}(\mathcal{Y}^+|\mathbf{x})$, we treat separately the terms marked in the equation above by $a(\mathbf{y})$, for $\mathbf{y} \in \mathcal{Y}^+$, and $b(\mathbf{y}')$, for $\mathbf{y}' \in \mathcal{Y} \setminus \mathcal{Y}^+$. Starting with $a(\mathbf{y})$:

$$a(\mathbf{y}) = \sum\nolimits_{\mathbf{y}'\in\mathcal{Y}}\pi_{\theta(t)}(\mathbf{y}'|\mathbf{x})(r_{\mathrm{RM}}(\mathbf{x},\mathbf{y}) - r_{\mathrm{RM}}(\mathbf{x},\mathbf{y}'))$$

$$\geq \pi_{\theta(t)}(\mathcal{Y}^+|\mathbf{x})\left(r_{\mathrm{RM}}(\mathbf{x},\mathbf{y}) - \max\nolimits_{\mathbf{y}'\in\mathcal{Y}^+}r_{\mathrm{RM}}(\mathbf{x},\mathbf{y}')\right)$$

$$+ \sum\nolimits_{\mathbf{y}'\in\mathcal{Y}\backslash\mathcal{Y}^+}\pi_{\theta(t)}(\mathbf{y}'|\mathbf{x})(r_{\mathrm{RM}}(\mathbf{x},\mathbf{y}) - r_{\mathrm{RM}}(\mathbf{x},\mathbf{y}')).$$

By the definitions of $\mathcal{Y}^+$ and $\mathcal{Y}_{\mathrm{bad}}$, any $\mathbf{y}' \in \mathcal{Y} \setminus (\mathcal{Y}^+ \cup \mathcal{Y}_{\mathrm{bad}})$ satisfies $r_{\mathrm{RM}}(\mathbf{x}, \mathbf{y}') \leq V_{\mathrm{RM}}(\theta(0); \mathbf{x})$. Thus, $r_{\mathrm{RM}}(\mathbf{x}, \mathbf{y}) - r_{\mathrm{RM}}(\mathbf{x}, \mathbf{y}') \geq r_{\mathrm{RM}}(\mathbf{x}, \mathbf{y}) - V_{\mathrm{RM}}(\theta(0); \mathbf{x})$ for each such $\mathbf{y}'$. Adding and subtracting $\sum_{\mathbf{y}' \in \mathcal{Y}_{\mathrm{bad}}} \pi_{\theta(t)}(\mathbf{y}'|\mathbf{x})(r_{\mathrm{RM}}(\mathbf{x}, \mathbf{y}) - V_{\mathrm{RM}}(\theta(0); \mathbf{x}))$ from the inequality above then leads to:

$$
\begin{aligned}
a(\mathbf{y}) \geq\ & \pi_{\theta(t)}\big(\mathcal{Y}^+|\mathbf{x}\big)\big(r_{\mathrm{RM}}(\mathbf{x}, \mathbf{y}) - \max_{\mathbf{y}' \in \mathcal{Y}^+} r_{\mathrm{RM}}(\mathbf{x}, \mathbf{y}')\big) \\
&+ \sum_{\mathbf{y}' \in \mathcal{Y} \setminus \mathcal{Y}^+} \pi_{\theta(t)}(\mathbf{y}'|\mathbf{x})(r_{\mathrm{RM}}(\mathbf{x}, \mathbf{y}) - V_{\mathrm{RM}}(\theta(0); \mathbf{x})) \\
&+ \sum_{\mathbf{y}' \in \mathcal{Y}_{\mathrm{bad}}} \pi_{\theta(t)}(\mathbf{y}'|\mathbf{x})\big[r_{\mathrm{RM}}(\mathbf{x}, \mathbf{y}) - r_{\mathrm{RM}}(\mathbf{x}, \mathbf{y}') - r_{\mathrm{RM}}(\mathbf{x}, \mathbf{y}) + V_{\mathrm{RM}}(\theta(0); \mathbf{x})\big] \\
=\ & \pi_{\theta(t)}\big(\mathcal{Y}^+|\mathbf{x}\big)\big(r_{\mathrm{RM}}(\mathbf{x}, \mathbf{y}) - \max_{\mathbf{y}' \in \mathcal{Y}^+} r_{\mathrm{RM}}(\mathbf{x}, \mathbf{y}')\big) \\
&+ \big(1 - \pi_{\theta(t)}\big(\mathcal{Y}^+|\mathbf{x}\big)\big)(r_{\mathrm{RM}}(\mathbf{x}, \mathbf{y}) - V_{\mathrm{RM}}(\theta(0); \mathbf{x})) \\
&+ \sum_{\mathbf{y}' \in \mathcal{Y}_{\mathrm{bad}}} \pi_{\theta(t)}(\mathbf{y}'|\mathbf{x})\big[V_{\mathrm{RM}}(\theta(0); \mathbf{x}) - r_{\mathrm{RM}}(\mathbf{x}, \mathbf{y}')\big].
\end{aligned}
$$

Since *(i)* $\pi_{\theta(t)}(\mathcal{Y}^+|\mathbf{x}) < \rho$ for all $t \in [0, t_\gamma)$, *(ii)* $r_{\mathrm{RM}}(\mathbf{x}, \mathbf{y}) - \max_{\mathbf{y}' \in \mathcal{Y}^+} r_{\mathrm{RM}}(\mathbf{x}, \mathbf{y}') \leq 0$, *(iii)* $r_{\mathrm{RM}}(\mathbf{x}, \mathbf{y}) - V_{\mathrm{RM}}(\theta(0); \mathbf{x}) \geq 0$, and *(iv)* $|V_{\mathrm{RM}}(\theta(0); \mathbf{x}) - r_{\mathrm{RM}}(\mathbf{x}, \mathbf{y}')| \leq 2$ for all $\mathbf{y}' \in \mathcal{Y}$ (recall $r_{\mathrm{RM}}$ is bounded within $[-1, 1]$), we obtain:

$$
\begin{aligned}
a(\mathbf{y}) & \\
\geq\ & \rho\big(r_{\mathrm{RM}}(\mathbf{x}, \mathbf{y}) - \max_{\mathbf{y}' \in \mathcal{Y}^+} r_{\mathrm{RM}}(\mathbf{x}, \mathbf{y}')\big) \\
&+ (1 - \rho)(r_{\mathrm{RM}}(\mathbf{x}, \mathbf{y}) - V_{\mathrm{RM}}(\theta(0); \mathbf{x})) \\
&- 2\pi_{\theta(t)}(\mathcal{Y}_{\mathrm{bad}}|\mathbf{x}) \\
=\ & r_{\mathrm{RM}}(\mathbf{x}, \mathbf{y}) - \rho \cdot \max_{\mathbf{y}' \in \mathcal{Y}^+} r_{\mathrm{RM}}(\mathbf{x}, \mathbf{y}') - (1 - \rho)V_{\mathrm{RM}}(\theta(0); \mathbf{x}) - 2\pi_{\theta(t)}(\mathcal{Y}_{\mathrm{bad}}|\mathbf{x}) \\
\geq\ & \min_{\mathbf{y}' \in \mathcal{Y}^+} r_{\mathrm{RM}}(\mathbf{x}, \mathbf{y}') - \rho \cdot \max_{\mathbf{y}' \in \mathcal{Y}^+} r_{\mathrm{RM}}(\mathbf{x}, \mathbf{y}') - (1 - \rho)V_{\mathrm{RM}}(\theta(0); \mathbf{x}) - 2\pi_{\theta(t)}(\mathcal{Y}_{\mathrm{bad}}|\mathbf{x}) \\
=\ & (1 - \rho) \max_{\mathbf{y}' \in \mathcal{Y}^+} r_{\mathrm{RM}}(\mathbf{x}, \mathbf{y}') - (1 - \rho)V_{\mathrm{RM}}(\theta(0); \mathbf{x}) - \max_{\mathbf{y}' \in \mathcal{Y}^+} r_{\mathrm{RM}}(\mathbf{x}, \mathbf{y}') + \min_{\mathbf{y}' \in \mathcal{Y}^+} r_{\mathrm{RM}}(\mathbf{x}, \mathbf{y}') \\
&- 2\pi_{\theta(t)}(\mathcal{Y}_{\mathrm{bad}}|\mathbf{x}) \\
=\ & \delta - 2\pi_{\theta(t)}(\mathcal{Y}_{\mathrm{bad}}|\mathbf{x}).
\end{aligned}
$$

Lemma 10, along with the assumed upper bounds on $\lambda$ and $\pi_{\theta(0)}(\mathcal{Y}_{\mathrm{bad}}|\mathbf{x})$, implies that:

$$
\begin{aligned}
\pi_{\theta(t)}(\mathcal{Y}_{\mathrm{bad}}|\mathbf{x}) &\leq \pi_{\theta(0)}(\mathcal{Y}_{\mathrm{bad}}|\mathbf{x}) \cdot \exp\left(\frac{1}{|\mathcal{S}|}\left(4 + 2\lambda \ln \frac{1}{\pi_{\theta(0)}(\mathbf{y}^m|\mathbf{x})}\right) \cdot T\right) \\
&\leq \pi_{\theta(0)}(\mathcal{Y}_{\mathrm{bad}}|\mathbf{x}) \cdot \exp\left(\frac{5}{|\mathcal{S}|} \cdot T\right) \\
&= \pi_{\theta(0)}(\mathcal{Y}_{\mathrm{bad}}|\mathbf{x}) \cdot \exp\left(\frac{20|\mathcal{Y}^+|}{(1 - \rho)\delta}\left(\frac{1}{\pi_{\theta(0)}(\mathcal{Y}^+|\mathbf{x})} - \frac{1}{\rho}\right)\right) \\
&\leq \frac{\pi_{\theta(0)}(\mathcal{Y}^+|\mathbf{x})^2(1 - \rho)\delta}{16|\mathcal{Y}^+|} \\
&\leq \frac{\delta}{16} \\
&\leq \frac{\delta}{4},
\end{aligned}
$$

where the second transition is due to:

$$
\begin{aligned}
2\lambda \ln \frac{1}{\pi_{\theta(0)}(\mathbf{y}^m|\mathbf{x})} &\leq \frac{\pi_{\theta(0)}(\mathcal{Y}^+|\mathbf{x})(1 - \rho)\delta}{4|\mathcal{Y}^+|\left(2\ln \frac{1}{\pi_{\theta(0)}(\mathbf{y}^m|\mathbf{x})} + \frac{1}{2e}\right)} \ln \frac{1}{\pi_{\theta(0)}(\mathbf{y}^m|\mathbf{x})} \\
&\leq \frac{\pi_{\theta(0)}(\mathcal{Y}^+|\mathbf{x})(1 - \rho)\delta}{8|\mathcal{Y}^+|} \\
&\leq 1.
\end{aligned}
\tag{14}
$$

Note that the inequality above used the fact that $\delta \leq 2$. Thus:

$$a(\mathbf{y}) \geq \delta - 2\pi_{\theta(t)}(\mathcal{Y}_{\text{bad}}|\mathbf{x}) \geq \frac{\delta}{2}.$$

Next, we upper bound $b(\mathbf{y}')$ from Equation (13) for $\mathbf{y}' \in \mathcal{Y} \setminus \mathcal{Y}^+$. Suppose first that $\mathbf{y}' \notin \mathcal{Y}_{\text{bad}}$, in which case $r_{\text{RM}}(\mathbf{x}, \mathbf{y}') \leq V_{\text{RM}}(\theta(0); \mathbf{x})$. We claim that $V_{\text{RM}}(\theta(0); \mathbf{x}) \leq V_{\text{RM}}(\theta(t); \mathbf{x})$. To see it is so, notice that under gradient flow it holds that:

$$\frac{d}{dt}\phi_{\text{RLHF}}(\theta(t); \mathbf{x}) = \frac{1}{|\mathcal{S}|}\big\|\nabla_{\theta_{:,\mathbf{x}}}\phi_{\text{RLHF}}(\theta(t); \mathbf{x})\big\|^2 \geq 0,$$

*i.e.*, $\phi_{\text{RLHF}}(\theta(t); \mathbf{x})$ is monotonically non-decreasing. Hence:

$$\begin{aligned}
V_{\text{RM}}(\theta(t); \mathbf{x}) - \lambda \cdot \text{KL}\big(\pi_{\theta(t)}(\cdot|\mathbf{x})||\pi_{\theta(0)}(\cdot|\mathbf{x})\big) &= \phi_{\text{RLHF}}(\theta(t); \mathbf{x}) \\
&\geq \phi_{\text{RLHF}}(\theta(0); \mathbf{x}) \\
&= V_{\text{RM}}(\theta(0); \mathbf{x}).
\end{aligned}$$

Since the KL divergence between any two distributions is always non-negative, this indeed implies that $V_{\text{RM}}(\theta(0); \mathbf{x}) \leq V_{\text{RM}}(\theta(t); \mathbf{x})$. Thus, $r_{\text{RM}}(\mathbf{x}, \mathbf{y}') \leq V_{\text{RM}}(\theta(0); \mathbf{x}) \leq V_{\text{RM}}(\theta(t); \mathbf{x})$, from which we conclude that in the case of $\mathbf{y}' \notin \mathcal{Y}_{\text{bad}}$:

$$b(\mathbf{y}') \leq 0.$$

As a result, we can discard these terms when lower bounding $\frac{d}{dt}\pi_{\theta(t)}(\mathcal{Y}^+|\mathbf{x})$. Otherwise, if $\mathbf{y}' \in \mathcal{Y}_{\text{bad}}$, since $r_{\text{RM}}(\mathbf{x}, \mathbf{y}') - V_{\text{RM}}(\theta(t); \mathbf{x}) \leq 2$ we have that:

$$b(\mathbf{y}') \leq 2\pi_{\theta(t)}(\mathbf{y}'|\mathbf{x})^2 \leq 2\pi_{\theta(t)}(\mathbf{y}'|\mathbf{x}).$$

Bounding each of the $a(\mathbf{y})$ and $b(\mathbf{y}')$ terms in Equation (13), and using the fact that $\pi_{\theta(t)}(\mathcal{Y}^+|\mathbf{x}) \leq \rho$, we arrive at:

$$\begin{aligned}
\frac{d}{dt}\pi_{\theta(t)}(\mathcal{Y}^+|\mathbf{x}) &\geq \frac{1}{|\mathcal{S}|}\sum_{\mathbf{y}\in\mathcal{Y}^+}\pi_{\theta(t)}(\mathbf{y}|\mathbf{x})^2\frac{(1-\rho)\delta}{2} \\
&\quad - \frac{2\rho}{|\mathcal{S}|}\cdot\pi_{\theta(t)}(\mathcal{Y}_{\text{bad}}|\mathbf{x}) \\
&\quad - \frac{\lambda}{|\mathcal{S}|}\cdot\pi_{\theta(t)}(\mathcal{Y}^+|\mathbf{x})\left(2\ln\frac{1}{\pi_{\theta(0)}(\mathbf{y}^m|\mathbf{x})} + \frac{1}{2e}\right) \\
&\geq \frac{(1-\rho)\delta}{2|\mathcal{S}||\mathcal{Y}^+|}\cdot\pi_{\theta(t)}(\mathcal{Y}^+|\mathbf{x})^2 \\
&\quad - \frac{2}{|\mathcal{S}|}\cdot\pi_{\theta(t)}(\mathcal{Y}_{\text{bad}}|\mathbf{x}) \\
&\quad - \frac{\lambda}{|\mathcal{S}|}\cdot\pi_{\theta(t)}(\mathcal{Y}^+|\mathbf{x})\left(2\ln\frac{1}{\pi_{\theta(0)}(\mathbf{y}^m|\mathbf{x})} + \frac{1}{2e}\right),
\end{aligned} \tag{15}$$

where the last transition is due to $\rho < 1$ and

$$\sum_{\mathbf{y}\in\mathcal{Y}^+}\pi_{\theta(t)}(\mathbf{y}|\mathbf{x})^2 \geq |\mathcal{Y}^+|\left(\frac{1}{|\mathcal{Y}^+|}\sum_{\mathbf{y}\in\mathcal{Y}^+}\pi_{\theta(t)}(\mathbf{y}|\mathbf{x})\right)^2 = \frac{1}{|\mathcal{Y}^+|}\pi_{\theta(t)}\big(\mathcal{Y}^+|\mathbf{x}\big)^2,$$

which follows from Jensen's inequality. Now, as shown in Equation (14):

$$2\lambda\ln\frac{1}{\pi_{\theta(0)}(\mathbf{y}^m|\mathbf{x})} \leq 1.$$

Combined with the assumed upper bound on $\pi_{\theta(0)}(\mathcal{Y}_{\text{bad}}|\mathbf{x})$, this implies that:

$$\begin{aligned}
&\pi_{\theta(0)}(\mathcal{Y}_{\text{bad}}|\mathbf{x}) \\
&\leq \frac{\pi_{\theta(0)}(\mathcal{Y}^+|\mathbf{x})^2(1-\rho)\delta}{16|\mathcal{Y}^+|}\exp\left(-\frac{20|\mathcal{Y}^+|}{(1-\rho)\delta}\left(\frac{1}{\pi_{\theta(0)}(\mathcal{Y}^+|\mathbf{x})} - \frac{1}{\rho}\right)\right) \\
&\leq \frac{\pi_{\theta(0)}(\mathcal{Y}^+|\mathbf{x})^2(1-\rho)\delta}{16|\mathcal{Y}^+|}\exp\left(-\left(4 + 2\lambda\ln\frac{1}{\pi_{\theta(0)}(\mathbf{y}^m|x)}\right)\frac{4|\mathcal{Y}^+|}{(1-\rho)\delta}\left(\frac{1}{\pi_{\theta(0)}(\mathcal{Y}^+|\mathbf{x})} - \frac{1}{\rho}\right)\right).
\end{aligned}$$

Going back to Equation (15), invoking Lemma 10 for upper bounding $\pi_{\theta(t)}(\mathcal{Y}_{\text{bad}}|\mathbf{x})$, applying the assumed upper bound on $\lambda$, and recalling that

$$t \leq T = \frac{4|\mathcal{S}||\mathcal{Y}^+|}{(1-\rho)\delta}\left(\frac{1}{\pi_{\theta(0)}(\mathcal{Y}^+|\mathbf{x})} - \frac{1}{\rho}\right)$$

gives:

$$\frac{d}{dt}\pi_{\theta(t)}(\mathcal{Y}^+|\mathbf{x}) \geq \pi_{\theta(t)}(\mathcal{Y}^+|\mathbf{x})^2 \cdot \frac{(1-\rho)\delta}{2|\mathcal{S}||\mathcal{Y}^+|} - \pi_{\theta(0)}(\mathcal{Y}^+|\mathbf{x})^2 \cdot \frac{(1-\rho)\delta}{8|\mathcal{S}||\mathcal{Y}^+|}$$
$$- \pi_{\theta(0)}(\mathcal{Y}^+|\mathbf{x})\pi_{\theta(t)}(\mathcal{Y}^+|\mathbf{x}) \cdot \frac{(1-\rho)\delta}{8|\mathcal{S}||\mathcal{Y}^+|} \, .$$

At time $t = 0$ it therefore holds that $\frac{d}{dt}\big(\pi_{\theta(t)}(\mathcal{Y}^+|\mathbf{x})\big)|_{t=0} \geq \pi_{\theta(0)}(\mathcal{Y}^+|\mathbf{x})^2 \cdot \frac{(1-\rho)\delta}{4|\mathcal{S}||\mathcal{Y}^+|} > 0$. Moreover, since $\frac{d}{dt}\pi_{\theta(t)}(\mathcal{Y}^+|\mathbf{x})$ is positive whenever $\pi_{\theta(t)}(\mathcal{Y}^+|\mathbf{x}) \geq \pi_{\theta(0)}(\mathcal{Y}^+|\mathbf{x})$, this implies that $\pi_{\theta(t)}(\mathcal{Y}^+|\mathbf{x})$ is monotonically increasing for all $t \in [0, T]$. Thus, $\pi_{\theta(0)}(\mathcal{Y}^+|\mathbf{x}) \leq \pi_{\theta(t)}(\mathcal{Y}^+|\mathbf{x})$ and:

$$\frac{d}{dt}\pi_{\theta(t)}(\mathcal{Y}^+|\mathbf{x}) \geq \pi_{\theta(t)}(\mathcal{Y}^+|\mathbf{x})^2 \cdot \frac{(1-\rho)\delta}{4|\mathcal{S}||\mathcal{Y}^+|} \, .$$

Finally, applying Lemma 2 we have that for all $t \in [0, T]$:

$$\pi_{\theta(t)}(\mathcal{Y}^+|\mathbf{x}) \geq \frac{1}{\frac{1}{\pi_{\theta(0)}(\mathcal{Y}^+|\mathbf{x})} - \frac{(1-\rho)\delta}{4|\mathcal{S}||\mathcal{Y}^+|} \cdot t} \, .$$

For

$$t = T = \frac{4|\mathcal{S}||\mathcal{Y}^+|}{(1-\rho)\delta}\left(\frac{1}{\pi_{\theta(0)}(\mathcal{Y}^+|\mathbf{x})} - \frac{1}{\rho}\right),$$

the inequality above implies that $\pi_{\theta(t)}(\mathcal{Y}^+|\mathbf{x}) \geq \rho$. We have therefore reached a contradiction — as shown in Equation (12), when $\pi_{\theta(t)}(\mathcal{Y}^+|\mathbf{x}) \geq \rho$ the expected ground truth reward is higher by at least $\gamma$ than its initial value, and so $t_\gamma$ must be at most $T$. $\qquad\square$

## D.8 Proof of Theorem 5

The proof assumes familiarity with the notation introduced in Appendix D.1.

Define $r_{\text{RM}}, r'_{\text{RM}} : \mathcal{X} \times \mathcal{Y} \to [-1, 1]$ as follows. For the prompt $\mathbf{x}$ and $\mathbf{y} \in \mathcal{Y}$:

$$r_{\text{RM}}(\mathbf{x}, \mathbf{y}) := b_{\mathbf{y}} \quad , \quad r'_{\text{RM}}(\mathbf{x}, \mathbf{y}) := \begin{cases} 1 & , \mathbf{y} = \mathbf{y}^\gamma \\ c_{\mathbf{y}} & , \mathbf{y} \in \mathcal{Y} \setminus \{\mathbf{y}^\gamma\} \end{cases},$$

where $0 < b_{\mathbf{y}} < T^{-1}$, for all $\mathbf{y} \in \mathcal{Y}$, are chosen such that their ordering achieves the desired attainable accuracy $\alpha \in [0, 1]$, and $0 < c_{\mathbf{y}} < \pi_{\theta(0)}(\mathbf{y}^\gamma|\mathbf{x})$, for all $\mathbf{y} \in \mathcal{Y} \setminus \{\mathbf{y}^\gamma\}$, with the exact values of $\{c_{\mathbf{y}}\}_{\mathbf{y}\in\mathcal{Y}\setminus\{\mathbf{y}^\gamma\}}$ chosen such that the desired attainable accuracy subject to $\mathbf{y}^\gamma$ being ranked first $\alpha'$ is achieved by $r'_{\text{RM}}$. For the remaining prompts $\mathbf{x}' \neq \mathbf{x}$, the reward models $r_{\text{RM}}, r'_{\text{RM}}$ can be defined arbitrarily since, under a tabular policy (Equation (3)), $\pi_{\theta(t)}(\cdot|\mathbf{x})$ does not depend on $\theta_{:,\mathbf{x}'}(t)$. We denote $V_{\text{RM}}(\theta(t); \mathbf{x}) := \mathbb{E}_{\mathbf{y}\sim\pi_{\theta(t)}(\cdot|\mathbf{x})}[r_{\text{RM}}(\mathbf{x}, \mathbf{y})]$ and $V'_{\text{RM}}(\theta(t); \mathbf{x}) := \mathbb{E}_{\mathbf{y}\sim\pi_{\theta(t)}(\cdot|\mathbf{x})}[r'_{\text{RM}}(\mathbf{x}, \mathbf{y})]$.

**Remark 3.** For simplicity, we defined $r_{\text{RM}}$ so that it induces low reward variance for any policy. However, it is also possible to define $r_{\text{RM}}$ such that for $\pi_{\theta(0)}$ it induces low reward variance, yet for other policies it induces a high reward variance (*e.g.*, by having it assign a high reward to an output that is not likely under $\pi_{\theta(0)}(\cdot|\mathbf{x})$). Furthermore, the rewards produced by $r_{\text{RM}}$ need not be near-zero and can be shifted by any constant.

**Proof of slow ground truth reward increase under $r_{\text{RM}}$.** Suppose that gradient flow is used to maximize the RLHF objective with respect to the reward model $r_{\text{RM}}$. The lower bound on $t_\gamma$ stems from the reward variance induced by $r_{\text{RM}}$ for the initial policy $\pi_{\theta(0)}$ being low. Namely:

$$\text{Var}_{\mathbf{y}\sim\pi_{\theta(0)}(\cdot|\mathbf{x})}[r_{\text{RM}}(\mathbf{x}, \mathbf{y})] = \sum\nolimits_{\mathbf{y}\in\mathcal{Y}} \pi_{\theta(0)}(\mathbf{y}|\mathbf{x})[r_{\text{RM}}(\mathbf{x}, \mathbf{y}) - V_{\text{RM}}(\theta(0); \mathbf{x})]^2$$
$$= \sum\nolimits_{\mathbf{y}\in\mathcal{Y}} \pi_{\theta(0)}(\mathbf{y}|\mathbf{x})\Big[\sum\nolimits_{\mathbf{y}'\in\mathcal{Y}} \pi_{\theta(0)}(\mathbf{y}'|\mathbf{x})(b_{\mathbf{y}} - b_{\mathbf{y}'})\Big]^2$$
$$\leq \frac{1}{T^2},$$

where the inequality is due to $|b_\mathbf{y} - b_{\mathbf{y}'}| \leq T^{-1}$ for all $\mathbf{y}, \mathbf{y}' \in \mathcal{Y}$. Proposition 3 then implies:

$$t_\gamma \geq 2|\mathcal{S}|\left(1 - \exp\left(-\frac{\gamma}{2}\right)\right) \cdot T = \Omega(T).$$

**Proof of fast ground truth reward increase under $r'_{\mathrm{RM}}$.** Suppose that gradient flow is used to maximize the RLHF objective with respect to the reward model $r'_{\mathrm{RM}}$. We will use Proposition 4 for upper bounding $t_\gamma$. To that end, define $\mathcal{Y}^+ := \{\mathbf{y}^\gamma\}$ and notice that it satisfies the conditions laid out in Proposition 4. Namely, $r_{\mathrm{G}}(\mathbf{x}, \mathbf{y}^\gamma) > V_{\mathrm{G}}(\theta(0); \mathbf{x}) + \gamma$ and $r'_{\mathrm{RM}}(\mathbf{x}, \mathbf{y}^\gamma) = 1 > V'_{\mathrm{RM}}(\theta(0); \mathbf{x})$ (the latter trivially holds since the maximal reward is one and $0 < \pi_{\theta(0)}(\mathbf{y}|\mathbf{x}) < 1$ for any $\mathbf{y} \in \mathcal{Y}$, including outputs $\mathbf{y} \in \mathcal{Y} \setminus \{\mathbf{y}^\gamma\}$ whose reward under $r'_{\mathrm{RM}}$ is not maximal). Furthermore:

$$
\begin{aligned}
\delta &:= (1-\rho)\big(\max_{\mathbf{y}\in\mathcal{Y}^+} r'_{\mathrm{RM}}(\mathbf{x}, \mathbf{y}) - V'_{\mathrm{RM}}(\theta(0); \mathbf{x})\big) - \max_{\mathbf{y},\mathbf{y}'\in\mathcal{Y}^+} |r'_{\mathrm{RM}}(\mathbf{x}, \mathbf{y}) - r'_{\mathrm{RM}}(\mathbf{x}, \mathbf{y}')| \\
&= (1-\rho)(1 - V'_{\mathrm{RM}}(\theta(0); \mathbf{x})) \\
&\geq (1-\rho)\big(1 - \pi_{\theta(0)}(\mathbf{y}^\gamma|\mathbf{x}) - \big(1 - \pi_{\theta(0)}(\mathbf{y}^\gamma|\mathbf{x})\big)\pi_{\theta(0)}(\mathbf{y}^\gamma|\mathbf{x})\big) \\
&= (1-\rho)\big(1 - \pi_{\theta(0)}(\mathbf{y}^\gamma|\mathbf{x})\big)^2,
\end{aligned}
$$

where the second transition is due to $\mathcal{Y}^+$ containing only a single output and $r'_{\mathrm{RM}}(\mathbf{x}, \mathbf{y}^\gamma) = 1$, and the inequality is due to $r'_{\mathrm{RM}}(\mathbf{x}, \mathbf{y}) = c_\mathbf{y} < \pi_{\theta(0)}(\mathbf{y}^\gamma|\mathbf{x})$ for all $\mathbf{y} \in \mathcal{Y} \setminus \{\mathbf{y}^\gamma\}$. Recall:

$$\rho := \frac{V_{\mathrm{G}}(\theta(0); \mathbf{x}) + \gamma + 1}{r_{\mathrm{G}}(\mathbf{x}, \mathbf{y}^\gamma) + 1} \in (0, 1).$$

Thus, necessarily $\pi_{\theta(0)}(\mathbf{y}^\gamma|\mathbf{x}) < \rho$, as otherwise, if $\pi_{\theta(0)}(\mathbf{y}^\gamma|\mathbf{x}) \geq \rho$, we get a contradiction:

$$
\begin{aligned}
V_{\mathrm{G}}(\theta(0); \mathbf{x}) &\geq \rho \cdot r_{\mathrm{G}}(\mathbf{x}, \mathbf{y}^\gamma) + \sum\nolimits_{\mathbf{y}\in\mathcal{Y}\setminus\{\mathbf{y}^\gamma\}} \pi_{\theta(0)}(\mathbf{y}|\mathbf{x}) r_{\mathrm{G}}(\mathbf{x}, \mathbf{y}) \\
&\geq \rho \cdot r_{\mathrm{G}}(\mathbf{x}, \mathbf{y}^\gamma) - \pi_{\theta(0)}(\mathcal{Y} \setminus \{\mathbf{y}^\gamma\}|\mathbf{x}) \\
&\geq \rho \cdot r_{\mathrm{G}}(\mathbf{x}, \mathbf{y}^\gamma) - (1 - \rho) \\
&= \rho(r_{\mathrm{G}}(\mathbf{x}, \mathbf{y}^\gamma) + 1) - 1 \\
&= V_{\mathrm{G}}(\theta(0); \mathbf{x}) + \gamma.
\end{aligned}
$$

This implies that $\delta \geq (1-\rho)^3 > 0$. Lastly, $\mathcal{Y}_{\mathrm{bad}} := \{\mathbf{y} \in \mathcal{Y} : r'_{\mathrm{RM}}(\mathbf{x}, \mathbf{y}) > V'_{\mathrm{RM}}(\theta(0); \mathbf{x})\} \setminus \mathcal{Y}^+ = \emptyset$ since for all $\mathbf{y} \in \mathcal{Y} \setminus \mathcal{Y}^+$ it holds that $r'_{\mathrm{RM}}(\mathbf{x}, \mathbf{y}) = c_\mathbf{y} < \pi_{\theta(0)}(\mathbf{y}^\gamma|\mathbf{x}) \leq V'_{\mathrm{RM}}(\theta(0); \mathbf{x})$. Thus, $\mathcal{Y}^+, \delta, \lambda$, and $\mathcal{Y}_{\mathrm{bad}}$ uphold the requirements of Proposition 4. Invoking it concludes the proof:

$$
\begin{aligned}
t_\gamma &\leq \frac{4|\mathcal{S}|}{(1-\rho)\delta} \cdot \left(\frac{1}{\pi_{\theta(0)}(\mathbf{y}^\gamma|\mathbf{x})} - \frac{1}{\rho}\right) \\
&\leq \frac{4|\mathcal{S}|}{(1-\rho)^4} \cdot \left(\frac{1}{\pi_{\theta(0)}(\mathbf{y}^\gamma|\mathbf{x})} - \frac{1}{\rho}\right) \\
&= \mathcal{O}\left(\frac{1}{\pi_{\theta(0)}(\mathbf{y}^\gamma|\mathbf{x})}\right),
\end{aligned}
$$

where the second inequality is by $\delta \geq (1-\rho)^3$. $\qquad\square$

## D.9 Proof of Theorem 6

The proof assumes familiarity with the notation introduced in Appendix D.1.

Define $r_{\mathrm{RM}}, r'_{\mathrm{RM}} : \mathcal{X} \times \mathcal{Y} \to [-1, 1]$ as follows. For the prompt $\mathbf{x}$ and $\mathbf{y} \in \mathcal{Y}$:

$$
r_{\mathrm{RM}}(\mathbf{x}, \mathbf{y}) := \begin{cases} 1 & , \mathbf{y} = \mathbf{y}^\gamma \\ -1 & , \mathbf{y} \in \mathcal{Y} \setminus \{\mathbf{y}^\gamma\} \end{cases} \quad , \quad r'_{\mathrm{RM}}(\mathbf{x}, \mathbf{y}) := \begin{cases} 1 & , \mathbf{y} = \mathbf{y}'^\gamma \\ -1 & , \mathbf{y} \in \mathcal{Y} \setminus \{\mathbf{y}'^\gamma\} \end{cases}.
$$

For the remaining prompts $\mathbf{x}' \neq \mathbf{x}$, the reward models $r_{\mathrm{RM}}, r'_{\mathrm{RM}}$ can be defined arbitrarily since, under a tabular policy (Equation (3)), $\pi_{\theta(t)}(\cdot|\mathbf{x})$ does not depend on $\theta_{:,\mathbf{x}'}(t)$. Denote $V_{\mathrm{RM}}(\theta(t); \mathbf{x}) := \mathbb{E}_{\mathbf{y}\sim\pi_{\theta(t)}(\cdot|\mathbf{x})}[r_{\mathrm{RM}}(\mathbf{x}, \mathbf{y})]$ and $V'_{\mathrm{RM}}(\theta(t); \mathbf{x}) := \mathbb{E}_{\mathbf{y}\sim\pi_{\theta(t)}(\cdot|\mathbf{x})}[r'_{\mathrm{RM}}(\mathbf{x}, \mathbf{y})]$.

**Proof of fast ground truth reward increase under $r_{\mathrm{RM}}$ for $\pi_{\theta(0)} \in \Pi$.** Suppose that gradient flow, initialized at $\pi_{\theta(0)} \in \Pi$, is used to maximize the RLHF objective with respect to the reward model $r_{\mathrm{RM}}$. We use Proposition 4 for upper bounding $t_\gamma$. To that end, define $\mathcal{Y}^+ := \{\mathbf{y}^\gamma\}$ and notice that it satisfies the conditions laid out in Proposition 4. Namely, $r_{\mathrm{G}}(\mathbf{x}, \mathbf{y}^\gamma) > V_{\mathrm{G}}(\theta(0); \mathbf{x}) + \gamma$ and $r_{\mathrm{RM}}(\mathbf{x}, \mathbf{y}^\gamma) = 1 > V_{\mathrm{RM}}(\theta(0); \mathbf{x})$ (the latter trivially holds since the maximal reward is one and $0 < \pi_{\theta(0)}(\mathbf{y}|\mathbf{x}) < 1$ for any $\mathbf{y} \in \mathcal{Y}$, including outputs $\mathbf{y} \in \mathcal{Y} \setminus \{\mathbf{y}^\gamma\}$ whose reward under $r_{\mathrm{RM}}$ is not maximal). Observe that:

$$
\begin{aligned}
\delta &:= (1 - \rho)\big(\max_{\mathbf{y} \in \mathcal{Y}^+} r_{\mathrm{RM}}(\mathbf{x}, \mathbf{y}) - V_{\mathrm{RM}}(\theta(0); \mathbf{x})\big) - \max_{\mathbf{y}, \mathbf{y}' \in \mathcal{Y}^+} |r_{\mathrm{RM}}(\mathbf{x}, \mathbf{y}) - r_{\mathrm{RM}}(\mathbf{x}, \mathbf{y}')| \\
&= (1 - \rho)(1 - V_{\mathrm{RM}}(\theta(0); \mathbf{x})) \\
&= (1 - \rho)\Big(\sum\nolimits_{\mathbf{y} \in \mathcal{Y}} \pi_{\theta(0)}(\mathbf{y}|\mathbf{x})(1 - r_{\mathrm{RM}}(\mathbf{x}, \mathbf{y}))\Big) \\
&= 2(1 - \rho) \cdot \pi_{\theta(0)}(\mathcal{Y} \setminus \{\mathbf{y}^\gamma\}|\mathbf{x}),
\end{aligned}
$$

where the second transition is due to $\mathcal{Y}^+$ containing only a single output and $r_{\mathrm{RM}}(\mathbf{x}, \mathbf{y}^\gamma) = 1$, and the last transition is due to $r_{\mathrm{RM}}(\mathbf{x}, \mathbf{y}) = -1$ for all $\mathbf{y} \in \mathcal{Y} \setminus \{\mathbf{y}^\gamma\}$. Now, recall:

$$
\rho := \frac{V_{\mathrm{G}}(\theta(0); \mathbf{x}) + \gamma + 1}{r_{\mathrm{G}}(\mathbf{x}, \mathbf{y}^\gamma) + 1} \in (0, 1).
$$

We claim that $\pi_{\theta(0)}(\mathbf{y}^\gamma|\mathbf{x}) < \rho$. Otherwise, if $\pi_{\theta(0)}(\mathbf{y}^\gamma|\mathbf{x}) \geq \rho$ we have a contradiction:

$$
\begin{aligned}
V_{\mathrm{G}}(\theta(0); \mathbf{x}) &\geq \rho \cdot r_{\mathrm{G}}(\mathbf{x}, \mathbf{y}^\gamma) + \sum\nolimits_{\mathbf{y} \in \mathcal{Y} \setminus \{\mathbf{y}^\gamma\}} \pi_{\theta(0)}(\mathbf{y}|\mathbf{x}) r_{\mathrm{G}}(\mathbf{x}, \mathbf{y}) \\
&= \rho \cdot r_{\mathrm{G}}(\mathbf{x}, \mathbf{y}^\gamma) - \pi_{\theta(0)}(\mathcal{Y} \setminus \{\mathbf{y}^\gamma\}|\mathbf{x}) \\
&\geq \rho \cdot r_{\mathrm{G}}(\mathbf{x}, \mathbf{y}^\gamma) - (1 - \rho) \\
&= \rho(r_{\mathrm{G}}(\mathbf{x}, \mathbf{y}^\gamma) + 1) - 1 \\
&= V_{\mathrm{G}}(\theta(0); \mathbf{x}) + \gamma.
\end{aligned}
$$

Thus, $\pi_{\theta(0)}(\mathcal{Y} \setminus \{\mathbf{y}^\gamma\}|\mathbf{x}) \geq 1 - \rho$, which implies that:

$$
\delta \geq 2(1 - \rho)^2.
$$

Finally, $\mathcal{Y}_{\mathrm{bad}} := \{\mathbf{y} \in \mathcal{Y} : r_{\mathrm{RM}}(\mathbf{x}, \mathbf{y}) > V_{\mathrm{RM}}(\theta(0); \mathbf{x})\} \setminus \mathcal{Y}^+ = \emptyset$ since for all $\mathbf{y} \in \mathcal{Y} \setminus \mathcal{Y}^+$ it holds that $r_{\mathrm{RM}}(\mathbf{x}, \mathbf{y}) = -1 \leq V_{\mathrm{RM}}(\theta(0); \mathbf{x})$. Thus, $\mathcal{Y}^+, \delta, \lambda$, and $\mathcal{Y}_{\mathrm{bad}}$ uphold the requirements of Proposition 4. Invoking it, we conclude:

$$
\begin{aligned}
t_\gamma &\leq \frac{4|\mathcal{S}|}{(1 - \rho)\delta} \cdot \left(\frac{1}{\pi_{\theta(0)}(\mathbf{y}^\gamma|\mathbf{x})} - \frac{1}{\rho}\right) \\
&\leq \frac{2|\mathcal{S}|}{(1 - \rho)^3} \cdot \left(\frac{1}{c} - \frac{1}{\rho}\right) \\
&= \mathcal{O}(1).
\end{aligned}
$$

**Proof of slow ground truth reward increase under $r'_{\mathrm{RM}}$ for $\pi_{\theta(0)} \in \Pi$.** Suppose that gradient flow, initialized at $\pi_{\theta(0)} \in \Pi$, is used to maximize the RLHF objective with respect to the reward model $r'_{\mathrm{RM}}$. The lower bound on $t_\gamma$ follows by showing that $r'_{\mathrm{RM}}$ induces low reward variance for $\pi_{\theta(0)}$. Since $\pi_{\theta(0)}(\mathbf{y}'^\gamma|\mathbf{x}) \leq T^{-2}$ and $|r'_{\mathrm{RM}}(\mathbf{x}, \mathbf{y}) - V'_{\mathrm{RM}}(\theta(0); \mathbf{x})| \leq 2$ for all $\mathbf{y} \in \mathcal{Y}$, we may write:

$$
\begin{aligned}
\mathrm{Var}_{\mathbf{y} \sim \pi_{\theta(0)}(\cdot|\mathbf{x})}\big[r'_{\mathrm{RM}}(\mathbf{x}, \mathbf{y})\big] &= \sum\nolimits_{\mathbf{y} \in \mathcal{Y}} \pi_{\theta(0)}(\mathbf{y}|\mathbf{x})\big[r'_{\mathrm{RM}}(\mathbf{x}, \mathbf{y}) - V'_{\mathrm{RM}}(\theta(0); \mathbf{x})\big]^2 \\
&\leq \frac{4}{T^2} + \sum\nolimits_{\mathbf{y} \in \mathcal{Y} \setminus \{\mathbf{y}'^\gamma\}} \pi_{\theta(0)}(\mathbf{y}|\mathbf{x})[r'_{\mathrm{RM}}(\mathbf{x}, \mathbf{y}) - V'_{\mathrm{RM}}(\theta(0); \mathbf{x})]^2.
\end{aligned}
$$
(16)

For every $\mathbf{y} \in \mathcal{Y} \setminus \{\mathbf{y}'^\gamma\}$ we have that:

$$
\begin{aligned}
|r'_{\mathrm{RM}}(\mathbf{x}, \mathbf{y}) - V'_{\mathrm{RM}}(\theta(0); \mathbf{x})| &= \left|\sum\nolimits_{\mathbf{y}' \in \mathcal{Y}} \pi_{\theta(0)}(\mathbf{y}'|\mathbf{x})(r'_{\mathrm{RM}}(\mathbf{x}, \mathbf{y}) - r'_{\mathrm{RM}}(\mathbf{x}, \mathbf{y}'))\right| \\
&\leq \left|\sum\nolimits_{\mathbf{y}' \in \mathcal{Y} \setminus \{\mathbf{y}'^\gamma\}} \pi_{\theta(0)}(\mathbf{y}'|\mathbf{x})(r'_{\mathrm{RM}}(\mathbf{x}, \mathbf{y}) - r'_{\mathrm{RM}}(\mathbf{x}, \mathbf{y}'))\right| + \frac{2}{T^2} \\
&= \frac{2}{T^2},
\end{aligned}
$$

where the second transition is by the triangle inequality, the fact that $|r'_{\mathrm{RM}}(\mathbf{x},\mathbf{y}) - r'_{\mathrm{RM}}(\mathbf{x},\mathbf{y}')| \leq 2$ for all $\mathbf{y},\mathbf{y}' \in \mathcal{Y}$, and $\pi_{\theta(0)}(\mathbf{y}'^{\gamma}|\mathbf{x}) \leq T^{-2}$, and the last transition is due to the $r'_{\mathrm{RM}}(\mathbf{x},\mathbf{y}) = r'_{\mathrm{RM}}(\mathbf{x},\mathbf{y}') = -1$ for all $\mathbf{y},\mathbf{y}' \in \mathcal{Y} \setminus \{\mathbf{y}'^{\gamma}\}$.

Going back to Equation (16), we get that:

$$\mathrm{Var}_{y \sim \pi_{\theta(0)}(\cdot|x)}\big[r'_{\mathrm{RM}}(\mathbf{x},\mathbf{y})\big] \leq \frac{4}{T^2} + \sum\nolimits_{\mathbf{y} \in \mathcal{Y} \setminus \{\mathbf{y}'^{\gamma}\}} \pi_{\theta(0)}(\mathbf{y}|\mathbf{x}) \cdot \frac{4}{T^4}$$
$$\leq \frac{4}{T^2} + \frac{4}{T^4}$$
$$\leq \frac{8}{T^2},$$

where the second inequality is due to $\sum_{\mathbf{y} \in \mathcal{Y} \setminus \{\mathbf{y}'^{\gamma}\}} \pi_{\theta(0)}(\mathbf{y}|\mathbf{x}) \leq 1$ and in the last we used $T^4 \geq T^2$ to simplify terms (recall $T \geq 1$). Applying Proposition 3 we may conclude:

$$t_{\gamma} \geq 2|\mathcal{S}|\Big(1 - \exp\Big(-\frac{\gamma}{2}\Big)\Big) \cdot \frac{T}{\sqrt{8}} = \frac{|\mathcal{S}|}{\sqrt{2}}\Big(1 - \exp\Big(-\frac{\gamma}{2}\Big)\Big) \cdot T = \Omega(T).$$

**Proof of fast ground truth reward increase under $r'_{\mathrm{RM}}$ for $\pi_{\theta(0)} \in \Pi'$.** The proof follows by steps analogous to those used for proving fast ground truth reward increase under $r_{\mathrm{RM}}$ for $\pi_{\theta(0)} \in \Pi$.

**Proof of slow ground truth reward increase under $r_{\mathrm{RM}}$ for $\pi_{\theta(0)} \in \Pi'$.** The proof follows by steps analogous to those used for proving slow ground truth reward increase under $r'_{\mathrm{RM}}$ for $\pi_{\theta(0)} \in \Pi$. $\quad\square$

## E   Additional Experiments

### E.1   More Accurate Reward Models Are Not Necessarily Better Teachers (Section 4.1)

Listed below are additional experiments, omitted from Section 4.1.

- Figure 4 supplements Figure 2 by plotting the increase in rewards against the KL divergence from the initial policy, instead of against the number of epochs.
- Table 3, Table 4, Figure 5, and Figure 6 are analogous to Table 1, Table 2, Figure 2, and Figure 4, respectively, reporting the results of identical experiments, except that the initial policy was a Pythia-1B language model instead of a Pythia-2.8B model.
- Table 5, Table 6, Figure 7, and Figure 8 are analogous to Table 1, Table 2, Figure 2, and Figure 4, respectively, reporting the results of identical experiments, except that: *(i)* the initial policy was a Pythia-1B language model instead of a Pythia-2.8B model; and *(ii)* SFT was carried out over the (UltraFeedback-based) reward model training set instead of AlpacaFarm.
- Table 7, Table 8, Figure 9, and Figure 10 are analogous to Table 1, Table 2, Figure 2, and Figure 4, respectively, reporting the results of identical experiments, except that the ground truth reward model was GRM-gemma2-2B-rewardmodel-ft [87] instead of ArmoRM.
- Table 9, Figure 11, and Figure 12 are analogous to Table 4, Figure 5, and Figure 6, respectively, reporting the results of identical experiments, except that during training reward models were normalized using a different scheme. Specifically, each reward model was normalized by the square root of the average initial reward variance over prompts in the training set, instead of by the reward standard deviation taken across training prompts jointly (see Appendix F.1 for details on the original normalization scheme). These experiments demonstrate that the relationship between reward variance (as computed via the original normalization scheme that accounts for reward scale) and optimization persists under other reward normalization schemes.

### E.2   For Different Language Models, Different Reward Models Are Better (Section 4.2)

Listed below are additional experiments, omitted from Section 4.2.

- Table 10 supplements Figure 3 by reporting the reward variance and accuracy (both on-policy and off-policy) for each combination of reward model and initial policy therein.

- Figure 13 extends the experiment of Figure 3 by considering larger language models of roughly 3B parameters. Table 11 supplements Figure 13 by reporting reporting the reward variance and accuracy (both on-policy and off-policy) for each combination of reward model and initial policy therein.

- Figure 14 reports the results of an experiment identical to that of Figure 3, except that we used GRPO [69] instead of RLOO as the policy gradient method. Table 12 supplements Figure 14 by reporting reporting the reward variance and accuracy (both on-policy and off-policy) for each combination of reward model and initial policy therein.

# F    Additional Implementation Details

In this appendix, we provide implementation details omitted from Section 4 and Appendix E. Code for reproducing our results, based on the PyTorch [59] and Hugging Face TRL [81] frameworks, can be found at `https://github.com/princeton-pli/what-makes-good-rm`.

## F.1    More Accurate Reward Models Are Not Necessarily Better Teachers (Section 4.1)

**Data.** We took the binarized version of UltraFeedback [75], filtered out samples in which the prompt or one of the outputs exceeded 512 tokens according to a Pythia [8] tokenizer, and relabeled output preferences using the ground truth reward model. The resulting training and test sets had 41419 and 1329 samples, respectively. Then, we split the training set into two subsets: 80% of the samples (*i.e.*, 33135 samples) were used for reward model training and the rest (*i.e.*, 8284 samples) for the policy gradient step of RLHF. The test set was used for evaluating reward models and policies. Though, as mentioned in Section 4.2.1, since results over the training and test sets were nearly identical and our interest lies in optimization, which manifests in ability to maximize the reward over training prompts, we report only quantities computed over the training set.

For each prompt $\mathbf{x}$ and output $\mathbf{y}$, we used the following chat template (unless the model already had a chat template defined, in which case we used the original chat template):

$$[USER]\mathbf{x}[ASSISTANT]\mathbf{y}[EOS]$$

where [USER], [ASSISTANT], and [EOS] are defined as special tokens.

**Generation hyperparameters.** When generating outputs from a policy, we used a temperature of 1 and a maximum output length of 512 tokens

**SFT.** When performing SFT over AlpacaFarm [22], we took the corresponding split that contains 10000 samples. In the remaining experiments, we used the preferred outputs in the UltraFeedback-based reward model training set (using the original UltraFeedback outputs, but taking the output preferred by the ground truth reward model). We minimized the cross-entropy loss (as implemented by the TRL framework) for one epoch via the Adam optimizer [41] with a learning rate of 1e-6 and batch size of 32 (emulated via gradient accumulation steps). The learning rate was chosen to be large enough for the training loss to decrease yet not too large to ensure stability.

**Reward models.** We trained each reward model by minimizing the standard Bradley-Terry log-likelihood loss (as implemented by the TRL framework) for one epoch via the Adam optimizer with a learning rate of 5e-7 and batch size of 32 (emulated via gradient accumulation steps). The learning rate was chosen to be large enough for the training loss to decrease yet not too large to ensure stability. Following [23], we set the `center_rewards_coefficient` hyperparameter to 0.01. For creating the perfectly accurate reward model that induces a low reward variance, we took the ground truth reward model and scaled down the rewards that it produces for 50% of the prompts in the policy gradient training set by a factor of 0.001. We note that varying the percentage of prompts and downscaling factor did not significantly modify the trends observed in the experiments of Section 4.1.

**Reward normalization.** To ensure fair comparison across reward models, we normalized their rewards to be on the same scale. Specifically, we sampled 500 prompts from the policy gradient training set and generated using the initial policy 10 outputs per prompt. Then, for each reward model separately (as well as the ground truth reward), we computed the reward mean and standard deviation across all of the 5000 generated outputs. Lastly, during policy gradient training and evaluation, we shifted and normalized the rewards by the corresponding mean and standard deviation.

**Reward model evaluation.** For each reward model and initial policy, we evaluated on-policy accuracy and reward variance over the same 500 prompts used for reward normalization, where for each prompt 10 outputs were generated. For off-policy accuracy, we used the same 500 prompts, but the original UltraFeedback outputs instead of outputs sampled from the initial policy.

**Policy gradient.** Our RLOO and GRPO implementations are based on the RLOOTrainer class from the TRL framework, which uses the Adam optimizer. We set the learning rate to 1e-7, batch size to 32 (emulated via gradient accumulation steps), and the `num_mini_batches` hyperparameter to 2. We kept the KL coefficient at its default value of 0.05. Varying the learning rate or KL coefficient did not have a noticeable effect on the trends observed in Section 4. Though, as expected, too large learning rates led to instability while too small learning rates or a too high KL coefficient resulted in extremely slow progress. As in [2], for each prompt in a batch, we sampled two outputs (*i.e.*, we set the RLOO $k$ parameter to 2). According to the TRL framework terminology, this implies that a full pass over all prompts in the training set is completed every two epochs (*i.e.*, in TRL, an epoch is measured by the number of gradient updates as opposed to the number of prompts). Lastly, all policies were evaluated based on the same 500 prompts used for reward model evaluation.

**KL divergence from initial policy.** Each KL divergence value reported in Figures 4, 6, 8, and 10 is an average across the last 20 batches of an epoch.

**Hardware.** All experiments ran on Nvidia H100 GPUs with 80GB memory. For SFT and reward model training, we used a single GPU. For policy gradient (*i.e.*, RLOO and GRPO), we used two GPUs in runs with language models of roughly 1B parameters and four GPUs in runs with language models of roughly 3B parameters.

### F.2 For Different Language Models, Different Reward Models Are Better (Section 4.2)

The experimental setup was identical to that detailed in Appendix F.1, but instead of training reward model ourselves, we used the following publicly available reward models:

- `https://huggingface.co/Ray2333/GRM-Llama3.2-3B-rewardmodel-ft`,
- `https://huggingface.co/Ray2333/GRM-gemma2-2B-rewardmodel-ft`,
- `https://huggingface.co/allenai/llama-3-tulu-2-8b-uf-mean-rm`,
- `https://huggingface.co/weqweasdas/RM-Gemma-2B`.

Furthermore, we ran policy gradient for two epochs in this set of experiments (in accordance with the common practice of running policy gradient for only a few epochs [55, 37, 2]).

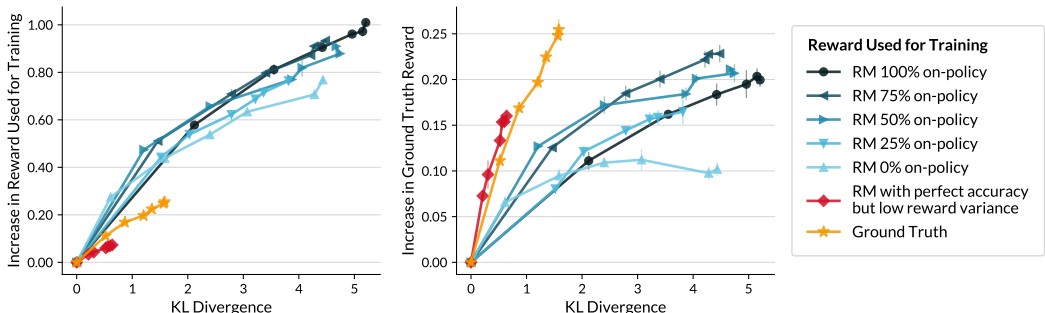

Figure 4: This figure supplements Figure 2 by plotting reward increase against the KL divergence from the initial policy, instead of against the number of epochs. A benefit of accurate reward models is that they are typically more KL efficient. In particular, at a given ground truth reward value, the KL divergence when using a perfectly accurate reward model is significantly smaller than when using a less accurate reward model.

Table 3: This table is analogous to Table 1, presenting the reward variance (induced for the initial policy) and accuracy of reward models that are based on a Pythia-1B initial policy, as opposed to a Pythia-2.8B one. All quantities were averaged across prompts in the policy gradient training set (their values on the test set were nearly identical). As mentioned in Section 4.1.1, to ensure fair comparison of reward variance, the reward models and ground truth reward were normalized so that they produce rewards on the same scale.

| | RM On-Policy % | | | | | RM with Perfect Acc. but Low Reward Variance | Ground Truth Reward |
| | 100% | 75% | 50% | 25% | 0% | | |
|---|---|---|---|---|---|---|---|
| Reward Variance | 0.660 | 0.625 | 0.626 | 0.605 | 0.374 | 0.131 | 0.239 |
| On-Policy Acc. | 0.662 | 0.650 | 0.643 | 0.614 | 0.553 | 1.000 | 1.000 |
| Off-Policy Acc. | 0.546 | 0.702 | 0.732 | 0.734 | 0.746 | 1.000 | 1.000 |

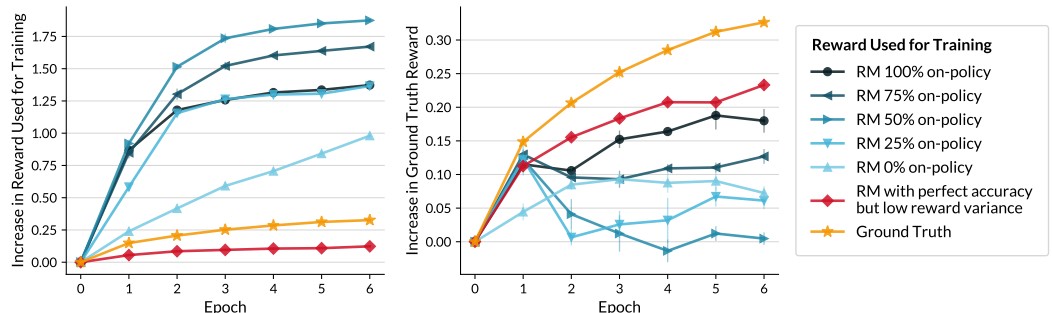

Figure 5: **More accurate reward models are not necessarily better teachers.** Plotted are the results of an experiment analogous to that of Figure 2, where the initial policy was a Pythia-1B language model instead of a Pythia-2.8B model. See Table 3 for characteristics of the reward models used in this experiment.

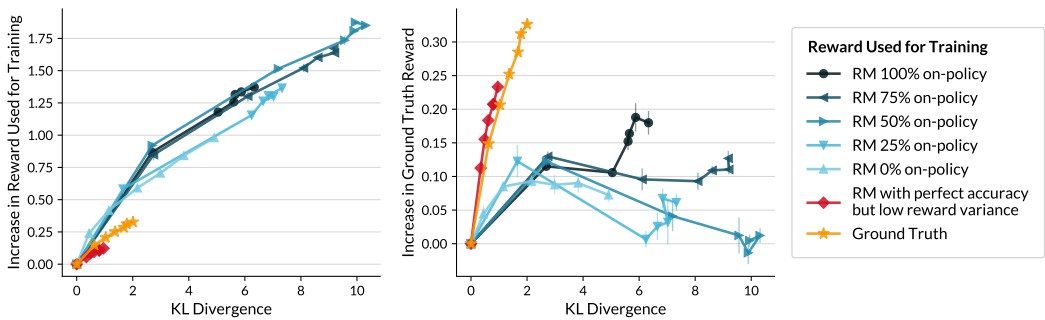

Figure 6: This figure supplements Figure 5 by plotting reward increase against the KL divergence from the initial policy, instead of against the number of epochs.

Table 4: **Reward variance strongly correlates with reward increase, while accuracy on its own may not.** For the experiments in Figure 5, we report the Pearson and Spearman correlations of different reward model properties (see Table 3) with reward increase after one epoch of policy gradient. "On- & Off-Policy Acc." refers to accuracy measured both on output pairs sampled from the initial policy and output pairs from UltraFeedback. "Reward Variance & Acc." averages this accuracy with reward variance (induced for the initial policy). Notably, the latter combined measure is more indicative of ground truth reward increase than each measure separately.

| | Increase in Reward Used for Training | | Increase in Ground Truth Reward | |
| --- | --- | --- | --- | --- |
| | Pearson Corr. | Spearman Corr. | Pearson Corr. | Spearman Corr. |
| Reward Variance | **0.941** | **0.943** | 0.409 | 0.428 |
| On-Policy Acc. | −0.524 | −0.028 | 0.290 | 0.028 |
| Off-Policy Acc. | −0.780 | −0.828 | −0.070 | −0.543 |
| On- & Off-Policy Acc. | −0.690 | −0.200 | 0.125 | 0.143 |
| Reward Variance & Acc. | 0.853 | 0.771 | **0.892** | **0.828** |

Table 5: This table is analogous to Table 1, presenting the reward variance (induced for the initial policy) and accuracy of reward models that are based on a Pythia-1B initial policy that was SFTed on the (UltraFeedback-based) reward model training set, as opposed to a Pythia-2.8B initial policy that was SFTed on AlpacaFarm. All quantities were averaged across prompts in the policy gradient training set (their values on the test set were nearly identical). As mentioned in Section 4.1.1, to ensure fair comparison of reward variance, the reward models and ground truth reward were normalized so that they produce rewards on the same scale.

| | RM On-Policy % | | | | | RM with Perfect Acc. but Low Reward Variance | Ground Truth Reward |
| --- | --- | --- | --- | --- | --- | --- | --- |
| | 100% | 75% | 50% | 25% | 0% | | |
| Reward Variance | 0.755 | 0.725 | 0.685 | 0.728 | 0.746 | 0.162 | 0.317 |
| On-Policy Acc. | 0.702 | 0.687 | 0.656 | 0.627 | 0.567 | 1.000 | 1.000 |
| Off-Policy Acc. | 0.570 | 0.706 | 0.700 | 0.732 | 0.754 | 1.000 | 1.000 |

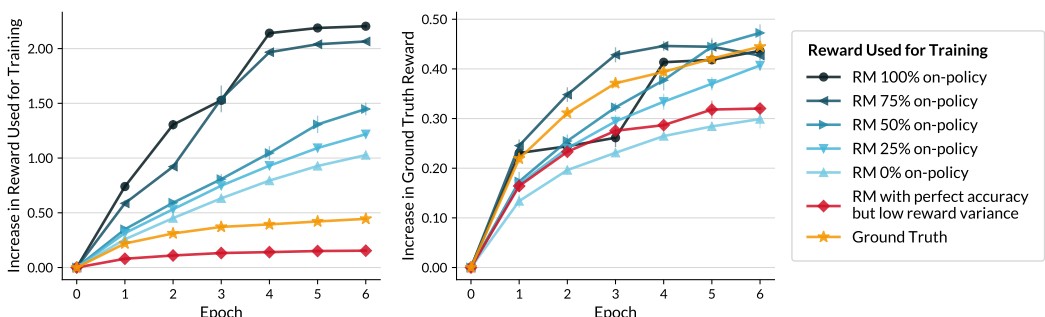

Figure 7: **More accurate reward models are not necessarily better teachers.** Plotted are the results of an experiment analogous to that of Figure 2, where: *(i)* the initial policy was a Pythia-1B language model instead of a Pythia-2.8B model; and *(ii)* SFT was carried out over the (UltraFeedback-based) reward model training set instead of AlpacaFarm. See Table 5 for characteristics of the reward models used in this experiment.

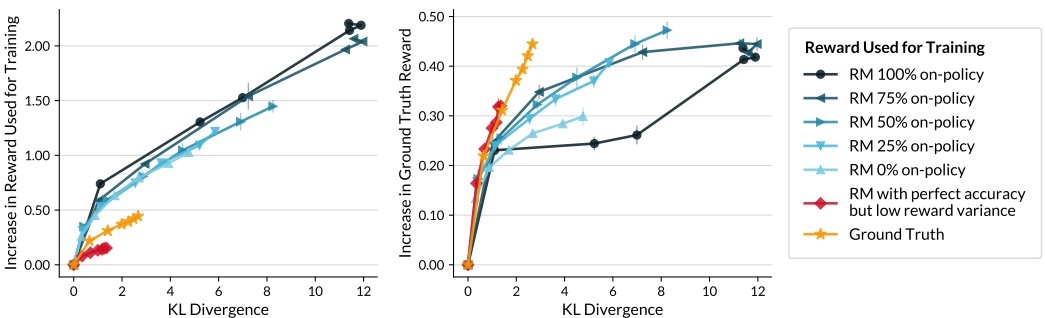

Figure 8: This figure supplements Figure 7 by plotting reward increase against the KL divergence from the initial policy, instead of against the number of epochs.

Table 6: **Reward variance strongly correlates with reward increase, while accuracy on its own may not.** For the experiments in Figure 7, we report the Pearson and Spearman correlations of different reward model properties (see Table 5) with reward increase after one epoch of policy gradient. "On- & Off-Policy Acc." refers to accuracy measured both on output pairs sampled from the initial policy and output pairs from UltraFeedback. "Reward Variance & Acc." averages this accuracy with reward variance (induced for the initial policy). In this set of experiments, the initial policy was SFTed on the (UltraFeedback-based) reward model training set. Compared to Tables 2 and 4, where initial policies were SFTed on AlpacaFarm, reward variance exhibits a lower (yet still positive) correlation with reward increase. This difference seems to arise from a reduced diversity in reward models. Namely, since SFT is performed on the reward model training set, the initial on-policy and off-policy output distributions are more similar, leading to less variation among reward models trained on different percentages of on-policy and off-policy output pairs. Indeed, as shown in Table 5, these reward models induce similar reward variances for the initial policy, making reward variance less indicative of RLHF outcomes.

| | Increase in Reward Used for Training | | Increase in Ground Truth Reward | |
| --- | --- | --- | --- | --- |
| | Pearson Corr. | Spearman Corr. | Pearson Corr. | Spearman Corr. |
| Reward Variance | **0.662** | **0.486** | 0.259 | 0.086 |
| On-Policy Acc. | $-0.388$ | 0.143 | 0.050 | 0.371 |
| Off-Policy Acc. | $-0.874$ | $-0.943$ | $-0.507$ | $-0.771$ |
| On- & Off-Policy Acc. | $-0.668$ | $-0.486$ | $-0.235$ | $-0.028$ |
| Reward Variance & Acc. | 0.637 | 0.371 | **0.286** | **0.371** |

Table 7: This table is analogous to Table 1, presenting the reward variance (induced for the initial policy) and accuracy of reward models that are based on a Pythia-1B initial policy. The ground truth reward model in this set of experiments was the GRM-gemma2-2B-rewardmodel-ft [87] model instead of the ArmoRM model. All quantities were averaged across prompts in the policy gradient training set (their values on the test set were nearly identical). As mentioned in Section 4.1.1, to ensure fair comparison of reward variance, the reward models and ground truth reward were normalized so that they produce rewards on the same scale.

| | RM On-Policy % | | | | | RM with Perfect Acc. but Low Reward Variance | Ground Truth Reward |
| --- | --- | --- | --- | --- | --- | --- | --- |
| | 100% | 75% | 50% | 25% | 0% | | |
| Reward Variance | 0.682 | 0.676 | 0.596 | 0.453 | 0.405 | 0.090 | 0.346 |
| On-Policy Acc. | 0.678 | 0.673 | 0.650 | 0.618 | 0.578 | 1.000 | 1.000 |
| Off-Policy Acc. | 0.612 | 0.680 | 0.682 | 0.698 | 0.716 | 1.000 | 1.000 |

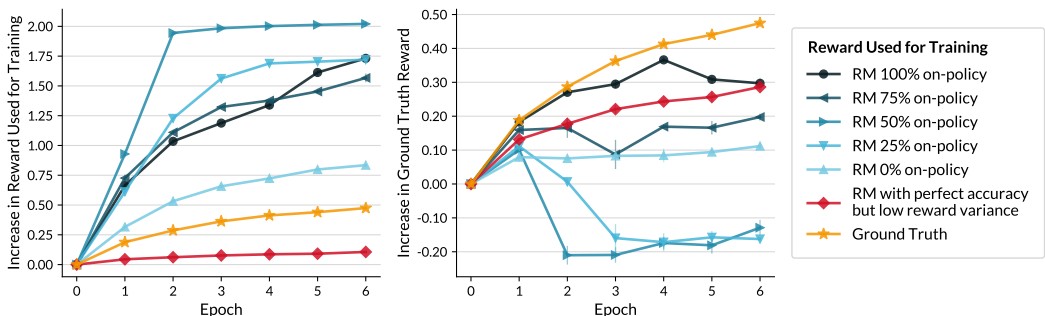

Figure 9: **More accurate reward models are not necessarily better teachers.** Plotted are the results of an experiment analogous to that of Figure 2, where: *(i)* the initial policy was a Pythia-1B language model instead of a Pythia-2.8B model; and *(ii)* GRM-gemma2-2B-rewardmodel-ft [87] served as the ground truth reward instead of ArmoRM. See Table 7 for characteristics of the reward models used in this experiment.

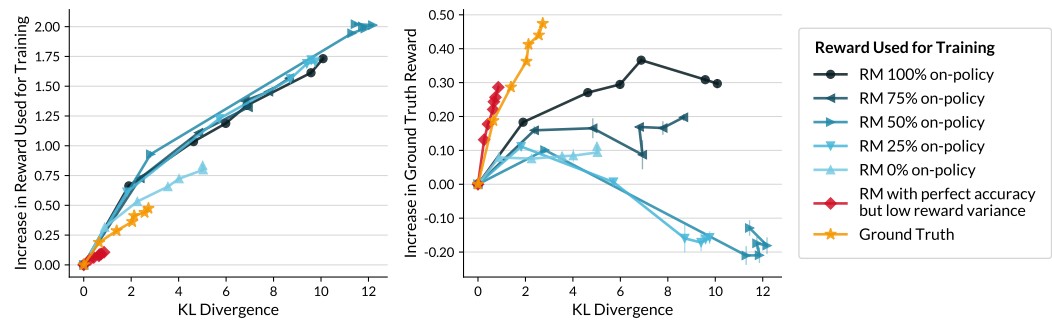

Figure 10: This figure supplements Figure 9 by plotting reward increase against the KL divergence from the initial policy, instead of against the number of epochs.

Table 8: **Reward variance strongly correlates with reward increase, while accuracy on its own may not.**
For the experiments in Figure 9, we report the Pearson and Spearman correlations of different reward model properties (see Table 7) with reward increase after one epoch of policy gradient. "On- & Off-Policy Acc." refers to accuracy measured both on output pairs sampled from the initial policy and output pairs from UltraFeedback. "Reward Variance & Acc." averages this accuracy with reward variance (induced for the initial policy). In these experiments, on-policy accuracy has the highest Spearman correlation with ground truth reward increase. However, if one averages reward variance just with on-policy accuracy (*i.e.*, excluding off-policy accuracy), then the correlations are even higher: 0.809 Pearson and 0.828 Spearman.

|  | Increase in Reward Used for Training | | Increase in Ground Truth Reward | |
|---|---|---|---|---|
|  | Pearson Corr. | Spearman Corr. | Pearson Corr. | Spearman Corr. |
| Reward Variance | **0.887** | **0.771** | 0.391 | 0.600 |
| On-Policy Acc. | −0.641 | −0.086 | 0.264 | **0.771** |
| Off-Policy Acc. | −0.813 | −0.771 | −0.176 | −0.600 |
| On- & Off-Policy Acc. | −0.744 | −0.086 | 0.057 | 0.028 |
| Reward Variance & Acc. | 0.725 | **0.771** | **0.732** | 0.714 |

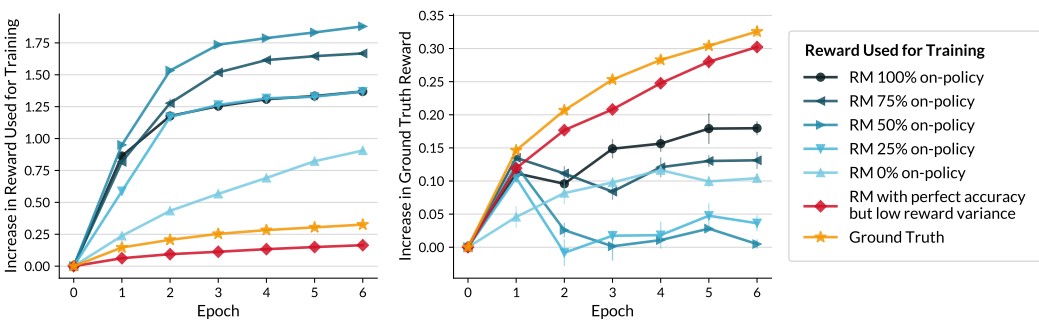

Figure 11: **More accurate reward models are not necessarily better teachers.** Plotted are the results of an experiment analogous to that of Figure 5, where reward models were normalized using a different scheme. Specifically, each reward model was normalized by the square root of the average initial reward variance over prompts in the training set, instead of by the reward standard deviation taken across training prompts jointly (*cf.* Appendix F.1). See Table 3 for characteristics of the reward models used in this experiment.

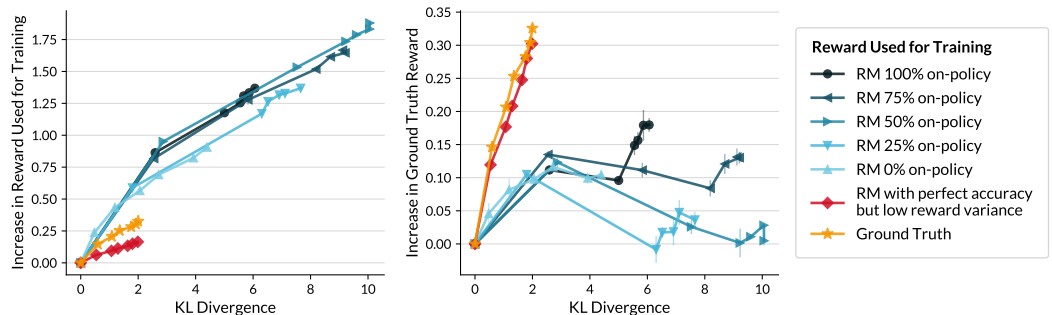

Figure 12: This figure supplements Figure 11 by plotting reward increase against the KL divergence from the initial policy, instead of against the number of epochs.

Table 9: **Reward variance strongly correlates with reward increase, while accuracy on its own may not.** For the experiments in Figure 11, we report the Pearson and Spearman correlations of different reward model properties (see Table 3) with reward increase after one epoch of policy gradient. "On- & Off-Policy Acc." refers to accuracy measured both on output pairs sampled from the initial policy and output pairs from UltraFeedback. "Reward Variance & Acc." averages this accuracy with reward variance (induced for the initial policy). Notably, the latter combined measure is more indicative of ground truth reward increase than each measure separately.

|  | Increase in Reward Used for Training | | Increase in Ground Truth Reward | |
|  | Pearson Corr. | Spearman Corr. | Pearson Corr. | Spearman Corr. |
| --- | --- | --- | --- | --- |
| Reward Variance | **0.936** | **0.943** | 0.279 | 0.314 |
| On-Policy Acc. | $-0.517$ | $-0.029$ | 0.416 | 0.543 |
| Off-Policy Acc. | $-0.768$ | $-0.829$ | 0.033 | $-0.371$ |
| On- & Off-Policy Acc. | $-0.680$ | $-0.200$ | 0.248 | 0.600 |
| Reward Variance & Acc. | 0.856 | 0.771 | **0.808** | **0.714** |

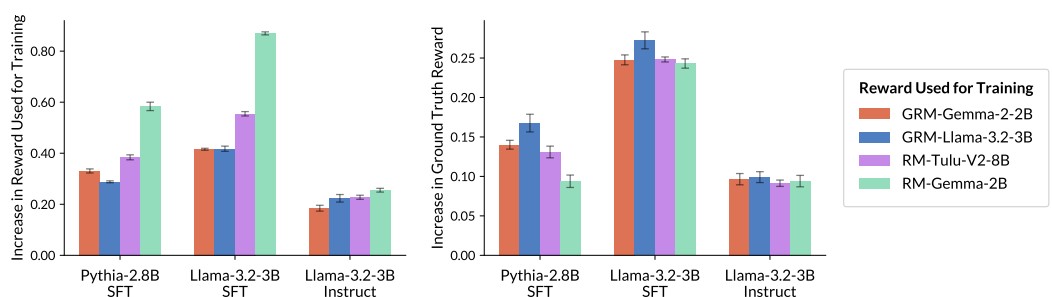

Figure 13: **For different language models, different reward models are better.** This figure extends the experiments of Figure 3 by considering additional (larger) landuage models. See Table 11 for characteristics of the reward models.

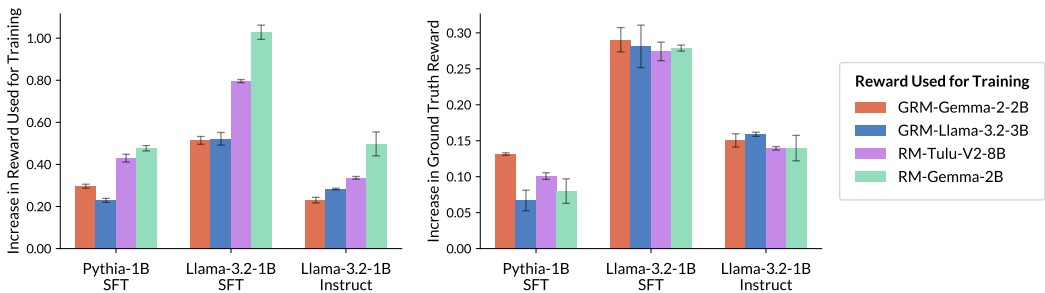

Figure 14: **For different language models, different reward models are better.** Plotted are the results of experiments identical to those in Figure 3, except that we used GRPO [69] instead of RLOO as the policy gradient method. See Table 12 for characteristics of the reward models.

Table 10: For each combination of reward model and initial policy in Figure 3, we report the: *(i)* ground truth reward increase after policy gradient (corresponds to the right bar plot in Figure 3); *(ii)* proxy reward increase after policy gradient (corresponds to the left bar plot in Figure 3); *(iii)* reward variance; and *(iv)* accuracy, measured on-policy (*i.e.*, on outputs sampled from the initial policy) and off-policy (*i.e.*, on outputs from UltraFeedback). All quantities were computed for prompts in the policy gradient training set (their values on the test set were nearly identical). As mentioned in Section 4.2.1, to ensure fair comparison of reward increase and reward variance, the reward models and ground truth reward were normalized so that they produce rewards on the same scale.

| **Language Model:** Pythia-1B SFT | | | | |
|---|---|---|---|---|
| | Reward Model | | | |
| | GRM-Gemma-2-2B | GRM-Llama-3.2-3B | RM-Tulu-V2-8B | RM-Gemma-2B |
| Reward Increase (G) | **0.176 ± 0.006** | 0.148 ± 0.004 | **0.173 ± 0.009** | 0.108 ± 0.017 |
| Reward Increase (RM) | 0.322 ± 0.006 | 0.272 ± 0.004 | 0.457 ± 0.017 | 0.474 ± 0.008 |
| Reward Variance | 0.405 | 0.528 | 0.544 | 0.549 |
| On-Policy Acc. | 0.700 | 0.700 | 0.705 | 0.639 |
| Off-Policy Acc. | 0.802 | 0.838 | 0.860 | 0.800 |

| **Language Model:** Llama-3.2-1B SFT | | | | |
|---|---|---|---|---|
| | Reward Model | | | |
| | GRM-Gemma-2-2B | GRM-Llama-3.2-3B | RM-Tulu-V2-8B | RM-Gemma-2B |
| Reward Increase (G) | 0.280 ± 0.003 | 0.278 ± 0.006 | **0.320 ± 0.014** | 0.274 ± 0.010 |
| Reward Increase (RM) | 0.487 ± 0.008 | 0.484 ± 0.004 | 0.743 ± 0.016 | 0.898 ± 0.013 |
| Reward Variance | 0.500 | 0.562 | 0.600 | 0.644 |
| On-Policy Acc. | 0.716 | 0.737 | 0.737 | 0.676 |
| Off-Policy Acc. | 0.802 | 0.838 | 0.860 | 0.800 |

| **Language Model:** Llama-3.2-1B-Instruct | | | | |
|---|---|---|---|---|
| | Reward Model | | | |
| | GRM-Gemma-2-2B | GRM-Llama-3.2-3B | RM-Tulu-V2-8B | RM-Gemma-2B |
| Reward Increase (G) | 0.105 ± 0.003 | **0.124 ± 0.007** | 0.097 ± 0.004 | 0.101 ± 0.003 |
| Reward Increase (RM) | 0.194 ± 0.004 | 0.232 ± 0.009 | 0.213 ± 0.016 | 0.339 ± 0.003 |
| Reward Variance | 0.261 | 0.323 | 0.299 | 0.336 |
| On-Policy Acc. | 0.731 | 0.758 | 0.751 | 0.647 |
| Off-Policy Acc. | 0.802 | 0.838 | 0.860 | 0.800 |

Table 11: For each combination of reward model and initial policy in Figure 13, we report the: *(i)* ground truth reward increase after policy gradient (corresponds to the right bar plot in Figure 13); *(ii)* proxy reward increase after policy gradient (corresponds to the left bar plot in Figure 13); *(iii)* reward variance; and *(iv)* accuracy, measured on-policy (*i.e.*, on outputs sampled from the initial policy) and off-policy (*i.e.*, on outputs from UltraFeedback). All quantities were computed for prompts in the policy gradient training set (their values on the test set were nearly identical). As mentioned in Section 4.2.1, to ensure fair comparison of reward increase and reward variance, the reward models and ground truth reward were normalized so that they produce rewards on the same scale.

| **Language Model:** Pythia-2.8B SFT | | | |
| --- | --- | --- | --- |
| Reward Model | | | |
| GRM-Gemma-2-2B | GRM-Llama-3.2-3B | RM-Tulu-V2-8B | RM-Gemma-2B |
| Reward Increase (G) | $0.140 \pm 0.005$ | $\mathbf{0.167 \pm 0.011}$ | $0.131 \pm 0.007$ | $0.094 \pm 0.008$ |
| Reward Increase (RM) | $0.331 \pm 0.008$ | $0.288 \pm 0.004$ | $0.384 \pm 0.010$ | $0.583 \pm 0.016$ |
| Reward Variance | 0.419 | 0.518 | 0.527 | 0.540 |
| On-Policy Acc. | 0.720 | 0.735 | 0.741 | 0.643 |
| Off-Policy Acc. | 0.802 | 0.838 | 0.860 | 0.800 |

| **Language Model:** Llama-3.2-3B SFT | | | |
| --- | --- | --- | --- |
| Reward Model | | | |
| GRM-Gemma-2-2B | GRM-Llama-3.2-3B | RM-Tulu-V2-8B | RM-Gemma-2B |
| Reward Increase (G) | $0.248 \pm 0.006$ | $\mathbf{0.272 \pm 0.011}$ | $0.248 \pm 0.003$ | $0.243 \pm 0.006$ |
| Reward Increase (RM) | $0.415 \pm 0.004$ | $0.418 \pm 0.010$ | $0.554 \pm 0.008$ | $0.870 \pm 0.006$ |
| Reward Variance | 0.397 | 0.453 | 0.472 | 0.552 |
| On-Policy Acc. | 0.736 | 0.763 | 0.756 | 0.674 |
| Off-Policy Acc. | 0.802 | 0.838 | 0.860 | 0.800 |

| **Language Model:** Llama-3.2-3B-Instruct | | | |
| --- | --- | --- | --- |
| Reward Model | | | |
| GRM-Gemma-2-2B | GRM-Llama-3.2-3B | RM-Tulu-V2-8B | RM-Gemma-2B |
| Reward Increase (G) | $\mathbf{0.096 \pm 0.007}$ | $\mathbf{0.099 \pm 0.007}$ | $\mathbf{0.091 \pm 0.004}$ | $\mathbf{0.094 \pm 0.007}$ |
| Reward Increase (RM) | $0.184 \pm 0.011$ | $0.223 \pm 0.015$ | $0.227 \pm 0.008$ | $0.255 \pm 0.007$ |
| Reward Variance | 0.249 | 0.301 | 0.279 | 0.320 |
| On-Policy Acc. | 0.714 | 0.743 | 0.730 | 0.651 |
| Off-Policy Acc. | 0.802 | 0.838 | 0.860 | 0.800 |

Table 12: For each combination of reward model and initial policy in Figure 14, we report the: *(i)* ground truth reward increase after policy gradient (corresponds to the right bar plot in Figure 14); *(ii)* proxy reward increase after policy gradient (corresponds to the left bar plot in Figure 14); *(iii)* reward variance; and *(iv)* accuracy, measured on-policy (*i.e.*, on outputs sampled from the initial policy) and off-policy (*i.e.*, on outputs from UltraFeedback). All quantities were computed for prompts in the policy gradient training set (their values on the test set were nearly identical). As mentioned in Section 4.2.1, to ensure fair comparison of reward increase and reward variance, the reward models and ground truth reward were normalized so that they produce rewards on the same scale.

**Language Model:** Pythia-1B SFT (GRPO)

| | Reward Model | | | |
| --- | --- | --- | --- | --- |
| | GRM-Gemma-2-2B | GRM-Llama-3.2-3B | RM-Tulu-V2-8B | RM-Gemma-2B |
| Reward Increase (G) | $\mathbf{0.131 \pm 0.002}$ | $0.067 \pm 0.014$ | $0.101 \pm 0.005$ | $0.080 \pm 0.017$ |
| Reward Increase (RM) | $0.296 \pm 0.010$ | $0.229 \pm 0.009$ | $0.430 \pm 0.018$ | $0.477 \pm 0.013$ |
| Reward Variance | 0.405 | 0.528 | 0.544 | 0.549 |
| On-Policy Acc. | 0.700 | 0.700 | 0.705 | 0.639 |
| Off-Policy Acc. | 0.802 | 0.838 | 0.860 | 0.800 |

**Language Model:** Llama-3.2-1B SFT (GRPO)

| | Reward Model | | | |
| --- | --- | --- | --- | --- |
| | GRM-Gemma-2-2B | GRM-Llama-3.2-3B | RM-Tulu-V2-8B | RM-Gemma-2B |
| Reward Increase (G) | $\mathbf{0.290 \pm 0.017}$ | $\mathbf{0.281 \pm 0.029}$ | $\mathbf{0.274 \pm 0.013}$ | $\mathbf{0.279 \pm 0.004}$ |
| Reward Increase (RM) | $0.514 \pm 0.018$ | $0.522 \pm 0.030$ | $0.796 \pm 0.007$ | $1.028 \pm 0.034$ |
| Reward Variance | 0.500 | 0.562 | 0.600 | 0.644 |
| On-Policy Acc. | 0.716 | 0.737 | 0.737 | 0.676 |
| Off-Policy Acc. | 0.802 | 0.838 | 0.860 | 0.800 |

**Language Model:** Llama-3.2-1B-Instruct (GRPO)

| | Reward Model | | | |
| --- | --- | --- | --- | --- |
| | GRM-Gemma-2-2B | GRM-Llama-3.2-3B | RM-Tulu-V2-8B | RM-Gemma-2B |
| Reward Increase (G) | $\mathbf{0.150 \pm 0.009}$ | $\mathbf{0.159 \pm 0.003}$ | $0.139 \pm 0.002$ | $\mathbf{0.140 \pm 0.018}$ |
| Reward Increase (RM) | $0.230 \pm 0.013$ | $0.283 \pm 0.004$ | $0.336 \pm 0.006$ | $0.498 \pm 0.057$ |
| Reward Variance | 0.261 | 0.323 | 0.299 | 0.336 |
| On-Policy Acc. | 0.731 | 0.758 | 0.751 | 0.647 |
| Off-Policy Acc. | 0.802 | 0.838 | 0.860 | 0.800 |

