# OpenReview forum: "What Makes a Reward Model a Good Teacher? An Optimization Perspective"
_NeurIPS.cc/2025/Conference — NeurIPS 2025 spotlight_

### Official Review · Reviewer_G5Cf · 2025-06-05

**Clarity:** 4
**Significance:** 3
**Originality:** 3
**Rating:** 5
**Confidence:** 4

**Summary:**

This paper studies the effects that reward variance has on RLHF fine-tuning. Specifically, the paper claims that low reward variance (measured in terms of how well the reward separates different responses), although coupled with high reward accuracy (measured in terms of how 'aligned' with the ground-truth reward the proxy is), may imply very slow convergence of the regularized policy optimization step in RLHF. The authors provide a systematic study of the phenomenon. They start by stating that convergence (in the sense of the increase in proxy reward beyond a certain threshold) is inversely proportional to the used reward variance. Then, under tabular softmax parametrization, they state that the same policy has much faster convergence when using a proxy reward with much lower accuracy (but higher variance) than a given reward. They also provide an additional result stating that different policies have different sensitivities to this phenomenon. Finally, they provide experimental results on language models to corroborate their theoretical findings. They use Pythia and Llama models on different proxy rewards (which have been manually tuned to induce high variance, with consequential lower accuracy) and show that higher variance does indeed imply faster growth, although coupled with lower accuracy at times.

**Questions:**

1. There is an important question related to the factors which are hidden in the statements provided in the main paper. When looking at the formal statements in the Appendix, the bounds seem to depend on the maximum norm of the Jacobian of the network over iterations. First, it is a bit strange to consider this dependence as irrelevant. To me this seems like a very important factor because it contains a lot of information on the landscape of the current network. Second, it is not clear to me how large or small this quantity is or what is its relation to the variance of the reward. I think this deserves some discussion. Can the authors please explain first the connection between variance and Jacobian in this particular setting? I understand that the reward does not depend on it. But surely the reward affects it over time, since it reflects changes that are occurring through updates until convergence.
2. To what extent is the theory here applicable to the standard RL setting? I assume the reward variance might have direct implications in RL due to poorly designed rewards. In this context, what is the relationship between a high-variance reward which might facilitate faster convergence, and potential-based rewards? As far as I understand (correct me if I'm wrong), reward variance as you have defined it directly corresponds to the average squared advantage function, which might directly provide the intuition of the relation between variance and gradient.
3. As mentioned in the previous section, it is indeed unfortunate that the analysis is not fully complete. I understand that non-concavity is a serious issue when analyzing the general setting, but, as you point out, the tabular instantiation does not resolve non-concavity. So, in itself, this might not be the real issue. I think this might be an important gap. Can you provide more details on the challenges you face when considering the general setting? And also, what challenges did the current analysis pose?
4. How does the inexact gradients exacerbate/ameliorate the issue? Due to stochasticity, might it not be the case that gradient estimates might somehow amortize the effect of low reward variance? Can you comment on this? Is it possible to provide additional analysis for inexact gradients? Would that be too challenging?

**Ethical Concerns:**

["NO or VERY MINOR ethics concerns only"]

**Final Justification:**

I believe the paper's results are overall significant to the NeurIPS community. The paper provides both justified theoretical claims and support them with experiments on relevant benchmarks. The major issue that I had with the paper was adequately resolved by the authors. Thus, I increase my rating accordingly.

**Limitations:**

Yes.

**Quality:**

3

**Strengths And Weaknesses:**

**Strengths**
1. The paper studies a very important problem and provides interesting insights that are relevant to theoreticians and practitioners in RLHF.
2. The paper is very well-written, clear and coherent. The results are well-explained and associated with corresponding intuition.
3. Theoretical results support all the claims made in the paper.
4. Finally, the paper includes experiments on actual language models for all the settings that are discussed.

**Weaknesses**
1. An immediate weakness of the paper is related to the scope of the provided theoretical intuition. Although I agree with the authors that theoretical analysis is hard beyond tabular settings, having one result on general differentiable functions, and then having additional results on tabular instantiations provides an incomplete theoretical picture. (See Questions for more details).
2. In addition, it is not clear at this point to what extent the analysis done in this paper borrows (or is a direct corollary of) the analysis done in [65].
3. Experimental results in additional benchmarks would significantly strengthen the impact of the paper.

---

> ### Author Rebuttal · Authors · 2025-07-30
>
> Thank you for the thoughtful feedback and for highlighting the interesting insights that our work provides and its clarity. We treat your comments and questions below. If our response addresses your comments, we would greatly appreciate it if you would consider raising your score. Please let us know if you have any further questions; we will gladly respond.
>
> &nbsp;
>
> **W1: Scope of theoretical results.**
> The core issue that we analyze – slow optimization due to low reward variance and its interplay with accuracy – does not depend on the neural network parameterization. It stems from the RLHF objective and applies to any softmax parameterized policy. That is, any policy that produces a distribution over outputs through the softmax function, which includes language models as a special case.
>
> Our work includes three main theorems showing that: (i) the time it takes to increase the reward grows inversely with the initial reward variance (Theorem 1), (ii) more accurate reward models are not necessarily better teachers (Theorem 2), and (iii) for different language models different reward models can work better (Theorem 3). Indeed, while we prove Theorem 1 for general differentiable parameterizations, Theorems 2 and 3 are proved for tabular policies since they require establishing separations between the optimization time of reward models, which entails substantial technical difficulty (as further discussed below in response to other comments). However, given that the phenomenon that we analyze stems primarily from the objective rather than any particular neural network parameterization, we do not believe that this is a major limitation. This belief is strengthened by the experiments of Section 4, which empirically demonstrate that conclusions drawn from our theory carry over to practical settings.
>
> &nbsp;
>
> **W2 + Q3: Relation to the analysis in [65] and technical challenges.**
> While our work is inspired by the observations made in [65], our theory (and experiments) go far beyond them. In particular, all three main theorems in our work are by no means corollaries of the results in [65], as can be seen from the proof appendix (Appendix E). Specifically, as discussed in the proof sketch of Theorem 1 (Section 3.2), [65] only provides a bound on the gradient norm at a given point. This alone does not imply a lower bound on optimization time. It is necessary to show that the gradient norm will not increase during training by proving that reward variance cannot increase rapidly along the trajectory of optimization. Moreover, Theorems 2 and 3 require proving separations in terms of optimization time between reward models. That is, aside from the lower bound on optimization time, we established upper bounds. This required proving a sufficient condition (Proposition 4 in Appendix E) for efficient optimization via a fine-grained analysis of the optimization trajectory when maximizing the non-concave RLHF objective. Lastly, aside from the technical novelty of our theoretical results, the focus of our work is different from that of [65], which did not consider the question of what makes a good reward model for RLHF or the interplay between accuracy and reward variance.
>
> Regarding challenges in extending Theorems 2 and 3 to more general neural network architectures (considered in Theorem 1). Analyzing optimization for practical neural networks is still a major open problem even in simpler settings where the objective with respect to the neural network outputs is convex (e.g., supervised learning with the squared or cross entropy losses). While we were able to establish a lower bound, with the architecture contributing a dependence on its Jacobian norm, the challenges in establishing upper bounds on convergence rates are exacerbated due to the non-concavity of the RLHF objective with respect to the neural network outputs. Specifically, the architecture introduces complex interaction between the evolution of logits for different outputs, and as mentioned earlier, the understanding of these interactions is limited even in simpler settings. Hopefully, future technical advances may allow generalizing our theoretical results, as well as other optimization analyses of neural networks in the literature, to additional architectures.
>
> &nbsp;
>
> **Q1: Dependence on Jacobian norm in Theorem 1.**
> Good question! Our interest lies in comparing, for a given language model, how a reward model affects RLHF optimization. Initially, the Jacobian of the language model does not depend on the reward model (rather just on the language model architecture and parameters) and we empirically found that its norm does not change much during policy gradient (see details below). Thus, for simplicity, we omitted the dependence on the Jacobian norm from the abridged Theorem 1, but of course included it in the full theorem statement (Theorem 4). The goal in doing so was to emphasize the parts which we found to be the most crucial and relevant to our study. Please let us know if you believe that it would be clearer to include this term in the abridged version as well and we will gladly accommodate.
>
> *Details:* In a setting analogous to Section 4.1, with a Pythia-410m language model, we ran policy gradient using two different reward models (one trained on 100% on-policy data and another on 0% on-policy data). For efficiency, instead of the spectral norm, we tracked the average Frobenius norm of randomly selected rows in the Jacobian throughout training. In both runs the average Frobenius norm of rows hovered around 200 – 300, i.e., it was on the same order of magnitude throughout policy gradient and for both reward models. Note that since the Frobenius norm of the Jacobian upper bounds its spectral norm, this indicates that the spectral norm does not grow by much as well.
>
> &nbsp;
>
> **Q2: Applicability of our results to other RL settings.**
> This is a great point! Thank you for raising it. While the positioning of our work focuses on RLHF, the theoretical results can be straightforwardly extended to support any RL setting with softmax parameterized policies (i.e., policies that produce a distribution over outputs via softmax). In accordance with your comment, we believe it can be a valuable direction to examine whether low reward variance empirically poses a problem in other RL settings and design potential-based reward modifications to increase reward variance. We will clarify this point in the manuscript.
>
> Note 1: As discussed in Appendix A, we are not aware of existing reward shaping techniques that aim to increase reward variance or consider the effect of reward shaping on policy gradient optimization.
>
> Note 2: Indeed, in bandit settings reward variance is equivalent to the expected squared advantage function, where the expectation is with respect to outputs sampled from the policy.
>
> &nbsp;
>
> **Q4: Effect of inexact gradients.**
> The limitations paragraph in Section 3.1 and the first paragraph of the future work section (Appendix B) briefly discuss this point. In light of your question, we will expand the discussion on the effect of inexact gradients in the future work section along the lines of the answer below.
>
> Our experiments consider standard policy gradient methods that compute gradient updates based on batches of sampled outputs, because computing the exact expected gradient is computationally infeasible. Thus, they empirically demonstrate that the implications of our theory carry over to settings with inexact gradients. Intuitively, the reason that low reward variance remains problematic in the inexact gradient setting is as follows. When the reward variance is low, the expected gradient vanishes. In this case, batch gradients may not necessarily be small (whether they will be small or not can depend on which policy gradient method you use). However, since the expected gradient is small, this implies that the batch gradient is often noisy, and so updates do not lead to substantial progress. Formally extending our analysis to the inexact gradient setting is highly non-trivial and we believe that it could be an interesting challenge for future work, which may further shed light on the relative advantages and disadvantages of different policy gradient methods.
>
> &nbsp;
>
> **W3: Experimental results on additional benchmarks.**
> Our experiments are mostly based on the UltraFeedback dataset, which is a standard dataset for language model alignment. Since running the RLHF pipeline with large models is computationally expensive, we preferred to cover multiple language models, reward models, and ground truth rewards instead of additional datasets. The current experiments (in Section 4 and Appendix F) clearly demonstrate that the conclusions of our theory apply in practical RLHF settings. Thus, while we agree that additional datasets can give further evidence, we do not believe they are necessary to strengthen the claims made in the paper. Nonetheless, if there is any particular benchmark that you find is especially important, we would be happy to accommodate.

---

> > ### Comment · Reviewer_G5Cf · 2025-08-05
> >
> > I thank the authors for addressing my concerns. I am, though, still a bit concerned about the dependence on the Jacobian. While the authors provide experimental results which show that the Frobenius norm remains within a relatively small margin (although 200 - 300 is still a bit debatable whether it implies stability or not), this does not mean that the spectral norm, which indeed is the dependent quantity in the theoretical results, does not vary drastically. Since the Frobenius norm upper bounds the spectral norm, we may very well be looking at a change from 0.1 to 300 in terms of the spectral norm, which, of course, might seem like a improbable case, but it's possible, even within the provided experimental results. This implies that the Jacobian can vary drastically, in the spectral norm sense.
> >
> > While it is true that the Jacobian of the initial policy is independent of the reward and its properties, that does not seem to be the case during training. So, it is not clear to what extent the interdependence between reward variance, its observations, and the parametrization of the updated network, which, subsequently affect the Jacobian, plays a role in this. Also, from a theoretical viewpoint, the concern is that you treating as constant something that depends on time. That is, in Theorem 4, for instance, the quantity $t_\gamma$ that you are lower-bounding also affects the "constant" $J$, which is a supremum over steps up to $t_\gamma$.
> >
> > Finally, how are $J$ and reward variance compared? Can $1/J^2$ be small enough that, even for some considerable reward variance, the final lower bound is vacuous, that is, something like $t_\gamma \geq 0.2$?

---

> > > ### Author Response · Authors · 2025-08-05
> > >
> > > Thank you for the response; we greatly appreciate your engagement! We address your comments below and would be happy to elaborate if you have any further questions.
> > >
> > > The evolution of the Jacobian norm during training depends on the particular parameterization. For simple parameterizations, such as tabular or linear parameterizations, the Jacobian norm does not change. On the other hand, for practical neural network architectures it does change, and this change can depend on the reward model. As you suggest, in theory, it may be the case that for a certain policy parameterization the Jacobian norm varies significantly when trained using different reward models or that the exact numerical bound becomes vacuous.
> > >
> > > However, the experiments in Section 4 clearly demonstrate that the implications of Theorem 1, as well as of Theorems 2 and 3 that are established based on it, hold in practical RLHF settings. In particular, as Figure 2 and Table 2 show, reward variance strongly correlates with reward increase. In our view, the significance and utility of a theory are closely related to its predictive power in practice. Thus, given the above, we believe that the dependence on the spectral norm of the Jacobian does not limit the significance of Theorem 1 (at least not substantially), since it does not seem to impact the conclusions in practice. Nonetheless, we regard obtaining a tighter control on the Jacobian norm for practical neural networks as an interesting challenge for future work, which may provide insights into whether particular parameterizations can avoid slow optimization due low reward variance.
> > >
> > > Lastly, thank you for bringing to our attention that the notation $J$ in Theorem 4 can make it seem as if it is a constant! We will modify this notation to $J_{t_{\gamma}}$ to ensure that the dependence on optimization time is clearer.

---

> > > > ### Comment · Reviewer_G5Cf · 2025-08-05
> > > >
> > > > That is a fair point. I agree that, while theoretically, there might still be some unclear dependence of your results on the Jacobian, the experimental results in the paper do seem to confirm the implications of their statements. I would nevertheless suggest adding, not only the dependence on $t_\gamma$ of $J$, but also at least a remark discussing such dependence. Overall, I did enjoy reading the paper and have no additional concerns.

---

> ### Author Response · Authors · 2025-08-06
>
> Thank you again for the fruitful and insightful discussion. In the revised manuscript, we will modify the notation in Theorem 4 and incorporate a discussion of the dependence on the Jacobian norm, following the line of our responses above.

---

### Official Review · Reviewer_PFVe · 2025-06-29

**Clarity:** 4
**Significance:** 2
**Originality:** 2
**Rating:** 4
**Confidence:** 2

**Summary:**

This paper raise an question about what makes a reward model a good teacher for RLHF. For current literature, reward accuracy is consider as the most significant factor. However this paper show that the low variance of reward model will also cause slow learning.

**Questions:**

1. Could the authors provide more background on how this specific problem differs from classical RL challenges, such as sparse reward settings?
2. Could the authors include an additional experiment using the same reward model but with different reward scaling factors, to directly assess how reward scale and induced variance affect learning performance?
3. During training, do all methods use the same input prompt x during the same iteration (x for iter i may be different than i+1 iter but within i iteration will method have same set of x)? If so, this would strongly support the validity of the theoretical analysis presented in the paper.

I am willing to increase the score if the auther provided a clearer explanation of the unique challenges this problem presents, especially in contrast to classical RL issues, and addressed some of the weaknesses in its formulation and assumptions.

**Ethical Concerns:**

["NO or VERY MINOR ethics concerns only"]

**Final Justification:**

The rebuttal have addressed most of my concerns. However, I cannot give full accept assessment (5) since the reward scale issue. From the result provided by auther, this issue seens minor however from the fundamental mathmatic perspective, the change scale cannot be equalivant to change learning rate. For example, if you change learning rate, every model change the same amount which makes the comparsion fair. However, when normalize to the same scale, different model will be applied different factors and it will be even worse for the variance change (will be the square of the factor).

**Limitations:**

yes

**Quality:**

3

**Strengths And Weaknesses:**

My research mainly focus on RL/DRL control so I cannot thoroughly evaluate this paper from the LLM perspective. I will more focus on the fundamental RL perspective.

>Strengths:
1. the writing is clear.
2. this paper proves that even a perfectly accurate reward model can lead to slow learning if it induces low reward variance.
3. the paper has comprehensive experiments with different models and different datasets.

>Weaknesses:

1. The novalty of this paper. In RL, the dense and sparse reward has been studied for decades. The sparse reward problem (naturely very low reward variance) is very similar to the problem studied in this paper. It is common sense that sparse reward is challenging than the dense problem. Additionally, the paper does not provide solution to this issue. However, RL and DRL has been studied and prove a lot useful framework such as reward shaping or transfer learning.
2. Studying variance along is not sufficent. In [1], it has been shown that the reward scale can significantly impact performance. Even when using the same reward model, applying different scaling factors results in different reward scale and variances will affect performance.
3. The definition of reward variance which is based on same x but different y. This formulation ignore the fact that during batch training, within the batch the data tuple are from differet x and y.

[1] Henderson, Peter, et al. "Deep reinforcement learning that matters." Proceedings of the AAAI conference on artificial intelligence. Vol. 32. No. 1. 2018.

---

> ### Author Rebuttal · Authors · 2025-07-30
>
> Thank you for the thoughtful feedback and for highlighting its clarity, the theoretical results, and their comprehensive empirical validation. We treat your comments and questions below. If our response addresses your concerns, we would greatly appreciate it if you would consider raising your score. Please let us know if you have any further questions; we will gladly respond.
>
> &nbsp;
>
> **W1 and Q1: Relation of low reward variance to sparse rewards.**
> When rewards are sparse (i.e., only a few outputs/trajectories receive a non-zero reward), often the reward variance is low. However, the reward variance can be low even if rewards are not sparse (i.e., dense). Importantly, in RLHF rewards are typically not sparse – reward models provide a reward for each output, which is usually non-zero. As our work highlights, despite rewards not being sparse, policy gradient still suffers from a flat objective landscape that leads to slow optimization when reward variance is low. This occurs when the reward model does not sufficiently distinguish between outputs that are probable under the policy (i.e., it gives them a similar, not necessarily near-zero, reward) and stems from the objective landscape induced by policies that produce a distribution over outputs via the softmax function.
>
> As discussed in the related work section (Appendix A), the literature on reward shaping can indeed be seen as relevant to our work. Though, to the best of our knowledge, prior work did not characterize how reward shaping affects policy gradient optimization. In the related work section we further discuss the fact that, in theory, the poor optimization landscape of policy gradient can be solved via natural policy gradient (a preconditioning method). However, this approach is currently impractical in the context of large language models. Specifically, natural policy gradient requires inverting a $P \times P$ matrix, where $P$ is the number of language model parameters.
>
> Lastly, while our work does not give a solution to the issue of low reward variance, it highlights limitations of existing reward model benchmarks and provides insight into what determines the effectiveness of a reward model, beyond accuracy. We hope that it can inspire future work on developing reward model training procedures that encourage a higher reward variance and policy gradient methods that can optimize more efficiently under low reward variance. Due to the extent of our theoretical and empirical results, we believe that this important and highly non-trivial endeavour falls outside the scope of the current paper.
>
> &nbsp;
>
> **W2 and Q2: Reward scale.**
> You raise an excellent point! It is true that reward scale can impact performance. In particular, it can have an effect similar to modifying the learning rate (though not always exactly the same effect when applying KL regularization and using adaptive optimizers). For this reason, to ensure fair comparison across reward models, in our experiments we normalized all reward models so that they produce rewards of roughly the same scale. This normalization is also necessary for ensuring that reward variance is a meaningful measure when comparing different reward models.
>
> Nonetheless, per your request we ran experiments using the same reward model, in which we scaled the reward by different constants. As expected, lower reward scales led to slower optimization while the larger reward scales that we tried led to unstable optimization that resulted in a lower ground truth reward.
>
> *Details:* To obtain results in time for the response period, we considered a setting similar to Section 4.1 with Pythia-1B models and ran a single policy gradient run per scale factor (as shown in the paper, results were typically stable across random seeds). In all runs, we used the RM trained on 75% on-policy data.
>
> | Scale | Ground Truth Reward Increase |
> |-------|------------------------------|
> | 0.001 | 0.022                        |
> | 0.01  | 0.039                        |
> | 0.1   | 0.110                        |
> | 1     | 0.109                        |
> | 10    | 0.080                        |
> | 100   | 0.086                        |
> | 1000  | 0.083                        |
>
> As noted in the implementation details appendix, we observed similar results when tweaking the learning rate. Trends across reward models of the same learning rate remained intact.
>
> &nbsp;
>
> **W3: Definition of reward variance.**
> Similarly to accuracy and many other evaluation measures, reward variance is defined for a given prompt/input. Such measures are typically aggregated over a set of inputs by taking the mean. In that regard, there is nothing special about reward variance and its definition. Indeed, the reward variance values we report in Table 1, as specified in its caption, are an average over the training prompts. In other words, the definition of reward variance does not ignore the existence of multiple examples in a dataset or batch, both in theory and in practice.
>
> &nbsp;
>
>
> **Q3: Prompt order during training.**
> We did not fix the order of prompts in the experiments of Section 4. Instead, we used three different seeds for random data orders and reported the standard deviation of results across the seeds. As our results show (e.g., the almost indiscernible error bars in Figure 2), the data order did not have a significant impact on the results. Nonetheless, as detailed below, during the response period **we ran additional experiments to further verify** that our results are robust to fixing the data order.
>
> *Details:* To obtain results in time for the response period, we considered a setting similar to Section 4.1 with Pythia-1B models and ran a single policy gradient per reward model (as shown in the paper, results were typically stable across random seeds). We include a table analogous to Table 2, showing the correlation between different reward model properties and the increase in reward used for training and ground truth reward. As was the case in our original experiments, reward variance is highly correlated with reward (used for training) increase, while accuracy was not, with a combination of both being most indicative of ground truth reward increase.
>
> |                                | Reward Increase Pearson | Reward Increase Spearman | GT Reward Increase Pearson | GT Reward Increase Spearman |
> |--------------------------------|--------------------------|---------------------------|-----------------------------|------------------------------|
> | Reward Variance                | 0.958                    | 0.942                     | 0.426                       | 0.600                        |
> | On-Policy Acc.                 | -0.550                   | -0.028                    | 0.225                       | 0.485                        |
> | Reward Variance & On-Policy Acc. | 0.788                    | 0.771                     | 0.864                       | 0.771                        |
>
> Note: We omitted off-policy accuracy for brevity. It had lower correlations than on-policy accuracy.

---

> > ### Comment · Reviewer_PFVe · 2025-08-04
> > **follow up questions**
> >
> > Thank you for your details response. I still have some questions not fully cleared.
> >
> > > **Reward Scale Issue**
> >
> > The results appear reasonable, but the concern remains unaddressed that if you change the scale the variance will also change. The authors state that "we normalized all reward models so that they produce rewards of roughly the same scale," but this normalization introduces the exact issue I raised.
> >
> > Consider two reward models, A and B. Model A has a reward range of \[0, 1\] and variance 0.5, while model B has a range of \[0, 10] and the same variance of 0.5. After normalizing model B's rewards to [0, 1], its variance becomes \(0.1^2 * 0.5 = 0.005\), whereas model A's variance remains 0.5.
> >
> > This shows that normalization not only affects the scale but also alters the variance. As a result, it becomes unclear whether any observed performance differences stem from changes in reward scale or from differences in reward variance.
> >
> > > **W3 and Q3 follow up**
> >
> > I’m not questioning the correctness of the variance definition itself, nor am I disputing that the order of data may influence performance. I also appreciate the authors conducting additional experiments. However, my concern lies in whether the definition accurately reflects the actual experimental setup.
> >
> > In Definition 2, variance is defined by fixing x and varying y, and the subsequent theoretical analysis is based on this formulation. As I previously asked, during your actual experiments—particularly in batch training—is x truly fixed?
> > If x is indeed fixed during training, then your experimental setting aligns well with the theoretical assumptions, and the analysis is both solid and compelling. However, if both x and y vary during training, the setting departs from the theoretical conditions, and the applicability of the analysis becomes uncertain—since in real experiments, not only does reward variance change, but batch variance also comes into play, making it difficult to disentangle their individual effects.

---

> > > ### Author Response · Authors · 2025-08-04
> > >
> > > Thank you for the response; we greatly appreciate your engagement! We address your additional questions below. We would be glad to further elaborate if you have any more questions.
> > >
> > > &nbsp;
> > >
> > > **Reward Scale.**
> > > Your question highlights an important, yet nuanced, point regarding the relation between reward scale and reward variance! In light of your question, we plan to clarify this in a dedicated discussion after Definition 2, following a line similar to the explanation below.
> > >
> > > Reward variance captures two aspects of a reward model: (1) the scale of rewards and (2) the degree of separation between outputs for a given reward scale. As mentioned in the previous reply, modifying reward scale can have an effect similar to modifying the learning rate. Thus, for fair comparison and ease of tuning hyperparameters across reward models (e.g., learning rate), it is common practice to normalize reward models so that they produce rewards of roughly the same scale. Our experiments follow this practice, which helps avoid slow progress or instabilities that are simply due to a low or high reward scale.
> > >
> > > Since reward scale can easily be accounted for and modified, our work focuses more on the impact of reward variance on optimization through the degree of separation between outputs. Our results demonstrate that, even when accounting for reward scale, low reward variance leads to slow optimization. For example, in the experiments of Figure 2, after normalization the reward models with lower reward variance (e.g., the perfectly accurate RM) had a slightly higher scale than reward models with a higher reward variance (e.g., RM 100% on-policy). Yet, the former still led to slower optimization.
> > >
> > > *Table.* Reward scale, computed based on training prompts and outputs from initial policy, for the reward models in Figure 2.
> > > | RM 0% | RM 25% | RM 50% | RM 75% | RM 100% | Perfect RM w/ low variance | GT reward |
> > > |-------|--------|--------|--------|---------|------------------------|----------------------|
> > > | 0.8015| 0.7783 | 0.7834 | 0.8042 | 0.7917  | 0.8411                 | 0.8254               |
> > >
> > > Technical notes: Before normalization, the perfectly accurate reward model and the ground truth (GT) reward had a lower reward scale than the other reward models. So it is not the case that their lower reward variance is due to normalization scaling down their rewards, as in the example given in your question. Furthermore, as mentioned in Appendix G.1, when modifying the learning rate within ranges that optimization remained stable, we observed similar trends across reward models. In particular, it was not the case that through modifying the learning rate, which has a similar effect to reward scale, the performance after policy gradient when using the perfectly accurate reward model with low reward variance was similar or higher than the performance obtained with RM 75%.
> > >
> > >
> > > &nbsp;
> > >
> > > **W3 and Q3.**
> > > Thank you for clarifying your question. As detailed in Section 3.1, our analysis considers optimization using (full-batch) gradients computed over the whole training set of prompts. In practice, gradients are typically computed over mini-batches of prompts as opposed to the whole training set. Nonetheless, the experiments of Section 4 demonstrate the implications of our theory apply to this setting as well. Formally extending our analysis to the mini-batch gradient setting is highly non-trivial and we believe that it could be an interesting challenge for future work.
> > >
> > > We note that, due to the complexity of analyzing non-concave/convex objectives in the mini-batch setting, it is common in the literature to obtain theoretical insights by analyzing the full-batch setting instead, and subsequently verifying that the conclusions transfer to the mini-batch setting empirically (e.g., [1,2,3]). Given that Section 4 demonstrates an excellent match between our theory and experiments in practical settings, we do not believe that this limitation of our analysis (which is explicitly stated in Section 3.1) is a significant one.
> > >
> > > &nbsp;
> > >
> > > [1] Chizat, Lenaic, and Francis Bach. "On the global convergence of gradient descent for over-parameterized models using optimal transport." NeurIPS 2018.
> > >
> > > [2] Lyu, Kaifeng, and Jian Li. "Gradient descent maximizes the margin of homogeneous neural networks." ICLR 2020.
> > >
> > > [3] Frei, Spencer, Gal Vardi, Peter L. Bartlett, Nathan Srebro, and Wei Hu. "Implicit bias in leaky relu networks trained on high-dimensional data." ICLR 2023.

---

> > > > ### Comment · Reviewer_PFVe · 2025-08-05
> > > >
> > > > Thank you for your explaination. I have no more questions.

---

> > > > > ### Author Response · Authors · 2025-08-06
> > > > >
> > > > > We greatly appreciate the fruitful and insightful discussion. In light of it, we will include in the revised manuscript a clarification on the relation between reward variance and scale. Thank you again for raising this point!

---

### Official Review · Reviewer_8fPr · 2025-07-01

**Clarity:** 2
**Significance:** 2
**Originality:** 2
**Rating:** 4
**Confidence:** 3

**Summary:**

This paper investigates reward model effectiveness in RLHF from an optimization perspective, arguing that reward variance is as crucial as accuracy. The authors prove that low reward variance leads to flat objective landscapes and slow optimization, even for perfectly accurate reward models. They provide theoretical analysis showing different reward models work better for different language models, with experiments on models up to 8B parameters.

**Questions:**

**Q1. Variance Isolation Experiments:** Can you provide controlled experiments that artificially manipulate reward variance while keeping all other factors constant? Specifically, could you demonstrate the variance effect by taking identical reward models and applying temperature scaling to increase/decrease variance, then evaluating their effectiveness as teachers? This would directly test whether variance per se (rather than data distribution matching) drives the observed benefits.

**Q2. Same-Distribution Comparisons:** Would you be willing to conduct experiments where multiple reward models are trained on identical data distributions but exhibit different variance-accuracy trade-offs? For instance, training 10-20 models on the same dataset with different architectural choices or hyperparameters, then comparing high-variance/low-accuracy models against low-variance/high-accuracy ones. This would help isolate variance effects from the current confounding with data distribution differences.

**Q3. Practical Variance Enhancement Methods:** Can you demonstrate validated approaches for beneficially increasing reward model variance in practice? Your limitations section mentions that directly scaling rewards by factors >1 did not yield benefits, which appears to contradict your theoretical predictions. Could you provide a detailed analysis of why this approach failed and propose alternative methods that successfully leverage your theoretical insights?

**Q4. Ground Truth Validation:** Would you consider validating your experimental setup using tasks with objective ground truth (e.g., code generation, mathematical reasoning) or stronger foundation models as ground truth? The current reliance on ArmoRM as ground truth without validation against human preferences or objective correctness creates potential cascading errors throughout the evaluation.

**Evaluation Criteria:** My assessment would improve to "Accept" (5) if you can provide convincing evidence that variance effects persist when data distribution is controlled for, and demonstrate practical methods for improving reward models through variance manipulation. The score would move to "Borderline Accept" (4) if you address the confounding issues with additional controlled experiments, even without demonstrating practical applications.

**Ethical Concerns:**

["NO or VERY MINOR ethics concerns only"]

**Final Justification:**

I have increased my score to 4. The authors have addressed my primary concern regarding the confounding of data distribution and reward variance. Their new experiments, which trained multiple reward models on an identical data distribution while varying hyperparameters to produce different variance-accuracy trade-offs, provide the necessary controlled comparison. While I still have some reservations about the practical applicability and the full explanatory power of the current theoretical framework, the additional evidence is enough to show that reward variance is a significant factor, independent of data distribution.

**Limitations:**

The authors acknowledge some limitations but fail to address the most critical issue: the confounding between data distribution effects and variance effects. This represents a fundamental threat to the validity of the core claims.

**Paper Formatting Concerns:**

No major formatting issues observed.

**Quality:**

2

**Strengths And Weaknesses:**

**Strength:**

**S1. Novel theoretical framework:** The optimization perspective on reward model evaluation is original and theoretically rigorous. The mathematical analysis connecting reward variance to gradient flow dynamics is well-executed.

**S2. Important research question:** Understanding what makes reward models effective beyond accuracy is practically relevant for RLHF practitioners.

**S3. Comprehensive theoretical coverage:** The progression from general autoregressive to tabular policies provides thorough theoretical grounding.


**Weakness:**

**W1. Fundamental Experimental Confounding:** The core experimental design conflates data distribution effects with reward variance effects. Models with higher variance (100% on-policy) were trained on the target distribution, while lower variance models (0% on-policy and ArmoRM) face out-of-distribution evaluation. This confounding makes it impossible to determine whether performance gains come from variance itself or from better distribution matching—a well-established factor in model performance.

- **Missing Controls:** The paper lacks experiments that isolate variance effects, such as artificially manipulating variance via temperature scaling while keeping data distribution fixed, or comparing multiple models trained on identical data.

- **Correlation vs Causation:** While the paper demonstrates correlation between variance and performance, it cannot establish that variance—rather than domain adaptation—drives the observed benefits.

**W2. Theory-Practice Validation Gap**

The paper fails to provide convincing positive validation of its core theoretical claims:
- **Contradictory evidence:** In the limitations section, the authors mention that directly scaling rewards by a factor >1 (which increases variance) did not yield benefits, contradicting their theoretical predictions.
- **Insufficient analysis:** This negative result lacks detailed explanation or analysis, leaving a critical theory-experiment mismatch unresolved.
- **Limited practical guidance:** While the paper shows that artificially reducing variance hurts performance (unsurprising), it provides no validated methods for beneficially increasing variance—the practically relevant direction.

**W3. Questionable Ground Truth Validation**

**Unvalidated Ground Truth:** The paper uses ArmoRM as "ground truth" without validating its alignment with actual human preferences or objective correctness. This creates potential cascading errors throughout the experimental evaluation.

**Better Alternatives Available:**
- Human evaluation to verify the ground truth model's actual accuracy
- Stronger generative models (e.g., Claude-4 Sonnet, DeepSeek-R1) as more reliable ground truth
- Tasks with verifiable ground truth (e.g., code generation, mathematical problems) where correctness can be objectively determined

To validate the core hypothesis, the paper needs experiments that:
- Artificially increase variance through methods like temperature scaling while maintaining identical training distributions
- Compare multiple models trained on identical data to isolate variance effects from accuracy effects
- Use tasks with verifiable ground truth (e.g., code generation, mathematical reasoning) where correctness can be objectively determined or use stronger generative models (e.g., Claude-4 Sonnet, DeepSeek-R1) as more reliable ground truth

---

> ### Author Rebuttal · Authors · 2025-07-30
>
> Thank you for the thoughtful feedback and for highlighting the importance of our results. We treat your comments and questions below. In particular, we emphasize that to a large extent the concerns raised in the review are already accounted for in the paper and provide additional experimental results, as requested in the review. If our response addresses your concerns, we would greatly appreciate it if you would consider raising your score. Please let us know if you have any further questions; we will gladly respond.
>
> &nbsp;
>
> **W1 and Q1: Experiments isolating confounding factors.**
> We first want to emphasize that **the paper already includes experiments that isolate the effects of reward variance** from potential confounding factors (e.g., data distribution over which the reward model was trained). Specifically, in the experiments of Section 4.1 and Appendix F.1, the perfectly accurate reward model with low reward variance (e.g., the red reward model in Figure 2) is obtained from the ground truth reward by reducing in a controlled manner how well it separates different outputs while preserving their rankings. Thus, the ground truth reward (in yellow) and the perfectly accurate reward model with low reward variance (in red) were trained on the same data, have the exact same (perfect) accuracy, yet differ in their reward variance due to the controlled intervention. As our results show (e.g., Figure 2), the perfectly accurate reward model with low reward variance substantially underperforms the ground truth reward, in the sense that it leads to substantially slower optimization.
>
> &nbsp;
>
> **W1 and Q2: Further experiments comparing reward models trained on the same data.**
> This is a good point! Thank you for raising it. Studying how reward variance and accuracy affect the outcome of RLHF requires reward models that vary in these properties. Our approach of training reward models on different data mixtures is one way of achieving this. Indeed, it is also possible to train reward models over the same data and vary instead different training hyperparameters.
>
> We first note that, as discussed in the response to the previous comment, Section 4.1 and Appendix F.1 already include experiments that demonstrate the effect of reward variance for reward models trained on the same data distribution. To further address your concern, as requested in the review, during the response period **we conducted additional experiments with reward models trained on the same data distribution** (details and results are provided below). The results are analogous to the ones reported in Section 4.1, demonstrating that the connection between reward variance on optimization persists when reward models are trained on the same data.
>
> *Details:* We considered a setting similar to Section 4.1 and trained, based on the Pythia-1B language model, 10 reward models over the same data (100% on-policy) using different training hyperparameters (e.g., learning rate and reward centering regularization coefficient). To obtain results in time for the response period, using each reward model we carried out a single policy gradient run as opposed to three (note that, as shown in the paper, results were typically stable across random seeds). Analogously to Section 4.1, we found that more accurate reward models are not necessarily better – e.g., a reward model with 0.67 accuracy but low reward variance led to an increase of 0.07 in ground truth reward while another reward model with 0.65 accuracy and higher reward variance led to an increase of 0.12 in ground truth reward. Furthermore, the table below presents the correlation between different reward model properties and increase in reward used for training and ground truth reward. As was the case in our original experiments, reward variance is highly correlated with increase in reward (used for training), while accuracy was not, with a combination of both being most indicative of increase in ground truth reward.
>
> |                                | Reward Increase Spearman | GT Reward Increase Spearman |
> |--------------------------------|---------------------------|------------------------------|
> | Reward Variance                | 0.588                     | 0.576                        |
> | On-Policy Acc.                 | -0.696                    | 0.519                        |
> | Reward Variance & On-Policy Acc. | 0.455                     | 0.661                        |
>
> Note 1: We omitted off-policy accuracy for brevity. It had lower correlations than on-policy accuracy.
>
> Note 2: The reward models were trained using learning rates 1e-7, 5e-7, and 1e-6, reward centering regularization coefficients 0, 0.1, 10, and scaling factors applied to the rewards of a fraction of the prompts (after scaling we still normalized reward models such that on average over prompts they have roughly the same scale).
>
> &nbsp;
>
> **W2 + Q3: Theory-practice validation gap.**
> We respectfully disagree that the “paper fails to provide convincing positive validation of its core theoretical claims”. Each of our theoretical claims is backed by extensive empirical evidence in Sections 4.1 and 4.2 and Appendix F. Specifically, our theory establishes that: (i) low reward variance can lead to slow optimization, (ii) more accurate reward models are not necessarily better teachers, and (iii) for different language models different reward models can work better. Our experiments clearly demonstrate that these conclusions apply in practical RLHF settings.
>
> Furthermore, the discussion in the future work section on why naively scaling the reward by a constant $c > 1$ is problematic **does not contradict our theoretical analysis**. Namely, by no means does our analysis suggest that naively scaling the reward by a large factor will improve performance. As explained in the future work section, when the reward variance is low the expected reward gradient vanishes. Thus, sample-based estimates of the gradient may be dominated by noise, which will be amplified by such reward scaling. We note that for reward variance to be a meaningful measure that allows comparing reward models, one must normalize reward models to have roughly the same scale, which is standard in practice and is already accounted for in our experiments and theoretical analysis. In light of your comment, we will clarify this point in the manuscript.
>
> Regarding practical guidelines. The main purpose of our work is to advance the understanding of what makes a good reward model. While it does not give a solution to the issue of low reward variance, it highlights limitations of existing reward model benchmarks and provides insight into what determines the effectiveness of a reward model, beyond accuracy. We hope that our work inspires further research into reward model training procedures that encourage higher reward variance, as well as policy gradient methods that can optimize more efficiently under low reward variance. The future work section outlines a couple of possible directions going forward. However, due to the extent of our theoretical and empirical results, we believe that this important and highly non-trivial endeavour falls outside the scope of the current paper.
>
> &nbsp;
>
> **W3 + Q4: Ground truth validation.**
> As discussed in Section 4, since evaluating policies via human labelers can be prohibitively costly, it has become common practice to use an existing large reward model as the ground truth reward (e.g., see references [27, 13, 74, 7, 10, 81, 73] in the paper). This is especially true for studies, such as ours, whose main goal is to further the understanding of different aspects of the RLHF pipeline, as opposed to training state-of-the-art models. The ArmoRM reward model in particular has already been used as the ground truth reward in numerous papers, and was validated in [77] to accurately accord with existing labeled preferences (better than GPT4 judges).
>
> Moreover, using an existing reward model as the ground truth reward actually allows avoiding any potential “cascading errors” and inconsistencies due to noise in human preferences, since the exact same ground truth reward is applied for both labeling preferences and evaluating policies. Thus, **in our experiments, the ground truth reward already plays the role of a verifiable reward**, as you suggested we use in your review. Furthermore, note that we also verified our results are robust to the choice of ground truth reward — Appendix F provides experiments showing analogous results with a different ground truth reward.

---

> > ### Comment · Reviewer_8fPr · 2025-08-03
> > **Official Comment by Reviewer 8fPr**
> >
> > Thanks for your response and an additional experiment. We understand that due to the extent of your theoretical and empirical results, some problems are outside the scope of the current paper. We respect your solid theoretical and empirical results overall, but we also have some doubts.
> >
> > **About W1 and Q1: Experiments isolating confounding factors.**
> >
> > You only mentioned some of the experiments did control confounding factors, but did not directly reply to "Models with higher variance (100% on-policy) were trained on the target distribution, while lower variance models (0% on-policy and ArmoRM) face out-of-distribution evaluation. This confounding makes it impossible to determine whether performance gains come from variance itself or from better distribution matching—a well-established factor in model performance." Therefore, I suppose these experiments did not control confounding factors.
> >
> > **About W1 and Q2: Further experiments comparing reward models trained on the same data.**
> >
> > Experiments lack detail: did not tell hyperparameters when training each reward model, as well as accuracy and variance of each reward model after training, and reward increase of each policy model.
> > In addition, only emphasizing correlation of Reward Increase and GT Reward Increase doesn't mean much, as reward increase could be result of reward hacking, thus not representing real performance improvement in downstream tasks. Therefore, it's needed to test actual performances on downstream tasks after policy model is trained by these reward models.
> >
> > **About W2 + Q3: Theory-practice validation gap.**
> >
> > By "The paper fails to provide convincing positive validation of its core theoretical claims", I meant:
> >
> > (1) Your "theoretical claims" include "Reward variance strongly correlates with reward increase" in table 2 caption, suggesting higher variance would lead to faster reward increase.
> >
> > (2) In Appendix B, your explanation to "directly scaling rewards by factors >1 did not yield benefits" is "when the reward variance is low, the expected reward gradient vanishes. Thus, sample-based estimates of the gradient may be dominated by noise, which will be amplified by such reward scaling." This does not fit in your framework, instead it uses concept of "gradient vanish". I suppose this means your theory is not able to explain why multiplying all rewards by a factor of c > 1 cannot improve training efficiency.
> >
> > (3) Moreover, can "low reward variance lead to slow optimization" be eased by increasing learning rate?
> >
> >
> > Thanks for your response and an additional experiment. Looking forward to your further discussion.

---

> ### Author Response · Authors · 2025-08-03
> **Follow Up (Part 1/2)**
>
> Thank you for the swift response! We greatly appreciate your engagement and address your additional questions below. We believe that this should treat any remaining doubts.
>
> **W1 and Q1.**
> As detailed in Section 4.1 and noted in your review, our experiments indeed contained reward models trained on different data mixtures. This is one possible approach to obtain reward models that differ in terms of their accuracy and reward variance. However, as specified in our original response above, the paper already contains experiments accounting for this potential confounding factor.
>
> Furthermore, the previous response provided additional experiments, conducted per your request, with reward models that are trained on the exact same data, but with different training hyperparameters. The results demonstrate the validity of our conclusions regarding the interplay between accuracy, reward variance, and optimization with reward models that are trained on the same data.
>
> &nbsp;
>
> **W1 and Q2.**
> We gladly provide in part 2 of the response the full implementation details of our additional experiments, including the hyperparameters used for training the reward models, the accuracy and reward variance of these models, and reward increase after policy gradient. These details were omitted from our first response due to space limitations.
>
> As for reward hacking, we believe that there may be a misunderstanding with regards to what the ground truth reward stands for in our experiments. **The ground truth reward stands for the true “downstream” performance.** As discussed in the paper and our previous response, the approach of using an existing reward model as the ground truth reward is common in the literature. Since the ground truth reward is used for both labeling preferences and measuring ground truth performance, this methodology avoids any potential inconsistencies or reproducibility issues due to noise in human preferences, while circumventing the prohibitive cost of using human labelers. Note that reward hacking occurs when the reward used for training increases, but the ground truth reward decreases. This is why we report both rewards — it allows examining the effect on optimization speed, as manifested in the reward used for training increase, as well as how that translates to ground truth improvements.
>
> &nbsp;
>
> **W2 + Q3.**
> Thank you for clarifying your comment. Our theory establishes that low reward variance leads to slow optimization. Yet, as discussed in Appendix B, this does not imply that any method for increasing reward variance necessarily leads to faster optimization. Specifically, our experiments demonstrate, using reward models that we trained as well as publicly available reward models, that reward variance is indicative of the rate at which the reward increases. However, artificially inflating reward variance by scaling rewards by a large factor $c > 1$ can lead to instabilities. We therefore view developing methods for increasing reward variance in a meaningful way (e.g., higher reward variance at a given reward scale) as a valuable direction for future work.
>
> Note that scaling rewards by a factor $c > 1$ has an effect similar (or identical, depending on the particular policy gradient implementation) to scaling up the learning rate. Indeed, simply scaling them up does not give a “free lunch” of faster optimization. As mentioned in Appendix G.1, we experimented with different learning rates and found that analogous results are obtained when using reasonable values, though too large values led to instability.
>
> Lastly, the explanation based on vanishing gradient does fall under our theoretical framework. In particular, the proof of Theorem 1 builds on the fact that the expected gradient and higher-order derivatives vanish when the reward variance is low. The arguments on why in this case scaling up rewards can lead to instabilities in practice due to noise in inexact batch gradients are made on an intuitive level. Formally extending our analysis to the inexact gradient setting is highly non-trivial and we believe that it could be an interesting challenge for future work, which may further shed light on the relative advantages and disadvantages of different policy gradient methods.
>
> **Conclusion.**
> We would like to thank you again for the engagement and for taking the time to read our responses. Please let us know if you have any further questions.

---

> > ### Author Response · Authors · 2025-08-03
> > **Follow Up (Part 2/2)**
> >
> > **Experimental details.**
> >
> > **Table 1.** Accuracy and reward variance of the ten reward models considered in the additional experiments.
> >
> > |                                      | RM 1  | RM 2  | RM 3  | RM 4  | RM 5  | RM 6  | RM 7  | RM 8  | RM 9  | RM 10 |
> > |--------------------------------------|-------|-------|-------|-------|-------|-------|-------|-------|-------|--------|
> > | Reward Variance                      | 0.599 | 0.799 | 0.459 | 0.700 | 0.758 | 0.571 | 0.601 | 0.620 | 0.659 | 0.735  |
> > | On-Policy Acc.                       | 0.566 | 0.652 | 0.657 | 0.601 | 0.667 | 0.667 | 0.667 | 0.667 | 0.667 | 0.667  |
> > | Reward Variance & On-Policy Acc.     | 0.582 | 0.726 | 0.558 | 0.651 | 0.712 | 0.619 | 0.634 | 0.644 | 0.663 | 0.701  |
> >
> > ---
> >
> > **Table 2.** Increase in reward used for training and ground truth reward after policy gradient for each reward model.
> >
> > |                                   | RM 1   | RM 2   | RM 3   | RM 4   | RM 5   | RM 6   | RM 7   | RM 8   | RM 9   | RM 10  |
> > |-----------------------------------|--------|--------|--------|--------|--------|--------|--------|--------|--------|--------|
> > | Increase in Reward Used for Training | 0.797  | 0.921  | 0.720  | 0.879  | 0.822  | 0.106  | 0.267  | 0.385  | 0.586  | 0.711  |
> > | Increase in Ground Truth Reward     | -0.021 | 0.123  | 0.104  | 0.035  | 0.115  | 0.066  | 0.122  | 0.113  | 0.119  | 0.136  |
> >
> > ---
> >
> > **Reward models.** Below we specify the hyperparameters that differed from our original experiments (see Appendix G.1) for each reward model (numbered from 1 to 10).
> > 1. RM 1: The center_rewards_coefficient hyperparameter was set to 10.
> > 2. RM 2: The center_rewards_coefficient hyperparameter was set to 0.1.
> > 3. RM 3: The center_rewards_coefficient hyperparameter was set to 0.
> > 4. RM 4: The learning rate was set to 1e-7.
> > 5. RM 5: The learning rate was set to 1e-6.
> > 6. RM 6: The learning rate was set to 1e-6 and the rewards for 90% of the prompts were scaled down by a factor of 0.01.
> > 7. RM 7: The learning rate was set to 1e-6 and the rewards for 70% of the prompts were scaled down by a factor of 0.01.
> > 8. RM 8: The learning rate was set to 1e-6 and the rewards for 50% of the prompts were scaled down by a factor of 0.01.
> > 9. RM 9: The learning rate was set to 1e-6 and the rewards for 30% of the prompts were scaled down by a factor of 0.01.
> > 10. RM 10: The learning rate was set to 1e-6 and the rewards for 10% of the prompts were scaled down by a factor of 0.01.
> >
> > Note: After scaling rewards for a fraction of the prompts in RMs 6 to 10, we normalized rewards to ensure that on average over prompts all reward models have roughly the same scale. This type of scaling allows modifying reward variance in a controlled manner.

---

> > > ### Author Response · Authors · 2025-08-07
> > >
> > > Dear reviewer,
> > >
> > > We are grateful for your feedback and thank you again for the time and effort in reviewing our paper. We believe that our responses above fully address the points raised in the initial review and follow up questions. Since the discussion period is nearing its end, we kindly ask that you let us know whether you have any remaining questions or comments.
> > >
> > > Thank you!

---

> > > > ### Comment · Reviewer_8fPr · 2025-08-08
> > > > **Official Comment by Reviewer 8fPr**
> > > >
> > > > I have increased my score to 4. The authors have addressed my primary concern regarding the confounding of data distribution and reward variance. Their new experiments, which trained multiple reward models on an identical data distribution while varying hyperparameters to produce different variance-accuracy trade-offs, provide the necessary controlled comparison. While I still have some reservations about the practical applicability and the full explanatory power of the current theoretical framework, the additional evidence is enough to show that reward variance is a significant factor, independent of data distribution.

---

> > > > > ### Author Response · Authors · 2025-08-09
> > > > >
> > > > > Thank you for the fruitful and insightful discussion, which has helped clarify the significance of our results.

---

### Official Review · Reviewer_9BJi · 2025-07-02

**Clarity:** 3
**Significance:** 3
**Originality:** 3
**Rating:** 5
**Confidence:** 5

**Summary:**

The paper analyses the necessary properties for a reward model to enable effective policy optimisation in the RLHF setting. The paper demonstrates that the accuracy of the reward model is not the only factor in policy optimisation success, and that the reward variance induced by the reward model for a given policy is equally or even more important. The paper defines and investigates this reward variance in more detail, and demonstrates theoretically and empirically that low reward variance can lead to ineffective policy optimisation.

**Questions:**

none

**Ethical Concerns:**

["NO or VERY MINOR ethics concerns only"]

**Final Justification:**

I maintain my justification from the review, as I had few issues with the paper.

**Limitations:**

yes

**Paper Formatting Concerns:**

no concerns

**Quality:**

3

**Strengths And Weaknesses:**

# Strengths

* The paper is clear and well-written.
* The area of study is important and significant, and the contribution of the paper is meaningful in this area.
* The paper's approach to try and understand the topic of study rather than merely increase benchmark scores is beneficial.
* The analysis of the paper is insightful and interesting, and sheds light on a relatively lightly understood but highly important area (RLHF). The theoretical and empirical demonstration of the discussed phenomena is compelling and beneficial.

# Weaknesses

* The paper has relatively few weaknesses. It would be beneficial to discuss how to achieve higher reward variance during reward model training, and why rescaling the reward function in various ways is insufficient (if it is).

---

> ### Author Rebuttal · Authors · 2025-07-30
>
> Thank you for the support and for highlighting the significance and clarity of our work!
>
> The future work section, which was deferred to Appendix B due to lack of space, discusses why simply scaling up the reward is insufficient and lays out potential avenues for increasing reward variance in a meaningful way. We plan to expand this discussion and incorporate it into the body of the paper given the additional content page allowed for accepted papers. Thank you for the suggestion!
>
> If any questions arise during the discussion period, please let us know and we will gladly respond.

---

### Official Review · Reviewer_ngKr · 2025-07-13

**Clarity:** 4
**Significance:** 3
**Originality:** 3
**Rating:** 5
**Confidence:** 4

**Summary:**

Previous work argues that, when evaluating reward models (RMs) for RLHF, we should not just focus on the RM's accuracy, that is; how often it correctly picks the higher reward completion from a (prompt, (completion1, completion2)). Instead, what really matters is *how much the RM induces the LLM policy to obtain higher true reward* (final performance).

This paper then focuses on the variance of the RM under the base policy as a component that can affect the final performance, independently of the RM's accuracy. They support this with a theorem on gradient flow RLHF, showing that the time taken scales as Variance^{-1/3}, and with empirical experiments showing that the RM's variance is correlated with the LLM's final performance at close to rho=1, and that the highest-accuracy RM does not always train the LLM the best.

**Questions:**

As explained in detail above, this paper claims the following hypothetical causal model:

RM variance on-policy -> different final LLM accuracy

I believe this other hypothesis is highly plausible, the theory or experiments don't give it a higher likelihood ratio than the previous one, and gives very different takeaways:

scale of the RM on-policy -> RM variance on-policy -> different final LLM accuracy

Until these hypotheses are disentangled, or the authors successfully argue that I'm otherwise wrong, I cannot in good conscience let this paper be accepted. If this is successfully addressed, I think this is a good paper and will argue that it be accepted.

## Less relevant questions:

- In Table 2, how can the Spearman Correlation be literally 1.000 for variance vs. ground truth reward, when in Fig 2 the 75% model obtains more GT reward than the 100% model? We know the 100% model has more variance from Table 1. If the correlation was 1.000, this model should have more GT reward.

- A possible clarity improvement. I initially could not see why RM variance would affect results, but this intuition helped me, it could go in the intro. If there is more variability in rewards depending on different trajectories, there is more information for the LLM about which trajectories to take.

**Ethical Concerns:**

["NO or VERY MINOR ethics concerns only"]

**Final Justification:**

The authors have shown to my satisfaction their thesis: that reward variance affects optimization, beyond just the initial scale of the reward model. Thus I am happy to accept this paper.

**Limitations:**

No, see above.

**Paper Formatting Concerns:**

no concerns

**Quality:**

3

**Strengths And Weaknesses:**

# Strengths
The paper makes a very good point that, beyond accuracy, a RM variance matters a lot to the performance of the final post-trained LLM. They support this with a theorem and thorough evaluations on a relatively small but still meaningful LLM (Pythia 2.8B). In this respects, it is a well-written, solid paper; that addresses an important problem and makes an original point (high Clarity, Significance and Originality).

# Weaknesses
Unfortunately, the paper in its current form has one large weakness. What if the effect of increased variance in LLMs affects the performance *solely due to* an increase in effective learning rate? This would make the takeaways from the paper considerably different: it's not that we should train the RMs to have high variance, or evaluate it; it's that we should normalize the RM's variance **on-policy** to be 1, so we can compare tuned learning rates and other parameters. We can normalize at the beginning, or continue normalizing during training so the RM's variance stays 1.

Nothing in the theory or experiments can discard this explanation, because the only independent variable in the experiments is the variance, which is very affected by the scale. If we multiply the RM by c, its variance is multiplied c^2. At the same time the policy gradients objective gets multiplied by c , thus increasing the effective learning rate of SGD by c as well.

This would perfectly explain that the theoretical time taken by gradient flow scales with Var^{-1/2} (slightly larger than the paper's lower bound of Var^{-1/3}). In the experiments, Table 1 lists on-policy variances of the RMs (which get smaller as the RM's training data becomes more out of RLHF-distribution), and these are ~perfectly correlated with the model. Again this could be because of the scale only: L260-263 states reward models are normalized, but this is done on off-policy data.

To address this, the experiments could have two treatments: one in which they normalize the RMs so they have the same variance *on-policy* and one in which the RMs are left unnormalized. If the normalization makes it so that the RMs are essentially equal, or that RM accuracy then fully explains the results, then it's the scale of the RM that is affecting things. (Though necessarily, this would completely destroy any association of the variance with the final performance: because the variance would not change. Perhaps the relevant quantity would be come whether the variance comes from occasional extreme datapoints, or whether datapoints are relatively well-distributed across the reward range. Experiments could look at the reward histograms, since they're 1D, and check.)

Another possible experiment is to search over a scale hyperparameter alpha and set RM' = alpha * RM, which should show that for the same learning rate the best induced variances are similar for all the RMs.


Under the scale explanation, the theory should probably be deemphasized: it is fairly obvious that scaling the reward by c necessitates adjustments to the learning rate.

You could even attempt to show extra improvements from continuously normalizing the RM during training. But this could instead be a neat paper that shows that the scale of your RM is very important for its effectiveness.

---

> ### Author Rebuttal · Authors · 2025-07-30
>
> Thank you for the thoughtful feedback and for highlighting the clarity, significance, and originality of our work! Your review focuses on an important, yet nuanced, point regarding the relation between reward scale, “effective learning rate”, and reward variance. As detailed below, **the effect of reward variance on optimization is not solely due to differences in reward scale/effective learning rate**. In fact, for reward variance to be a meaningful measure that allows comparing reward models, rewards need to be normalized to have roughly the same scale. The paper already accounts for this and we further provide below additional empirical evidence to address your comments. Nonetheless, we recognize that the discussion of this point in the paper can be improved (currently, we mention it in the experimental section and future work section). In light of your review, we will incorporate a dedicated discussion of it after Definition 2. We hope that our response fully addresses your concern and would greatly appreciate it if you would consider raising your score. Please let us know if you have any further questions; we will gladly respond.
>
> &nbsp;
>
>
> ### **Effect of reward variance on optimization is not just due to scale**
>
> First, we would like to emphasize that reward variance captures more than just scale: two reward models can have roughly the same scale but different reward variances (note that reward variance can be low even when the reward scale is not small). For reward models of roughly the same scale, reward variance measures the degree of separation between outputs. A useful analogy is margin in linear prediction. Just as the linear predictor needs to be of normalized scale for margin to be a meaningful measure, the scale of rewards needs to be normalized for reward variance to faithfully measure degree of separation between outputs.
>
> Indeed, as mentioned in the review, scaling up the rewards affects the effective learning rate and reward variance. Accordingly, to avoid differences in effective learning rate and allow comparing reward models based on reward variance, we followed common practice and normalized rewards to have roughly the same scale. In general, there does not exist a scale $\alpha$ such that for two reward models RM and RM’ it holds that RM’ = $\alpha$ * RM. So usually rewards are normalized such that if two reward models did satisfy RM’ = $\alpha$ * RM, then after normalization they would be exactly equal.
>
> - As detailed in Appendix G, we normalized reward models based on outputs sampled from the initial policy over prompts in the training set. This ensured that the reward scale, i.e., average of absolute rewards, was roughly the same for all reward models. Interestingly, in the experiments of Figure 2, after normalization the reward models with lower reward variance (e.g., the perfectly accurate RM) had a slightly higher scale than reward models with a higher reward variance (e.g., RM 100% on-policy). Yet, the former still led to slower optimization. **Thus, the results of Figure 2 already demonstrate that the faster optimization of reward models with a higher reward variance is not due to a larger reward scale**.
>
>     *Details:* Reward scale, i.e., average absolute reward over training prompts and outputs from initial policy, for the reward models in Figure 2 (after normalization).
>     | RM 0% | RM 25% | RM 50% | RM 75% | RM 100% | Perfect RM w/ low variance | GT reward |
>     |-------|--------|--------|--------|---------|------------------------|----------------------|
>     | 0.8015| 0.7783 | 0.7834 | 0.8042 | 0.7917  | 0.8411                 | 0.8254               |
>
>
>
> - The normalization above is computed based on the initial policy. It is also possible to normalize rewards online. **We ran additional experiments, where in each batch rewards are normalized online** by the standard deviation across the batch of prompts and outputs. As shown below, the results are analogous to ones reported in the paper. This **further shows that differences in reward scale do not explain the relation between reward variance and optimization** in our results.
>
>     *Details:* To obtain results in time for the response period, we considered a setting similar to Section 4.1 with Pythia-1B models and ran a single policy gradient run per reward model as opposed to three (though, as shown in the paper, results were typically stable across random seeds). As was the case when normalizing rewards based on initial policy statistics, the perfectly accurate reward model led to a slower increase in ground truth reward than the reward models trained on 100%, 75%, and 50% on-policy data. Furthermore, reward variance was highly correlated with reward (used for training) increase, while accuracy was not, with a combination of both being most indicative of ground truth reward increase.
>
>     |                                | Reward Increase Pearson | Reward Increase Spearman | GT Reward Increase Pearson | GT Reward Increase Spearman |
>     |--------------------------------|--------------------------|---------------------------|-----------------------------|------------------------------|
>     | Reward Variance                | 0.937                    | 0.828                     | 0.292                       | 0.771                        |
>     | On-Policy Acc.                 | -0.529                   | -0.085                    | 0.416                       | 0.657                        |
>     | Reward Variance & On-Policy Acc. | 0.781                    | 0.714                     | 0.880                       | 0.942                        |
>
>     Note: We omitted off-policy accuracy for brevity. It had lower correlations than on-policy accuracy.
>
>
> &nbsp;
>
>
> ### **Challenges in making online reward variance normalization work**
>
> The above showed that the relation between reward variance (after normalization) and optimization cannot be explained by reward scale. Instead of normalizing rewards to have the same scale, an alternative option is to normalize rewards online so that each prompt in a batch has an empirical reward variance of 1. We believe that this is the experiment described in the review.
>
> First, it is important to note that this approach is not captured merely by reward scale or an effective learning rate since it is an **online per-prompt adaptive normalization** scheme. Second, we experimented with such approaches during our work on the project. However, **we found that such normalization can be problematic due to the difference between the empirical reward variance, which is estimated online over a few outputs, and the true reward variance** (results given below). Our results suggest that the reason is similar to why scaling rewards by a large factor $c > 1$ does not help, which we discussed in the future work section (Appendix B). Namely, the expected gradient vanishes when the true reward variance is low. Thus, gradient estimates based on a few online samples may be dominated by noise, which will be amplified by scaling. We believe that a promising direction for future work can be to further investigate whether similar normalization techniques, along with substantially increasing the number of outputs sampled online per prompt, can lead to better performance within a given compute budget.
>
> *Details:* In the setting of Section 4.1, we normalized online the rewards of each prompt separately to ensure its empirical reward variance is one. We ran experiments using two reward models – the ground truth reward and the 75% on-policy reward model – and different numbers of outputs sampled per prompt (2, 8, 16). The table below reports the increase in ground truth reward for each experiment. Notably, with 2 outputs normalization harms performance. Increasing the number of outputs sampled for each prompt led to higher ground truth reward. Yet, even with 16 outputs the imperfect reward model still worked better than using the (perfectly accurate) ground truth reward directly.
>
> |           | 2 Outputs (unnormalized) | 2 Outputs | 8 Outputs | 16 Outputs |
> |---------|--------------------------|-----------|-----------|------------|
> | RM 75% | 0.125                    | 0.115     | 0.155     | 0.152      |
> | GT          | 0.111                    | 0.092     | 0.143     | 0.145      |
>
>
> &nbsp;
>
> ### **Theory**
>
> In our analysis, we account for reward models needing to be of a similar scale by assuming that rewards are bounded within some interval – a standard assumption in the reinforcement learning literature. This precludes circumventing the lower bound in Theorem 1 by artificially multiplying all rewards by a large factor.
>
> &nbsp;
>
> ### **Less relevant questions**
>
> 1. As reported in Table 2, the Spearman correlation is 1 for the increase in reward used for training. For the ground truth reward it is 0.714.
>
> 2. Thank you for the suggestion! This can be useful intuition alongside the connection to the objective landscape. We note, however, that it does not fully capture the reason why reward variance affects optimization. The connection between reward variance and optimization depends on the fact that language models produce distributions through a softmax function (as mentioned in the proof sketch of Theorem 1). In theory, for other, non-practical, parameterizations (e.g., each token probability is its own trainable parameter), the objective landscape is not flat when the reward variance is low. Though, we are not aware of any practical replacement for softmax that avoids this issue, so exploring whether one can construct such replacements can be a valuable direction for future work.

---

> > ### Comment · Reviewer_ngKr · 2025-08-04
> > **I'm still unconvinced that the conclusion is correct, scale can fix it**
> >
> > > Just as the linear predictor needs to be of normalized scale for margin to be a meaningful measure, the scale of rewards needs to be normalized for reward variance to faithfully measure degree of separation between outputs.
> >
> > That's true, but I think *the thing to normalize with* is a very important choice that can lead us to different conclusions.
> >
> > I really appreciate your willingness to do more experiments, but I'm still unconvinced that "Effect of reward variance on optimization is not just due to scale". My current belief is that:
> >
> > 1. the variance of the reward model *on-policy* is very correlated with the final reward (arguably shown by the paper, though the paper looks at *off-policy* reward AFAICT.)
> > 2. By correcting the scale (and bias, my bad for not pointing this out earlier) of the various reward models so that their variance is the same *on-policy* at the start of training, any explanatory power of the *off-policy variance* goes away.
> >
> > Also, that the reason online scaling doesn't work has been very well explored by you in this rebuttal, so I'm convinced it's because it's hard to estimate the scale.
> >
> > This is to say, I think the conclusion of your paper is slightly off, but your contribution is still great! I have only started to consider this as a possible explanation because of the analysis and results you've already presented. But I think it's important to get the causal mechanism and conclusion correct, so that future readers extract the correct conclusions from this paper.
> >
> > **Could you please 1) measure the on-policy variance at the start of training of each reward model, so we can check how correlated it is with the final increase in training and GT reward, and 2) attempt another reward training run with Pythia-1B, using reward models that have been normalized so their on-policy (at the start of training) mean and variance are the same?**
> >
> > My guess is this will make all the correlations based on the initial measured variance of the models go away. The only thing that will remain correlated with the final reward increase, is how much % on-policy is the training data of the model.
> >
> > I believe the outcome of this should greatly affect the message of the paper, because if you can correct the reward scale to get good training then it's good to know.
> >
> > Either way this experiment comes up I'm happy to correct the score and recommend accept, but I'd really like this conclusion to be correct.
> >
> > > Details: Reward scale, i.e., average absolute reward over training prompts and outputs from initial policy, for the reward models in Figure 2 (after normalization).
> >
> >
> > # Minor things on which I spent a long time
> >
> > > As reported in Table 2, the Spearman correlation is 1 for the increase in reward used for training. For the ground truth reward it is 0.714.
> >
> > Thank you for this correction. I misread that table entirely.
> >
> > > We note, however, that it does not fully capture the reason why reward variance affects optimization. The connection between reward variance and optimization depends on the fact that language models produce distributions through a softmax function (as mentioned in the proof sketch of Theorem 1). In theory, for other, non-practical, parameterizations (e.g., each token probability is its own trainable parameter), the objective landscape is not flat when the reward variance is low.
> >
> > I disagree specifically with: "In theory [...] the objective landscape is not flat when the reward variance is low". It is straightforward to show that scaling the reward by some c^2, as c->0, makes it both so that 1) the variance goes to 0 and 2)  the policy gradient E[r' * \nabla log p] also goes to 0. This is completely independent of what parameterization for p you're using: because p doesn't change as you change the variance of r, the landscape does become flat. That is, so long as you're happy to assume Lipschitz continuity of the parameterization, which is unreasonable not to assume.
> >
> > I agree with this very conditional statement: if you want your bound to depend on the Jacobian up until the parameterization only, if you want to prove absolute bounds on the gradient like [65] (or on the optimization time, like you do), then you need to use specific properties of the softmax.
> >
> > (I dug into the proof of Theorem 1, which cites [65] for bounding the variance of the policy gradient with the variance of the reward, when using the softmax only. For why it's the "softmax only", [65, section 3] says to look in section D.1. I can see that indeed the softmax's gradient is being used in the proof, specifically to bound the gradients by 2\gamma in the equation above eq. 5 in D.1 and in the second branch also to bound something depending on \gamma. So indeed, you need the softmax to bound the total variance of the gradient).

---

> > > ### Author Response · Authors · 2025-08-05
> > > **Follow Up (Part 1/2)**
> > >
> > > Thank you for taking the time to read our response in detail! We greatly appreciate your engagement in a good-faith discussion, and willingness to update your score and recommend acceptance. We believe that the response below fully addresses any remaining concerns; please let us know if you have any further questions.
> > >
> > > **Reward variance vs scale.** After reading your additional comments closely, **it seems that we are actually in agreement with respect to what our results imply, and our perspectives only differ in terms of how we define reward scale**. To us, one straightforward definition of reward scale is the mean absolute reward. In that case, as detailed in our previous response, the experiments of Section 4.1 show that reward variance is indicative of reward increase even when accounting for scale. However, seeing your comment on correcting bias, it seems that you consider reward scale to be the expected absolute deviation of the reward from its mean. This quantity is known as the mean absolute deviation (MAD) and, when the expectations are taken with respect to the current policy, it is intimately related to reward variance. If one defines scale in this manner, then our results can indeed be interpreted in terms of (centered) scale. In fact, for a random variable $Z$ that takes values within $[-1, 1]$, it holds that $(MAD(Z))^2 \leq Var (Z) \leq 2 MAD (Z)$. Thus, one can reframe our results in terms of expected (centered) scale on-policy.
> > >
> > > Thank you for raising this point, we believe that the discussion has helped clarify the implications and potential different interpretations of our work!
> > >
> > > &nbsp;
> > >
> > > **Reward variance is measured on-policy.** Reward variance, as we defined it and measured in our experiments, is an on-policy quantity. In particular, as the caption of Table 1 specifies, the reward variance is measured with respect to the initial policy, i.e., in an on-policy manner at the start of training, and Table 2 specifies correlations of this reward variance with increase in rewards. Thus, **our experiments precisely show that the on-policy reward variance at the start of training is highly correlated with increase in training and ground truth rewards** (request (1) in your reply). We are not sure what may have caused the reviewer to get the impression that our experiments measure reward variance off-policy. Please let us know if there is a particular part in the paper that gave you this impression and we will gladly clarify it.
> > >
> > > &nbsp;
> > >
> > > **Additional experiments.**
> > > As requested (request (2) for additional experiments in your reply), we conducted experiments similar to those in Section 4.1, but with a Pythia-1B model, in which we normalized reward models to have roughly the same reward variance and mean with respect to the initial policy (it is exactly the same over the on-policy outputs sampled to compute the normalization and approximately the same if sampling fresh outputs from the initial policy). This normalization differs from the normalization done in our original experiments, which ensured reward models have roughly the same mean absolute reward, i.e., what we view as reward scale.
> > >
> > > Since after this type of normalization the (empirically measured) reward variance induced for the initial policy is the same across reward models, there is of course no correlation between it and the increase in rewards. However, when considering the correlation between reward variance as measured with our original normalization and reward increase (after training using reward models normalized via the new normalization scheme), the results are analogous to our original experiments. In particular, reward variance is highly correlated with increase in reward, while accuracy was not, with a combination of both being most indicative of increase in ground truth reward. We attribute this observation to the fact that the original normalization accounts for differences in reward scale yet maintains the ability to compare reward models via reward variance, which captures the degree of separation between outputs at a given scale. Along with our previous results, this substantiates the relation between reward variance (after accounting for reward scale) and optimization.
> > >
> > > |                             | Reward Increase Pearson | Reward Increase Spearman | GT Reward Increase Pearson | GT Reward Increase Spearman |
> > > |-----------------------------|--------------------------|---------------------------|-----------------------------|------------------------------|
> > > | Reward Variance             | 0.945                    | 0.943                     | 0.145                       | 0.657                        |
> > > | On-Policy Acc.              | -0.522                   | -0.029                    | 0.412                       | 0.600                        |
> > > | Reward Variance & On-Policy Acc. | 0.800              | 0.771                     | 0.660                       | 0.886                        |

---

> > > > ### Author Response · Authors · 2025-08-05
> > > > **Follow Up (Part 2/2)**
> > > >
> > > > **Minor comment on low reward variance and flat landscape.** It is true that when all rewards are taken to be the same (zero or non-zero) value, the expected gradient vanishes regardless of the policy parameterization. This is because the expected reward becomes constant, and so any policy is optimal. However, this is only a specific case of low reward variance. More broadly (and realistically), reward variance can be low when the rewards for some outputs are substantially different from the rewards of others. Namely, reward variance is low whenever the policy is concentrated on outputs that achieve roughly the same reward, even if there exist other outputs that achieve a substantially higher reward. In those cases, while for softmax policies the gradient vanishes, for other (non-practical) parameterizations it need not vanish. Thus, it is not true that the statement “low reward variance leads to vanishing gradients” applies to any policy parameterization. Note that this statement is different from saying “for any policy parameterization, there exist settings in which the reward variance and gradient both vanish”, which is indeed true.
> > > >
> > > > To give a concrete example, consider the non-practical parameterization mentioned in the previous response, in which each output probability is its own trainable parameter. That is, $\pi_\theta (y) = \theta_y$ (we omit the prompt from the notation for simplicity). For a reward model $r$, the gradient of $E_{y \sim \pi_\theta} [r(y)] = \sum\nolimits_{y \in \mathcal{Y}} \theta_y \cdot r(y)$ is simply a vector in $R^{|\mathcal{Y}|}$ holding the reward that $r$ assigns to each output. This gradient is independent of $\theta$. In particular, if the policy $\pi_\theta$ is concentrated on outputs that achieve roughly (or exactly) the same reward and there exists some other output $y’$ that achieves a substantially higher reward, then the reward variance will be low (or zero) while the gradient will be non-zero.
> > > >
> > > > &nbsp;
> > > >
> > > > **Relation between on-policy data and reward variance.** For the experiments of Section 4.1, to obtain reward models that vary in terms of their reward variance and accuracy, we trained the reward models on different data mixtures. As can be seen in Table 1, the percentage of on-policy data highly correlates with reward variance. This is to be expected since reward variance measures the separation between rewards assigned by the reward model to on-policy responses. Thus, as you rightfully commented, both the percentage of on-policy data and reward variance correlate with reward increase.
> > > >
> > > > We emphasize, however, that Section 4.1 (and Appendix F) includes experiments accounting for this potential confounding factor. Furthermore, per the request of another reviewer, we also conducted additional experiments with reward models trained on the exact same (100% on-policy) data. As detailed below, these experiments demonstrate that the relation between reward variance and optimization holds for reward models trained on the exact same data distribution.
> > > >
> > > > *Existing experiments.* The perfectly accurate reward model with low reward variance (e.g., the red reward model in Figure 2) is obtained from the ground truth reward by reducing in a controlled manner how well it separates different outputs while preserving their rankings. Thus, the ground truth reward (in yellow) and the perfectly accurate reward model with low reward variance (in red) were trained on the same data, have the exact same (perfect) accuracy, yet differ in their reward variance due to the controlled intervention. Nonetheless, as our results show (e.g., Figure 2), the perfectly accurate reward model with low reward variance substantially underperforms the ground truth reward, in the sense that it leads to substantially slower optimization.
> > > >
> > > > *Additional experiments.* We considered a setting similar to Section 4.1 and trained, based on the Pythia-1B language model, 10 reward models over the same data (100% on-policy) using different training hyperparameters (e.g., learning rate and reward centering regularization coefficient). The table below presents the correlation between different reward model properties and increase in reward used for training and ground truth reward.
> > > >
> > > > |                             | Reward Increase Pearson | Reward Increase Spearman | GT Reward Increase Pearson | GT Reward Increase Spearman |
> > > > |-----------------------------|--------------------------|---------------------------|-----------------------------|------------------------------|
> > > > | Reward Variance             | 0.488                    | 0.588                     | 0.257                       | 0.576                        |
> > > > | On-Policy Acc.              | -0.458                   | -0.696                    | 0.903                       | 0.519                        |
> > > > | Reward Variance & On-Policy Acc. | 0.305              | 0.455                     | 0.536                       | 0.661                        |

---

> ### Comment · Reviewer_ngKr · 2025-08-05
> **Overall the evidence leaves me very confused, but I'm happy to recommend acceptance.**
>
> Thank you very much for the extremely quick and thorough follow-up. I appreciate the time you took to explain your work to me, and hope that it helps you present it to other people in the future, though you already did a great job.
>
> I am very confused about the results you present in the sense that they don't fit any model I can think of.
>
> Perhaps somehow, the initial model variance also indicates how easy it is to increase variance by moving slightly off-policy, in a way in which re-scaling does not affect? Seems possible but still a bit unlikely to me. It would be interesting to see how reward variance evolves over training to see if that explains the velocity almost entirely.
>
> Still, the experiment above is firmly in the square of Future Work and I think the authors have shown that the reward variance matters a lot, independent of just a scaling factor, to my satisfaction.
>
> > However, when considering the correlation between reward variance as measured with our original normalization and reward increase (after training using reward models normalized via the new normalization scheme), the results are analogous to our original experiments
>
> Yeah, this is very strange, as detailed above. Nonetheless I believe you and just disbelieve my previous hypothesiszx`.
>
> > To give a concrete example, consider the non-practical parameterization mentioned in the previous response, in which each output probability is its own trainable parameter
>
> Thank you for working through this example. I understand why I was wrong now: in this parameterization the gradient is nonzero whenever there are any reward differences anywhere, but the softmax parameterization eats this difference.
>
> Regarding the final experiment, it's interesting that the correlations are lower; but they're still firmly positive and so aren't really evidence against the main theorem being meaningful.
>
> I'm recommending this paper for acceptance. You probably want to put the experiments you did here (the one that controls for variance at initialization and still finds a correlation between variance before controlling, and final increase in GT reward) somewhere in the paper or appendix.

---

> > ### Author Response · Authors · 2025-08-06
> >
> > We are grateful for the insightful feedback and discussion. We believe that it has helped clarify important, yet delicate, points regarding our contributions and claims. In the revised manuscript, we will include a discussion on the relation between reward variance and scale as well as the latest experiments.
> >
> > One last remark, in the hopes of alleviating remaining confusion. Overall, our experiments demonstrate that optimization issues arising due to low reward variance are unlikely to be solved via simple scaling (and/or shifting), i.e., by multiplying all rewards by the same fixed scalar. There are several immediate reasons we see for this observation.
> > - If the reward model initially induces low reward variance, its rewards will be scaled by a large constant throughout training. This can cause instability during training when the (original) reward variance becomes higher, which is typically the case when the policy starts making progress.
> > - Furthermore, as discussed in the future work section, when the (original) reward variance is low the expected gradient vanishes. Thus, aggressive scaling can exacerbate noise in sample-based estimates of the gradient.
> > - Lastly, the reward variance for prompts in the training set can differ. In particular, even if the mean reward variance over the training set is low, there can be prompts for which the reward variance is high.
> >
> > In light of the above, we believe that online per-prompt normalization schemes, such as the one mentioned in our initial response, may have promise, especially when coupled with better gradient estimates via sampling a larger number of outputs per prompt. However, as noted in that response, more work is required for establishing robust normalization guidelines that can lead to better performance within a given compute budget. We view exploring such schemes and their relation to how reward variance evolves over training (as noted in your comment) as valuable directions for future work.

---

### Decision · Program_Chairs · 2025-09-17

**Decision:**

Accept (spotlight)

**Comment:**

This paper investigates factors beyond accuracy that make the reward model an effective teacher in RLHF. In particular, it focuses on reward variance induced by the reward model and shows that this variance is important for efficient optimization, even when the model itself is accurate. Both theoretical analysis and empirical evidence with LLMs support these findings. The work addresses an important and underexplored problem in RLHF. It is well written and offers useful insights into reward models. While the reviewers raised several concerns---some suggesting that further experimentation was needed to validate the main takeaways---most were resolved during the discussion phase, where the authors provided additional experimental results that corroborate their claims. The authors are encouraged to include those results in the final version of the paper.